# Activated glucocorticoid receptor is an estrogen receptor silencer in ER+ metastatic breast cancer

Madhuri Manivannan [1,2], Charly Jehanno [1,2], Michal Kloc [1,2,3], Jorge Gomez Miragaya [1,2], Maren Diepenbruck[1,2], Katrin Volkmann[1,2], Marie-May Coissieux [1,2], Marta Palafox[1,2], Adelin Rouchon[1,2], Nicolas Kramer[1,2], Alexander Schmidt [4], Yannick Blum[1,2], Baptiste Hamelin [1,2], Helen Schuster [5], Martin Heidinger[1,2,6], Simone Muenst[5], Marcus Vetter [7,8], Christian Kurzeder[9], Walter P Weber[2,6] & Mohamed Bentires-Alj [1,2]✉

## Abstract

Estrogen Receptor alpha (ER)-positive, HER2-negative breast cancers are less aggressive than other subtypes and show good patient clinical outcome because they are likely to respond to endocrine therapies. Unfortunately, therapy-resistant metastases may develop and start an inexorable downhill course. *ESR1* mutations leading to resistance to endocrine therapy are prevalent in 20–55% of patients with ER+ metastatic breast cancer. Here, we found that glucocorticoid receptor (GR) activation by dexamethasone in *ESR1* mutant metastases-bearing mice decreases liver metastases and prolongs survival. Transcriptomic and proteomic profiling revealed that GR activation not only downregulates estrogen response signature but also induces dramatic loss of ER itself. ChIP-Seq analyses show that prolonged dexamethasone treatment almost completely abrogates ER chromatin binding and that GR binds a subset of ER-related genes, including *ESR1*. Finally, the GR activity signature predicts a good outcome in patients with ER+ breast cancer. In summary, we show that dexamethasone inhibits ER+ metastatic growth by depleting ER, and hence could be tested for treating patients with ER+ metastatic breast cancer, particularly those suffering from refractory *ESR1* mutant metastases.

**Keywords** Steroid Nuclear Receptors; GR; ER; Metastasis; Breast Cancer
**Subject Categories** Cancer; Metabolism

## Introduction

The transcription factor ERα (ER) is crucial to mammary gland development. It is expressed in hormone-sensing cells, and, in the presence of estrogen, it increases proliferation and activity of neighboring ER-negative responder cells via paracrine signals (Arnal et al, 2017). However, in the context of breast cancer, ER initiates uncontrolled mammary cell division and survival and is decisively involved in the development of hormone-dependent breast cancer (Carroll and Brown, 2006). ER+ breast cancer accounts for over 80% of cases (DeSantis et al, 2019) and patients are treated with standard-of-care endocrine therapies in combination with surgical removal of the primary tumor and other local and systemic treatment as indicated (National Comprehensive Cancer Network Guidelines, 2023). Among the mechanisms leading to disease progression, *ESR1* mutations are mostly found in patients with metastatic breast cancer that progressed after endocrine therapy, specifically aromatase inhibitors (McAndrew and Finn, 2022). They are present in up to 55% of resistant metastases (Li et al, 2022), but are rarely found in primary tumors (Martin et al, 2017; Zundelevich et al, 2020; Jeselsohn et al, 2014; Razavi et al, 2018). The most common sites of mutations are residues 537 (Y537S) and 538 (D538G) in the ligand-binding domain (LBD) of ER (Toy et al, 2013; Robinson et al, 2013). These mutations confer estrogen-independent activity (Zhang et al, 1997) and resistance to endocrine therapies (Bahreini et al, 2017; Katzenellenbogen et al, 2018) such as Tamoxifen, Fulvestrant (Aggelis and Johnston, 2019), and aromatase inhibitors (Ma et al, 2015), besides promoting metabolic rewiring (Hanker et al, 2020). Metastatic patients with refractory disease harboring *ESR1* mutations typically relapse in the liver, lungs, or bones (Robinson et al, 2013). Liver is a very common site of ER+ breast cancer metastasis, with a clinical incidence rate of 40–50% (Cummings et al, 2014) and a high frequency of *ESR1* alterations (Merenbakh-Lamin et al, 2013).

For patients with hormone receptor (HR) + HER2− metastatic breast cancer, combinatorial treatment with CDK4/6 inhibitors and endocrine therapy are administered as a first-line therapy (Finn et al, 2015; Spring et al, 2020). Selection of second-line treatment is based on biomarkers. For example, patients with *PIK3CA* mutation are administered a PI3K/p110α inhibitor and endocrine therapy such as Fulvestrant. Upon progression of the disease, or under visceral crisis (McAndrew and Finn, 2022; Rugo et al, 2016),

[1]Department of Biomedicine, University of Basel, University Hospital Basel, Basel, Switzerland. [2]Department of Surgery, University Hospital Basel, Basel, Switzerland. [3]Swiss Institute of Bioinformatics, Basel, Switzerland. [4]Proteomics Core Facility, Biozentrum, University of Basel, Basel, Switzerland. [5]Institute of Pathology and Medical Genetics, University Hospital Basel, University of Basel, Basel, Switzerland. [6]Breast Center, University Hospital Basel, University of Basel, Basel, Switzerland. [7]Cancer Centre Baselland, Medical University Clinic, Cantonal Hospital Baselland, Liestal, Switzerland. [8]Medical Faculty, University of Basel, Basel, Switzerland. [9]Department of Gynecology, University Hospital Basel, University of Basel, Basel, Switzerland. ✉E-mail: m.bentires-alj@unibas.ch

cytotoxic chemotherapy, e.g., taxanes (Ghersi et al, 2015; Belfiglio et al, 2012; Arbeitsgemeinschaft Gynäkologische Onkologie e. V. (AGO), Kommission Mamma. Guidelines Breast Version 2020.1, 2020), is administered, often in combination with synthetic glucocorticoids to alleviate secondary effects (Grunberg et al, 2009). The glucocorticoid receptor (GR) is a ligand-activated transcription factor encoded by the *NR3C1* gene and is a member of the nuclear receptor superfamily. Glucocorticoids have anti-inflammatory actions (Russell and Lightman, 2019) and are critically involved in the regulation of developmental processes, cell differentiation (So et al, 2007), the immune system, glucose metabolism, ATP production, and systemic response to stress (Quax et al, 2013). Interestingly, elevated GR expression is associated with a favorable outcome in early-stage ER+ breast cancer patients (West et al, 2016). Furthermore, activated GR has been reported to bind to a subset of ER target genes and to counteract ER signaling and estrogen-induced proliferation (Swinstead et al, 2016), presumably by decreasing the occupancy of ER at certain enhancer regions (Tonsing-Carter et al, 2019). This is of particular interest in ER+ tumors, where the interplay of nuclear receptors has been shown to have dramatic effects on disease progression. Indeed, activation of both the progesterone receptor (PR) and the androgen receptor (AR) have been shown to reprogram the ER cistrome: reinstalling a gene-expression program of favorable outcome (Mohammed et al, 2015) and repressing ER-regulated cell cycle genes, whilst upregulating known tumor suppressors (Hickey et al, 2021). Whether GR activation impinges on ER+ metastatic progression, the underlying mechanism, its effect on endocrine therapy, and its clinical relevance remain largely elusive. Besides, we have previously shown in triple-negative breast cancer (TNBC) that GR activation promotes lung metastatic colonization (Obradović et al, 2019), through a mechanism that involves the ROR1/WNT5a axis. It therefore appears that the phenotypic effect of GR activation on metastasis is highly subtype-dependent and that the crosstalk with ER may have distinct effects on disease progression. Hence, these observations prompted us to elucidate the consequences of GR activation in the context of ER+ metastatic disease.

In this study, we have investigated the effects of the synthetic GR agonist dexamethasone (Dex) on endocrine therapy-resistant *ESR1* mutant metastatic breast cancer. To this end, we used preclinical ER+ breast cancer in vivo and ex vivo models and found that GR activation reduces liver metastases and, thus, reduces the burden of overt metastases and prolongs animal survival. By performing transcriptomic profiling and global proteomics in *ESR1* mutant cells, we found that Dex significantly downregulates estrogen response genes and estrogen response-related proteins. Strikingly, our analyses also revealed that prolonged exposure to Dex induces *ESR1* gene downregulation and dramatic ER protein loss, uncovering an unexpected estrogen receptor silencing action of GR. In addition, ER and GR ChIP-Seq analyses revealed that prolonged Dex exposure almost completely eradicates ER binding from its regulatory elements genome-wide, therefore hindering ER-dependent gene transcription. We observed the loss of ER abundance upon Dex treatment in several ER+ patient-derived tumor organoids and found anti-correlation between ER and GR expression in metastatic ER+ patient samples. Finally, we generated a GR activity signature using transcriptomic profiling and found that its expression predicts good outcome for ER+

patients (METABRIC), and that it is inversely correlated with *ESR1* mRNA level. Altogether, these findings demonstrate that activated GR silences ER and that Dex may be repurposed and tested in patients with advanced ER+ breast cancer, particularly those with *ESR1* mutant metastases.

# Results

## GR activation reduces ER+ breast cancer metastatic burden and prolongs the overall survival of animals

To investigate whether GR signaling impinges on ER+ breast cancer metastatic disease progression and animal survival, we used the clinically relevant metastatic and endocrine therapy-resistant (Jeselsohn et al, 2014; Toy et al, 2013) *ESR1* mutant MCF-7 D538G and Y537S cells (Scott et al, 2017). These *ESR1* hotspot mutations are frequently detected in patients who progressed after endocrine therapy, specifically aromatase inhibitors, and result in resistance to Fulvestrant and Tamoxifen (Scott et al, 2017; Bahreini et al, 2017). We examined the protein abundance of GR and ER in MCF-7 D538G and Y537S cells compared to *ESR1* wild-type (wt) parental MCF-7 and T-47D cells (Fig. EV1A). Baseline GR abundance was higher in *ESR1* wt and mutant MCF-7 cells compared to T-47D cells (Fig. EV1A). Next, we verified GR nuclear translocation after Dex administration both in vitro (Fig. EV1B) and in vivo in liver metastases of an ER+ xenograft from mice intravenously injected with MCF-7 D538G cells (Fig. EV1C). We also detected upregulation of the classical GR target gene, serum glucocorticoid kinase 1 (*SGK1*) (Webster et al, 1993), after Dex treatment using qPCR (Fig. EV1D). To assess the effect of GR activation on disease progression in vivo (Fig. 1A), we generated GFP luciferase-expressing MCF-7 D538G and Y537S cells that were transduced with two different short hairpin RNAs (shRNAs) targeting GR and a non-targeting control. In GR-targeting shRNA-transduced cells, we observed significant downregulation of GR both at the mRNA and protein levels (Figs. 1B and EV1E).

As expected, *SGK1* mRNA expression was reduced upon Dex in these cells (Fig. EV1F). We then injected the cells into the tail vein of NOD-*scid-Il2rg*null (NSG) mice and immediately assessed their presence in the animals by bioluminescence imaging (Figs. 1A and EV1G). Then the mice were treated with clinically relevant doses of Dex or vehicle for 2 weeks. Metastatic burden was quantified by bioluminescence imaging once a week (Fig. EV1G,H). Mice were sacrificed when signs of distress appeared. Strikingly, we found that Dex treatment significantly prolonged overall survival of the animals compared to the vehicle-treated group (Fig. 1C). The increased overall survival was seen to be GR-mediated, as its knockdown abrogated this phenotype (Fig. 1C). Necropsy and hematoxylin-eosin staining revealed a high burden of liver metastases, which most likely accounts for the distress symptoms (Fig. 1D). Notably, *ESR1* mutations are associated with metastases in the liver (Merenbakh-Lamin et al, 2013), a frequent site of disseminated cancer cell growth (DiSibio and French, 2008) that is often associated with a poor prognosis in patients (Diamond et al, 2009; Zhao et al, 2018).

Next, we injected MCF-7 Y537S and D538G cells intravenously into NSG mice and 15 days later treated them for 2 weeks with Dex or vehicle to assess the effect of GR activation on established

**A** **Experimental Design**

*ESR1* mutant GFP-luci cells → *In vitro* GR activation (Dex 700 nM) → Cancer cell injection → Dex (0.1mg/kg) Vehicle (PBS) I.P. → Overall survival / Quantification of metastatic burden in distant organs

**B** MCF-7 D538G

shCtrl | shGR1 | shGR2

100 kDa / 75 kDa — GR
1 | 0.51 | 0.18

45 kDa / 35 kDa — ERK2

**C** MCF-7 D538G

Overall survival (%) vs Time (Days after injection)

- shCtrl Vehicle
- shCtrl Dex
- shGR1 Vehicle
- shGR1 Dex
- shGR2 Vehicle
- shGR2 Dex

0.0023
0.0026
0.0018

**D** Vehicle | Dex

shCtrl
shGR1
shGR2

100μm | 5mm | 100μm | 5mm

**E** MCF-7 Y537S

Vehicle | Dex

Lungs Liver Kidneys | Lungs Liver Kidneys

Bones (Femur) Brain Spleen & Ovaries | Bones (Femur) Brain Spleen & Ovaries

ph.s⁻¹.cm⁻².sr⁻¹
1.36e+08
7e+07
0

Image
Min: 2.7e+05
Max:1.4e+08
Display
Min: 2.65e+05
Max:1.35e+08
Gamma: 1.00

**F** MCF-7 Y537S

Normalized mean radiance (relative to day 0)

○ Vehicle  ● Dex

Liver: 2.53e-08
Lungs: 0.902
Bones: 0.835
Kidneys: 0.875
Spleen: 0.854
Ovaries: 0.573
Brain: 0.885

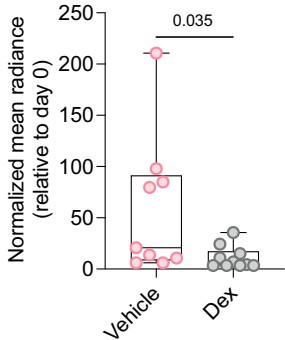

**G** MCF-7 D538G Liver metastases

Normalized mean radiance (relative to day 0)

0.035

Vehicle | Dex

◀

**Figure 1.  GR activation reduces metastatic burden, enhances chemoresponsiveness, and prolongs the survival of mice injected with *ESR1* mutant models.**

(A) Experimental design. (B) Immunoblots showing the knockdown of GR in MCF-7 D538G cells transduced with two different shRNAs targeting *NR3C1*. ERK2 was used as a loading control. (C) Kaplan–Meier survival analysis of mice intravenously injected with MCF-7 D538G cells transduced with control shRNA or shRNAs targeting GR (shGR1 and shGR2) and treated with vehicle or Dex; $n = 5$ mice per group. Log-rank (Mantel–Cox) test. (D) Haematoxylin & eosin staining of liver metastases from mice intravenously injected with MCF-7 D538G cells transduced with control shRNA or shRNAs targeting GR (shGR1 and shGR2) and treated with vehicle or Dex; $n = 5$ mice per group. (E) Representative bioluminescence images of distant organs from vehicle or Dex-treated MCF-7 Y537S metastases-bearing mice harvested at day 47; $n = 8–9$ mice per group. (F) Scatter dot plot depicting the quantification of metastases from distant organs upon GR activation by Dex compared to vehicle at day 47; $n = 9–10$ mice per group. Two-way ANOVA. Data are presented as mean ± SD. (G) Box plot showing the quantification of liver metastases in MCF-7 D538G-bearing mice after GR activation by Dex compared to vehicle at day 47; $n = 9–10$ mice per group. Two-tailed Mann–Whitney test. Boxes define the upper and lower quartiles; a central band indicates the median; whiskers define max to min values. Source data are available online for this figure.

metastases. Distant organs were resected to evaluate the metastatic burden (Fig. 1A,E). We found an increased penetrance and burden of metastases in the liver compared to other organs (Fig. 1F). Moreover, liver metastases decreased in both MCF-7 Y537S (Fig. 1F) and D538G metastases-bearing mice upon Dex treatment (Figs. 1G and EV1I).

Finally, we assessed the effect of Dex on response to Paclitaxel in ER+ cells, given that it offsets its efficacy in TNBC models (Obradović et al, 2019; Pang et al, 2006). Dex is indeed given in combination with Paclitaxel to alleviate its side effects (Gennari et al, 1996). We administered Dex together with Paclitaxel to NSG mice intravenously injected with MCF-7 Y537S and D538G cells. While Paclitaxel treatment alone had no effect on hepatic metastases (Appendix Fig. S1a,b), the addition of Dex reduced liver metastatic burden and prolonged animal survival (Appendix Fig. S1c–f).

Altogether, these results suggest that GR activation in *ESR1* mutant models prolongs animal survival by decreasing liver metastases via a cancer cell-autonomous mechanism.

## GR activation decreases the number of ER+ cancer cells and patient-derived organoids

To investigate the cellular mechanisms underlying the anti-metastatic effects of GR activation, we cultured ER+ cells in vitro and assessed multiple parameters. Cell viability assays revealed that cells treated with Dex are significantly less viable than their vehicle-treated counterparts (Fig. 2A). A sulforhodamine B (SRB) assay confirmed that the number of Dex-treated cells was significantly lower than the number of vehicle-treated cells on days 4 and 5 after treatment (Fig. 2B; Appendix Fig. S2a). We next assessed the effects of GR activation on the viability of ER+ patient-derived xenograft tumor organoids (PDXO) cultured in Matrigel ex vivo by treating PDXOs with Dex or vehicle for 10 days (Fig. 2C). DAPI-stained organoids were imaged by confocal microscopy and analyzed at single-cell resolution. The nuclei count of GR-activated ER+ organoids (PDXO_1 and PDXO_2) were significantly lower than those of the vehicle-treated organoids (Fig. 2D,E). FACS-based cell cycle analysis of EdU staining showed that the percentage of cancer cells in S-phase was significantly reduced by GR activation compared to the vehicle-only treatment (Fig. 2F), while no significant difference in the percentage of cancer cells in G2/M phase was observed upon GR activation (Appendix Fig. S2b). In addition, analysis of Annexin V staining showed increased cell death upon Dex treatment as compared to control, indicating that GR activation at the cellular level can act on both proliferation and apoptosis (Fig. 2G). Finally, as GR activation decreases liver

metastases, we co-cultured the GFP-labeled MCF-7 D538G cells with mCherry-labeled hepatocytes (Appendix Fig. S2c). In this context, Dex treatment also decreased the viability of cancer cells in a liver-like milieu (Appendix Fig. S2d–f).

## GR activation decreases ER transcriptional activity

To decipher the molecular mechanism(s) underlying the effects of GR activation, we performed transcriptomic profiling of MCF-7 D538G and Y537S cells treated with Dex or vehicle for 8 h (short exposure) or 24 h (prolonged exposure). Principal Component Analysis (PCA) showed separation of the treatment groups, timepoints, and cell lines (Fig. 3A). We identified the differentially expressed genes upon GR activation (FDR < 0.01, |log2FC|>1) (Dataset EV1) and performed functional annotation using the Hallmark gene signature (Gene set enrichment analysis, GSEA). Pathways such as oxidative phosphorylation, androgen response were upregulated upon GR activation in MCF-7 Y537S cells (Fig. EV2A). Conversely, we found downregulation of estrogen response early and estrogen response late terms in both cell lines and both time points, especially after prolonged Dex exposure (Fig. 3B). qPCR profiling confirmed the loss of expression of ER target genes upon Dex treatment (Fig. 3C). Finally, Ingenuity pathway analysis (IPA) validated that GR activation-dependent genes were negatively associated with ER and E2 signaling and positively correlated with stress hormones (Appendix Fig. S3a).

To further investigate the molecular mechanism underlying the effects of GR activation, we performed global proteomics profiling of MCF-7 D538G and Y537S cells treated with Dex or vehicle for 7 days in vitro. The respective PCA showed separation of the treatment groups (Fig. EV2B). Next, we identified differentially abundant proteins upon GR activation (FDR < 0.05, |log2FC| > 0.5) (Figs. 3D and EV2C; Dataset EV2), and found as expected, classical GR targets including *SGK1* and *FKBP5* upregulated upon Dex treatment. GSEA in both models, confirmed upregulation of pathways such as oxidative phosphorylation, fatty acid metabolism, and epithelial to mesenchymal transition (Fig. EV2D; Appendix Fig. S3b), as well as GR as top upstream regulator (Appendix Fig. S3c).

Next, GSEA analysis upon Dex stimulation revealed down-regulation of interferon alpha and gamma response, mTORC1 signaling, and estrogen response early terms (Figs. 3E and EV2E), therefore confirming the observation from RNA-Seq. Indeed, many proteins belonging to E2_response_early signature were found downregulated upon GR activation (Fig. EV2F,G), corroborating Dex inhibitory actions on ER signaling. Combined analysis of RNA-Seq and Proteomics datasets revealed high

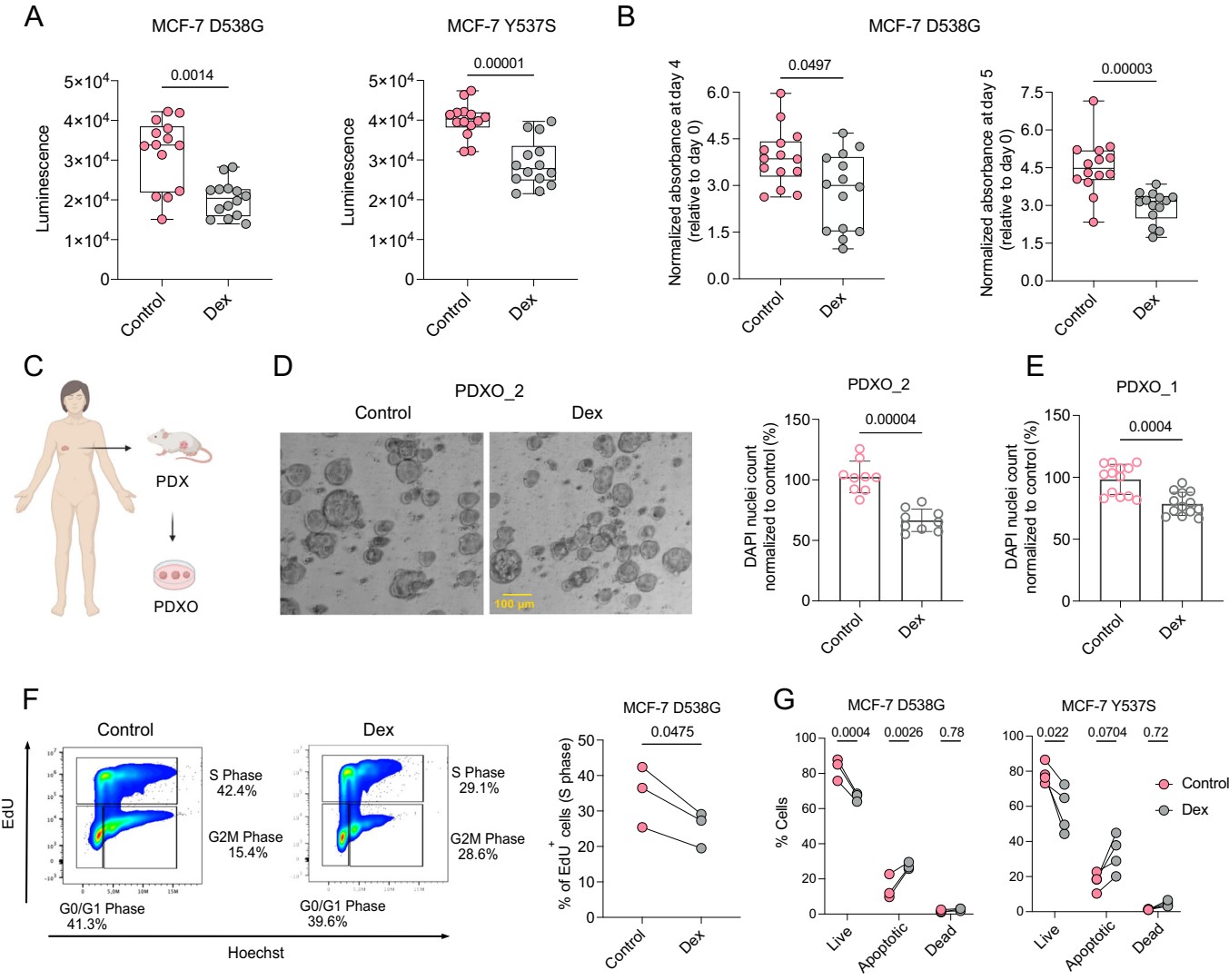

**Figure 2. GR activation decreases the viability of ER+ cancer cells and patient-derived tumor organoids.**

(A) Box plots depicting luminescence as a measure of viable cells after Dex treatment for 96 h of MCF-7 D538G and Y537S cells; $n = 3$ biological replicates with 4–5 technical replicates each. Two-tailed Mann–Whitney test. Boxes define the upper and lower quartiles; a central band indicates the median; whiskers define max. to min. values. (B) Box plots depicting normalized absorbance as a measure of cell number after Dex treatment for 4 and 5 days, respectively, of MCF-7 D538G cells; $n = 3$ biological replicates with 4–5 technical replicates each. Two-tailed Mann–Whitney test. Boxes define the upper and lower quartiles; a central band indicates the median; whiskers define max. to min. values. (C) Scheme of patient-derived xenograft tumor organoid (PDXO) generation from patients' ER+ breast tumors. (D) Left panel: representative brightfield images of tumor organoids (PDXO_2) cultured in Matrigel and treated for 10 days with vehicle or Dex. Right panel: scatter dot plot representing the nuclei counts of the indicated PDXO_2 upon GR activation or control treatments for 10 days; $n = 2$ biological replicates with 3–6 technical replicates each. Two-tailed Mann–Whitney test. Data are presented as mean ± SD. (E) Scatter dot plot representing the nuclei counts of PDXO_1 after GR activation or control treatments for 10 days; $n = 2$ biological replicates with 3–6 technical replicates each. Two-tailed Mann–Whitney test. Data are presented as mean ± SD. (F) Left panel: representative flow cytometry dot plots of EdU/Hoechst cell-cycle staining of MCF-7 D538G cells treated for 4 days with 700 nM Dex or vehicle. Single cells were gated based on forward and side scatter heights, and different cell cycle populations were gated and quantified based on EdU and Hoechst staining. Right panel: graph showing the percentage of EdU-stained live cells analyzed by FACS; $n = 3$ biological replicates. Two-tailed paired $t$ test. Individual values and lines represent pairing of data points. (G) Graphs showing the percentage of live, apoptotic, and dead MCF-7 D538G and Y537S cells treated for 4 days with 700 nM Dex or vehicle. Single cells were gated on forward and side scatter heights, and different cell populations were gated and quantified based on Annexin V and DAPI staining; $n = 3–4$ biological replicates. Two-tailed paired $t$ test. Individual values and lines represent paired data points. Source data are available online for this figure.

correlation in both MCF-7 D538G and Y537S models (Pearson correlation index >0.7) and confirmed that genes for which mRNA and proteins are silenced upon Dex are associated with estrogen response signature (Fig. EV2H). We found a similar Dex effect in an *ESR1* wt MCF-7 model, in which estrogen response early and late-related terms are downregulated following GR activation

(Appendix Fig. S3d). Remarkably, most of the genes from the E2-response signature that were downregulated after Dex treatment were identified as markers of bad prognosis in ER+ breast cancer (KMplotter (Győrffy, 2021)), specifically in patients who had undergone various therapies (Appendix Fig. S3e), suggesting that GR activation might be beneficial in advanced ER+ breast cancer.

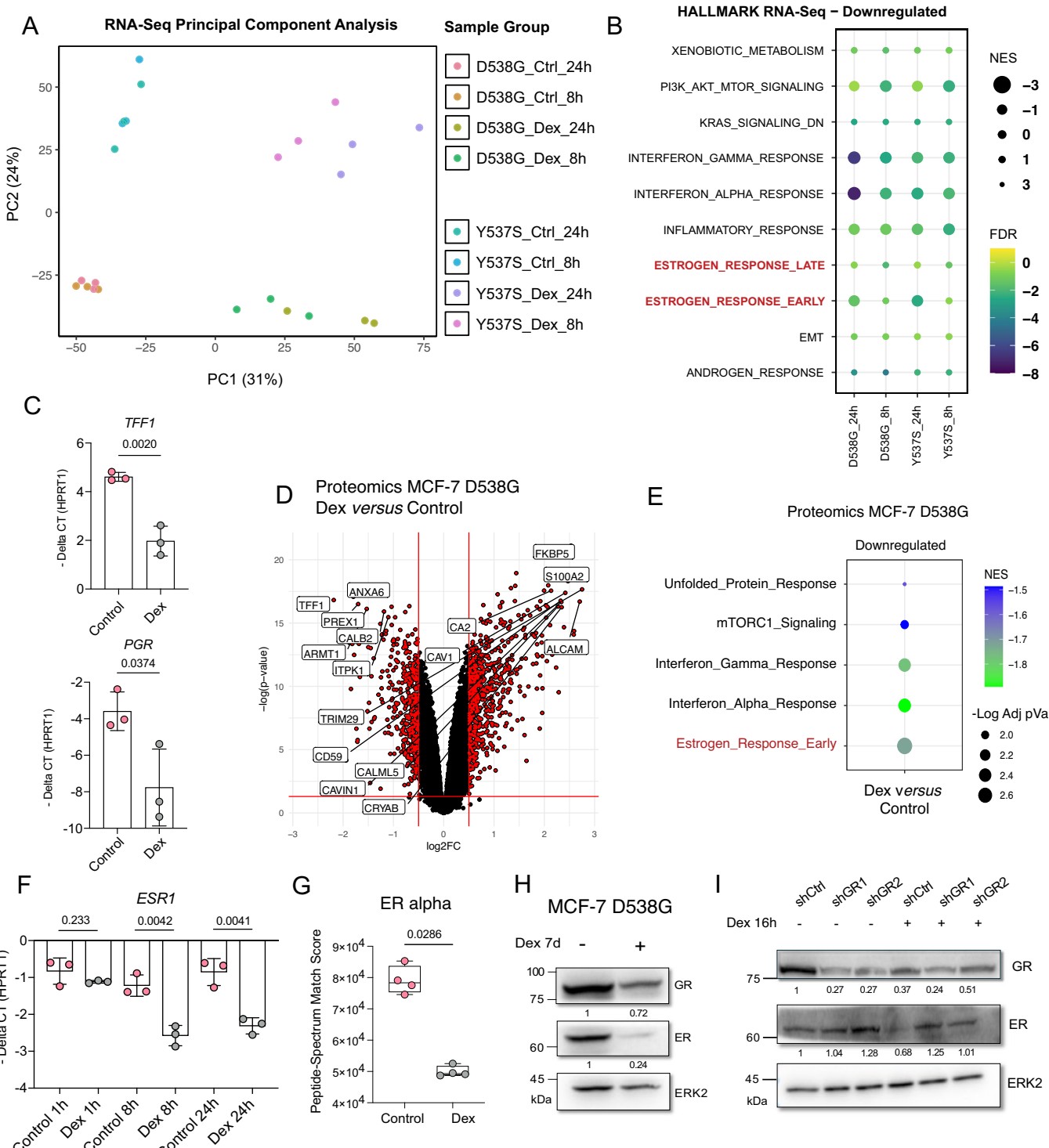

## Activated GR silences *ESR1* and ER target genes

RNA-Seq profiling surprisingly uncovered that the mRNA level of *ESR1* itself was decreased following 8 h and 24 h of GR activation (Fig. 3F), suggesting that GR may repress its transcription. We also found loss of ER protein abundance following GR activation in the proteomics data (Fig. 3G). ER abundance was also decreased upon

GR activation in MCF-7 D538G cells treated for 3 days and 7 days with Dex (Figs. EV3A and 3H). MCF-7 D538G cells transduced with shRNAs targeting GR and treated with Dex displayed no decrease in ER abundance compared to cells transduced with control shRNAs, confirming that ER loss upon Dex treatment is mediated by GR (Fig. 3I). Next, we tested the effect of Fulvestrant (the most widely used selective estrogen receptor degrader (SERD)

◀ **Figure 3.   GR activation inhibits ER signaling and triggers ER loss.**

(A) Principal Component Analysis of MCF-7 D538G and Y537S cells treated with Dex or vehicle for 8 h and 24 h and analyzed by RNA-Seq; $n = 12$ samples per cell line and three technical replicates per condition. (B) Bubble plot depicting the Hallmark signatures enriched in genes downregulated upon Dex treatment (8 h and 24 h), in MCF-7 D538G and Y537S models. Normalized enrichment score (NES) and False Discovery Rate (FDR) are indicated. (C) Scatter dot plots showing the mRNA levels of ER canonical targets, *TFF1* and *PGR*, after 72 h of Dex treatment in MCF-7 D538G cells. $n = 3$ biological replicates. Two-tailed *t* test. Data are presented as mean ± SD. (D) Volcano plot of the differentially abundant proteins (adjusted *P* value < 0.05 and |logFC| > 0.5) in Dex- *versus* vehicle-treated MCF-7 D538G cells. Empirical Bayes moderated *t*-statistics and Benjamini–Hochberg correction. (E) Bubble plot depicting the normalized enrichment score (NES) of downregulated (adjusted *P* value < 0.05) Hallmark gene sets after GR activation in the MCF-7 D538G model. Kolmogovo–Smirnov–like running-sum statistics with permutation test and Benjamini–Hochberg correction. (F) Scatter dot plot showing the *ESR1* mRNA levels after GR activation for the indicated times in MCF-7 D538G cells; $n = 3$ biological replicates. Two-tailed *t* test. Data are presented as mean ± SD. (G) Box plot representing the Peptide-Spectrum Match (PSM) score as a measure of ER protein abundance in MCF-7 D538G cells from the proteomics data; $n = 4$ technical replicates per condition. Two-tailed *t* test. Boxes define the upper and lower quartiles; the central band indicates the median; whiskers define max. to min. values. (H) Immunoblots showing the abundance of GR, ER, and ERK2 (loading control) proteins in MCF-7 D538G cells after treatment with Dex for 7 days. (I) Immunoblots showing the abundance of GR, ER, and ERK2 (loading control) proteins in MCF-7 D538G cells transduced with control shRNA or shRNAs targeting *NR3C1* (shGR1 and shGR2) after 16 h of GR activation with Dex. Source data are available online for this figure.

in the clinics) and other new SERDs, Elacestrant and Camizestrant on ER abundance in comparison to Dex. We found that the ability of Dex to reduce ER protein abundance is comparable to these SERDs (Fig. EV3B). In addition, we found that activated GR also decreases the protein abundance of wt ER, which rules out any mutation-specific effect (Fig. EV3C,D). Interestingly, we detected ER protein loss starting from 8 h post-Dex treatment, concomitant with *ESR1* gene silencing (Figs. EV3E and 3F). These results reveal an unexpected direct loss of ER itself upon GR activation and suggest that Dex may silence transcription of ER target genes simply by silencing ER.

RNA-Seq profiling showed two patterns of downregulation of ER target genes following GR activation, those repressed after short (8 h) and those after long (24 h) Dex treatment. Loss of expression at 8 h suggested direct GR-mediated repression, while loss of expression at 24 h Dex only may be a consequence of ER loss. To test these possibilities, we performed qPCR on MCF-7 D538G cells treated with Dex at 8 or 24 h. *SGK1* upregulation confirmed GR activation (Fig. EV3F). ER target genes such as *PGR, CXCL12*, and the pioneer factor *FOXA1* were silenced rapidly following GR activation (8 h of Dex) (like *ESR1* itself, Fig. 3F), suggesting either direct GR-mediated repression or GR-mediated displacement of ER at the chromatin level (Tonsing-Carter et al, 2019) (Figs. 4A and EV3G). Conversely, other ER target genes, such as *AREG* or *TFF1* were downregulated only at 24 h post GR activation (Fig. 4B). This delayed loss of expression may therefore result from ER loss. We also found that genes belonging to the E2 response early gene signature were repressed either 8 h or 24 h after GR activation, while *GREB1* was the only classical ER target gene upregulated by Dex (Fig. EV3H,I).

## Dex treatment abrogates ER genome-wide chromatin binding

To functionally address how Dex represses ER-dependent transcription and impinges on ER chromatin binding over time, we performed ChIP-Seq profiling for both ER and GR in the absence or presence of Dex (Fig. EV4A). Peak calling and filtering in all conditions enabled the identification of 1346 and 9332 GR-bound regions in the absence or presence of Dex for 1 h, respectively. Also, 5516, 7781, and 3164 ER-bound regions were identified in the absence or presence of Dex for 1 h and 24 h, respectively (Fig. EV4B). GR was recruited to the chromatin following Dex stimulation as shown by peak number and signal intensity

(Fig. EV4C,D). Transcription factor analysis on GR-bound genes, revealed FOXA1 and ER binding at their vicinity as top hits, confirming overlap of regulation between FOXA1, ER, and GR at the chromatin level (Swinstead et al, 2016) (Fig. EV4E). Interestingly, 50% and 29% of the early and late repressed ER target genes identified in RNA-Seq were located at the vicinity of GR peak, indicating that GR may actively induce gene silencing (Fig. EV4F). Next, we investigated ER binding dynamics upon 1 h and 24 h of Dex compared to control. Notably, we found a redistribution of ER cistrome following 1 h and 24 h Dex with 4215 and 1640 newly bound regions, respectively, and a massive loss of ER-bound regions following 24 h of GR activation (Fig. EV4G). Interestingly, only the newly ER-bound regions 1 h post Dex treatment remain associated with ER target genes, in contrast to 24 h Dex treatment (Fig. EV4H).

We then assessed signal intensity and found a partial displacement of ER binding after 1 h of Dex, suggesting displacement of ER by GR. In contrast, we found a dramatic loss of ER recruitment after 24 h of Dex (Fig. 4C,D), consistent with ER loss and decreased expression and abundance of its targets (Fig. 3B,E). Some genomic regions bound by both ER and GR (1 h Dex) were enriched in motifs corresponding to both transcription factors (Figs. 4C,E and EV4I). This active GR recruitment to ER-bound regions may also contribute to transcriptional silencing of ER target genes. Peak to gene annotation using GREAT revealed 330 genes commonly bound by ER and GR, and functional annotation confirmed that they are associated with estrogen response (HALL-MARK) and ER activity (ChEA) (Figs. 4F and EV4J).

Among the genes commonly bound by ER and GR, we found *ESR1* itself. First, ChIP-Seq and ChIP-qPCR showed decreased ER binding to its own regulatory elements after 24 h of Dex (Figs. 4G,H and EV4K). Given that ER can autoregulate itself (Castles et al, 1997), it is likely that this loss of recruitment accounts for *ESR1* downregulation. Second, we found reduced H3K27Ac enrichment on *ESR1* regulatory elements, which is consistent with *ESR1* downregulation and loss of enhancer activation (Fig. 4I). Third, ChIP-Seq and ChIP-qPCR revealed Dex-induced GR recruitment at the vicinity of the *ESR1* gene (Fig. 4G,J), indicating that GR may also directly silence *ESR1* expression.

In addition, ChIP-qPCR showed reduced ER recruitment and H3K27Ac at the vicinity of several of its target genes including *TFF1, PGR*, and *AREG*, thus explaining their loss of expression (Fig. 4H,I). Finally, we detected Dex-induced GR recruitment at the vicinity of *AREG* and *GREB1*, which are downregulated and

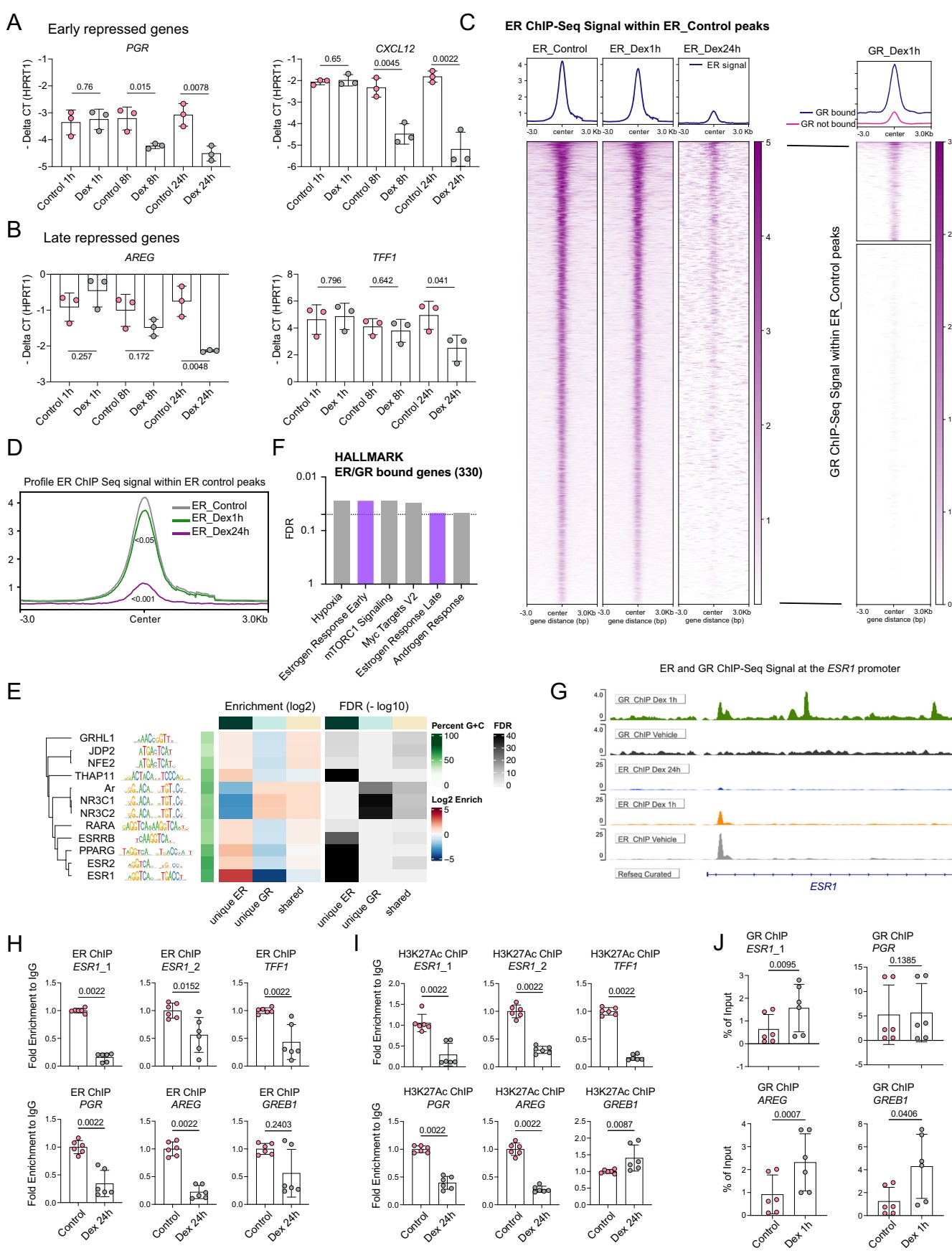

◄ **Figure 4. GR activation abrogates ER genome-wide binding to the chromatin and represses *ESR1* gene.**

(A, B) Scatter dot plots showing the mRNA levels of ER canonical targets that are repressed early (8 h, *PGR* and *CXCL12*) or late (24 h, *AREG* and *TFF1)* after GR activation for the indicated times in MCF-7 D538G cells: $n = 3$ biological replicates. Paired t-test. Data are presented as mean ± SD. (C) Heatmap of ER ChIP-Seq in the absence or presence of Dex for 1 h or 24 h, and of GR ChIP-seq in the absence or presence of Dex for 1 h, in MCF-7 D538G cells. The heatmap depicts the ChIP signal on ER-bound sites in the control condition. The heatmap is shown in a horizontal window of $-/+$ 3 kb around the peaks. (D) Average read density plots for ER (in the absence or presence of Dex for 1 h or 24 h) at the ER binding sites identified in the control (absence of Dex) condition. Kolmogorov–Smirnov test. (E) MonaLisa motif enrichment analysis in peaks identified to be bound by ER only, GR only, or both transcription factors. (F) Functional annotation of genes located near a peak bound by both ER and GR. Peak to gene annotation was performed using GREAT (http://great.stanford.edu/public/html/) with the basal plus extension settings (Proximal 5 kb upstream, 1 kb downstream, Distal up to 1 kb). (G) IGV browser image showing average ER (control, Dex 1 h, Dex 24 h) and GR (control, Dex 1 h) ChIP-Seq signal at the vicinity of the *ESR1* gene. (H, I) ER and H3K27Ac ChIP-qPCR at the promoter regions of *ESR1* (2 distinct sites) and known ER targets, *TFF1, PGR, AREG*, and *GREB1*, following 24 h of Dex treatment. Data are shown as fold enrichment over the IgG control; $n = 3$ biological replicates with two technical replicates each. Two-tailed Mann–Whitney test. Data are presented as mean ± SD. (J) GR ChIP-qPCR at the promoter regions of *ESR1* and ER targets, *PGR, AREG, and GREB1*, following 1 h of Dex treatment. Data are depicted as % of input DNA; $n = 3$ biological replicates with two technical replicates each. Two-tailed Mann–Whitney test. Data are presented as mean ± SD. Source data are available online for this figure.

upregulated by Dex, respectively, indicating that GR can modulate ER target gene expression in both directions (Fig. 4J). GR recruitment at the proximity of *SGK1* was used as a positive control (Fig. EV4I). Altogether, the data revealed that prolonged Dex exposure eradicates ER genome-wide binding to its regulatory elements.

## High GR activity is associated with enhanced survival in patients with ER+ breast cancer and is anti-correlated with *ESR1* expression

To assess the clinical relevance of GR expression in ER+ breast cancer, we stratified ER+ patients according to *NR3C1* mRNA level (high or low) in the METABRIC dataset. Kaplan–Meier analysis revealed that high expression of GR is associated with prolonged survival in ER+ breast cancer patients (West et al, 2016) (Fig. EV5A). As GR expression does not necessarily reflect its activity, we generated a GR activity signature and performed a multivariate analysis. The GR activity signature is composed of 52 protein-coding genes, upregulated upon Dex treatment for 8 h and 24 h, in both MCF-7 D538G and Y537S models (see "Methods" for signature generation) (Fig. EV5B; Dataset EV3). Cox regression analysis demonstrated that high expression of the GR activity signature is predictive of good prognosis in ER+ luminal patients (METABRIC), for both luminal A and B subtypes (Fig. 5A; Appendix Fig. S4a,b). We then stratified ER+ luminal tumors into three categories according to the GR activity signature as low, intermediate, and high. ER+ tumors with a high GR activity signature displayed reduced *ESR1* expression and vice versa in tumors with low GR activity score (Fig. 5B). As the GR activity signature includes five basal-like associated keratin-encoding genes (*KRT5, KRT6a, KRT6b, KRT6c* and *KRT16*) as direct GR targets, we removed them from the signature to assess whether their presence could bias the observed effects. Remarkably, the predictive power of the GR activity signature without these keratins on ER+ patients' outcome and its anti-correlation with *ESR1* expression remained unchanged (Fig. EV5C,D).

Next, we performed immunostaining and quantification of nuclear ER and GR in samples from patients with ER+ breast cancer. The cohort consists of primary tumors, matched metastases, and pre- and post-treated metastases obtained from surgical resections or biopsies of patients. Notably, we found that high abundance of ER is associated with low GR abundance in these patient samples (Fig. 5C; Appendix Fig. S5a,b).

## Activated GR decreases ER abundance in patient-derived tumor organoids

To assess Dex-mediated *ESR1* silencing in clinically relevant models, we employed ex vivo cultures of PDXOs and patient-derived organoids (PDO) established from surgical primary tumor resections or biopsies from metastases of ER+ patients (Fig. 5D). We treated the cultured PDOs/PDXOs as well as the MCF-7 D538G cells grown as 3D organoids with vehicle, Dex, or Fulvestrant alone or in combination. The nuclei count as well as ER abundance were analyzed by confocal microscopy for the different treatments. Nuclear ER was significantly lower in Dex-treated as compared to vehicle-treated MCF-7 D538G grown as organoids (Figs. 5E and EV5E), as well as in PDO and PDXO models, thus confirming the Dex-induced ER loss in clinically relevant models (Fig. 5F–H). In addition, qPCR analysis on PDXO_3 revealed the downregulation of *ESR1* and its target gene *TFF1* upon Dex treatment (Fig. 5I), similar to our findings in MCF-7 *ESR1* mutant cells. Upregulation of *SGK1* and *GREB1* served as positive controls for GR activation (Fig. 5I). Other ER-related genes including *AREG, CXCL12,* and *FOXA1* were not differentially regulated upon Dex treatment in the PDXO_3 (Fig. EV5F).

## Dex is more potent at reducing ER+ metastases than Fulvestrant and synergizes with CDK4/6 inhibitors

Fulvestrant is used to treat patients with ER+ metastatic disease. Hence, we compared the therapeutic effect of Dex *versus* Fulvestrant by evaluating the viability of cancer cells in vitro upon Dex and Fulvestrant treatments. Remarkably, the viability of MCF-7 D538G cells treated with Dex was reduced more than treatment with vehicle but also more than Fulvestrant (Fig. EV6A), reinforcing the clinical potential of GR activation as an estrogen receptor silencer (ERS) in *ESR1* mutant breast cancer.

Next, we compared the in vivo efficacy of Dex *versus* Fulvestrant on metastatic burden by injecting MCF-7 D538G cells intravenously into NSG mice without any prior in vitro treatments. After metastases were established, the animals were treated with clinically relevant doses of Dex, Fulvestrant or the combination for 6 weeks (Fig. EV6B). The progression of metastatic outgrowth was monitored weekly and at the endpoint by bioluminescence imaging (Fig. EV6C). Dex was more potent at decreasing liver metastases in comparison to the control and Fulvestrant treatment arms (Fig. EV6D). *ESR1* mutations Y537S and D538G in the LBD of

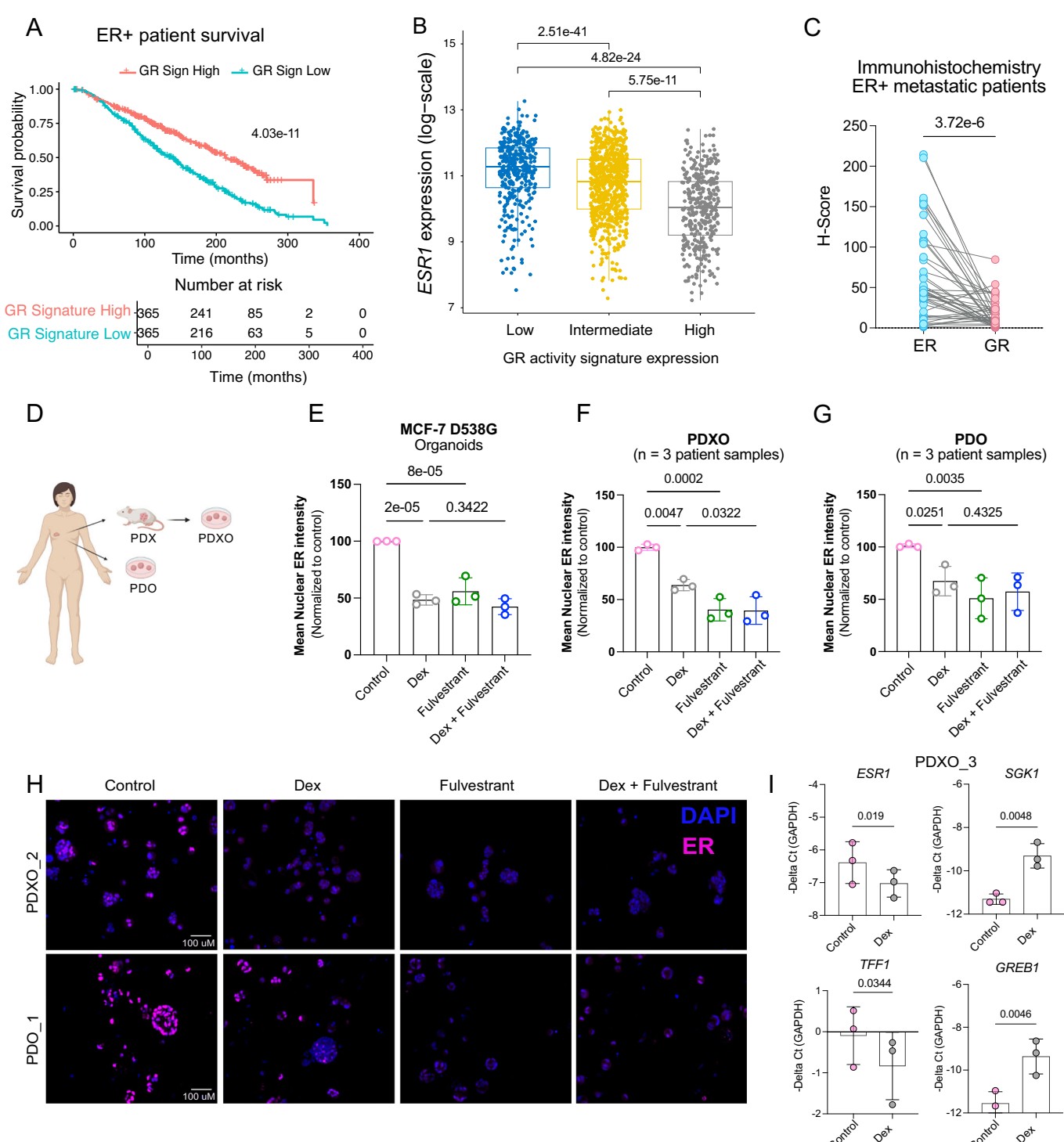

ER confer resistance to endocrine therapy in patients (Toy et al, 2013) and to Fulvestrant in preclinical models. These results demonstrate the potential therapeutic benefit of GR activation by Dex administration to reduce *ESR1* mutant liver metastases and to prolong survival.

Finally, CDK4/6 inhibitors such as Ribociclib, Palbociclib, and Abemaciclib are administered in combination with endocrine therapy as first-line therapy for treatment of advanced ER+ breast cancer (Hortobagyi et al, 2016; Finn et al, 2015; Turner et al, 2015; Dickler et al, 2017; Sledge et al, 2017). Hence, we evaluated the effect of CDK4/6 inhibitors in combination with Dex or Fulvestrant on MCF-7 D538G and Y537S cells. We found that the combination of Dex with Ribociclib or Palbociclib decreased the viability of MCF-7 D538G and Y537S cells synergistically (calculated as described in the "Methods"), similar to Fulvestrant and CDK4/6 inhibitors (Fig. EV6E–G). These results indicate that activation of

◀ **Figure 5. GR activation is associated with better survival in ER+ patient samples and anti-correlates with *ESR1* expression.**

(A) Kaplan–Meier survival plot showing GR activity signature predictive value in patients with ER+ luminal breast cancer (METABRIC). Patients were stratified based on high and low GR activity signature score; Cox proportional hazard model with log-rank test. GR signature is composed of 52 protein-coding genes, upregulated upon Dex treatment for 8 h and 24 h in both MCF-7 Y537S and D538G models (see "Methods" for signature generation). (B) Graph showing *ESR1* mRNA expression in breast tumors from patients with ER+ disease (METABRIC), stratified according to the GR activity signature score; low, intermediate, high. Wilcoxon matched-pairs signed-rank test. (C) Graph showing H-Score representing the protein abundance of nuclear ER and GR quantified by immunohistochemistry of primary tumors and metastatic tissues from patients with ER+ breast cancer. The H-Score was calculated with the software HALO; *n* = 43 samples. Individual values and lines represent the pairing of the respective samples from patients. Wilcoxon matched-pairs signed-rank test. (D) Scheme of the generation of breast cancer patient-derived organoids (PDO & PDXO). (E) Scatter dot plots representing the nuclear mean intensity of ER quantified from confocal images of MCF-7 D538G cells grown in 3D. *n* = 3 biological replicates. Ordinary one-way ANOVA test. Data are presented as mean ± SD. (F, G) Scatter dot plots representing the nuclear mean intensity of ER quantified from confocal images of ER+ patient-derived xenograft tumor organoids (PDXO; PDXO_1, PDXO_2, PDXO_3) and patient-derived tumor organoids (PDO; PDO_1, PDO_2, PDO_3). *n* = 3 biological replicates, each replicate corresponding to 1 individual patient-derived sample. Technical replicate(s) (*n* = 1–3) have been merged within each individual sample. Ordinary one-way ANOVA test. Data are presented as mean ± SD. (H) Representative maximum intensity projections of confocal image stacks (80 μM z-range, 5 μM z-distance) of ER+ patient-derived xenograft tumor organoids after the indicated treatments. Scale 100 μM. (I) Scatter dot plots showing the mRNA levels of *ESR1, SGK1, TFF1,* and *GREB1* after 24 h of Dex treatment in PDXO_3 sample: *n* = 3 experimental replicates. Two-tailed *t* test. Data are presented as mean ± SD. Source data are available online for this figure.

GR in addition to CDK4/6 inhibitors may also improve the outcome of patients with ER+ metastatic breast cancer. In summary, our data suggest that synthetic glucocorticoids could be repurposed and tested in patients with ER+ metastatic breast cancer, particularly those with the treatment-refractory *ESR1* mutations.

## Discussion

Synthetic glucocorticoids have been used for decades to alleviate side effects of chemotherapy in cancer patients (Lin and Wang, 2016; Cain and Cidlowski, 2017), but their consequences at the cancer cell intrinsic level have been largely overlooked. We have shown previously that GR activation by Dex in TNBC promotes metastatic colonization (Obradović et al, 2019). Here, we provide evidence that, in contrast to TNBC, activated GR in endocrine therapy-resistant ER+ breast cancer acts as an estrogen receptor silencer (ERS) that inhibits liver metastatic outgrowth by suppressing *ESR1* expression. Despite the approval of a new generation of SERDs in the clinics, such as Elacestrant (Pancholi et al, 2022), Camizestrant (Lawson et al, 2023) or Giredestrant (Liang et al, 2021), we propose that Dex, which has a different mode of action in regards to ER loss, could be evaluated as an option for treating patients with ER+ metastatic breast cancer that have become resistant to endocrine therapy, notably via acquisition of missense *ESR1* mutations (Schiavon et al, 2015; Toy et al, 2013; Li et al, 2022).

GR is expressed across all breast cancer subtypes but predicts different outcomes depending on the subtype considered. GR activity is also context dependent (Pan et al, 2011): while high GR expression and activation correlate with poor prognosis in TNBC (Obradović et al, 2019), it is associated with a positive outcome in ER+ breast cancer (West et al, 2016). Moreover, it has been recently reported that loss of GR activity correlates with proliferative capacity in luminal breast tumors (Prekovic et al, 2023). Intriguingly, the selective GR modulators (SGRMs) C134 and C335 were shown to reduce MCF-7 *ESR1* mutant primary tumor growth by decreasing the expression of *CCND1* (Tonsing-Carter et al, 2019). It also appears that GR expression and activation can yield distinct phenotypes on different metastatic sites. Indeed, using intraductal injection of ER+ invasive lobular carcinoma cells (SUM44), it was reported that GR over-expression increased bone metastases by enhancing expression of ECM-associated genes (Porter et al, 2023). Here, we show that GR

activation by Dex decreases liver metastases in MCF-7 *ESR1* mutant cells. It is possible that overexpression or activation of GR may act by unique mechanisms, or that the phenotypic effect of GR on metastasis may be specific to each organ.

Liver is one of the most permissive metastatic organs, and metastases in this organ are highly detrimental for breast cancer patients (Tsilimigras et al, 2021; Massagué and Obenauf, 2016; Nguyen et al, 2009). Using MCF-7 *ESR1* mutant models, we found that metastases in the liver are predominant compared to other organs and result in animal distress, precluding the study of long-term effects on other distant sites. We have shown that Dex treatment decreases established metastases in the liver and prolongs animal survival via a cell-autonomous mechanism. The phenotype contrasts with the pro-metastatic effect of GR in TNBC, wherein the kinase ROR1/WNT5a axis is upregulated in cancer cells after Dex treatment, which in turn promotes lung metastases (Obradović et al, 2019). We found neither *ROR1* nor *WNT5a* upregulation in ER+ models after Dex treatment, highlighting that the output of GR signaling is context and breast cancer subtype-dependent (Noureddine et al, 2021). To what extent the lung or liver metastatic niches exposed to Dex can influence metastatic growth beyond the tumor cell intrinsic mode of action of Dex remains to be investigated in immune-competent background models.

Proteomic and RNA-Seq profiling revealed that estrogen-related gene signatures were strongly downregulated upon GR activation. Unexpectedly, we found a dramatic loss of *ESR1* expression as well as ER protein abundance, which prompted us to decipher how activated GR mediates ER loss. Mechanistically, time-course analysis revealed that ER loss begins around 8 h post Dex treatment, concomitantly with *ESR1* gene silencing. Additionally, we identified early and late repressed ER target genes upon Dex treatment. Early repression may either result from direct GR binding and repression with loss of H3K27Ac or by ER displacement, as shown for *CCND1* (Tonsing-Carter et al, 2019). While late repression of the majority of ER target genes is likely due to the loss of ER chromatin binding some are possibly repressed by GR. We have not assessed H3K27me3 or DNA methylation, therefore we do not know whether Dex-induced *ESR1* silencing is reversible or not. Other reports focusing on short-term treatment with Dex have shown that GR can inhibit ER-dependent transcription either by occupying FOXA1 binding sites (Swinstead

et al, 2016), binding in trans to ER-occupied enhancers following SUMOylation and recruitment of co-repressors (Yang et al, 2017), or by displacing ER from the chromatin (Tonsing-Carter et al, 2019).

These molecular evidence therefore highlight how Dex decreases the metastatic ability of *ESR1* mutant breast cancer cells. This is of paramount clinical importance in the context of endocrine therapy resistance caused by *ESR1* mutations Y537S and D538G, which induce ligand-independent ER activation and ultimately resistance to classical SERDs such as Fulvestrant (Toy et al, 2017). In addition to antagonizing ER, proteomic profiling revealed that GR downregulates other key signaling pathways, such as mTORC1, which may also account for the reduced cell viability.

It is remarkable in our study that Dex is more potent than Fulvestrant at reducing liver metastases in MCF-7 *ESR1* mutant models. This suggests that Dex may benefit patients with metastatic disease who have progressed on Fulvestrant treatment. Moreover, we show a synergistic effect of CDK4/6 inhibitors such as Ribociclib and Palbociclib in combination with Dex in MCF-7 D538G and Y537S cells, in vitro. This is consistent with residual, albeit weak, ER signaling and protein abundance following Dex treatment. Furthermore, we found that in samples from patients with ER+ metastatic disease, the abundance of ER negatively correlates with the abundance of GR, emphasizing that activation of GR signaling in these patients would confer a growth disadvantage on cancer cells, thereby preventing distant metastasis in ER+ disease.

The presence of additional members of the nuclear receptor family, such as PR and AR and their crosstalk with ER, has been shown to be of major importance for clinical outcome and patient stratification (Arpino et al, 2005; Mohammed et al, 2015; He et al, 2019). Indeed, progesterone was shown to inhibit E2-mediated cell proliferation and tumor growth in ER+ breast cancer by reprogramming chromatin-binding of ER (Mohammed et al, 2015). Like PR, the activation of AR in ER+ breast cancer exerts anti-tumorigenic activity by reprogramming the ER cistrome and enhancing responsiveness to endocrine therapy (Hickey et al, 2021). Similarly, here we found that GR activation abrogates ER genome-wide recruitment, hence inhibiting the transcription of its target genes. It is therefore intriguing to observe that parallel activation of other members of the nuclear receptor family is deleterious for ER expression and/or genomic activity, as well as for hindering ER+ disease progression. This argues for patient stratification that systemically includes the AR and GR markers, in addition to ER and PR, as their activation may prove beneficial in the clinics for improving the treatment of patients with ER+ breast cancer metastasis. Additionally, it would be interesting to determine the effect of Dex in combination with other targeted therapies such as PI3K/p110 inhibitors for treating advanced ER+ metastatic disease. As novel estrogen receptor modulators and degraders are needed to help patients with ER+ metastatic breast cancer, our findings suggest that activated GR could act as an ERS in the treatment of ER+ breast cancer metastasis. Finally, screening for such ERS activity might uncover additional synthetic glucocorticoids that are even more potent than Dex at silencing *ESR1* expression and reducing liver metastases. Endocrine therapy-resistant metastasis remains a major clinical hurdle, but our results demonstrate that glucocorticoids could be repurposed as a promising therapeutic strategy for treating metastasis in patients with ER+ breast cancer. Evaluating this possibility in clinical settings is warranted.

# Methods

### Reagents and tools table

| Reagent/resource | Reference or source | Identifier or catalog number |
|---|---|---|
| **Experimental models** | | |
| NSG | In-house colony | |
| MCF-7 | ATCC | ATCC HTB - 22 |
| T-47D | ATCC | ATCC HTB - 133 |
| MCF-7 D538G | Prof. Ben Ho Park Lab | |
| MCF-7 Y537S | Prof. Ben Ho Park Lab | |
| Human hepatocytes immortalized by SV40 large T-antigen | ATCC | PTA-5565 |
| **Recombinant DNA/plasmid** | | |
| pFU-Luc2-eGFP (L2G) | | |
| shGR1 | Dharmacon | V3LHS_404051 |
| shGR2 | Dharmacon | V3LHS_404052 |
| pGIPZ empty vector | Dharmacon | |
| **Antibodies** | | |
| Anti-ERa | Thermo scientific | MA5-14501 |
| Anti-ERa SP1 | Thermo Scientific | MA1-39540 |
| Anti-GR | GeneTex | GTX101120 |
| Anti-ERK2 | Santa Cruz | sc-1647 |
| Anti-Actin | Sigma-Aldrich | A5441 |
| Anti-S6 | Cell Signaling | 2317 |
| Anti-mouse secondary antibody | Merck | GENA931 |
| Anti-rabbit secondary antibody | Merck | GENA934 |
| Alexa Fluor 647 Annexin V | Biolegend | 640912 |
| MHC-H2-PE | Biolegend | 125506 |
| Ki67 SP5 | Thermo scientific | MA5-14520 |
| Goat anti-rabbit Alexa 647 secondary antibody | Thermo scientific | A21245 |
| Peroxidase polymer Goat Anti-Mouse | Nicherei | 414131 |
| Goat Anti-Rabbit | Nicherei | 414142 |
| Anti-GR | Cell signaling technology | 47411 |
| Rabbit IgG antibody, H-270 | Thermo scientific | |
| **Oligonucleotides and other sequence-based reagents** | | |
| *GAPDH* | IDT | *Hs.PT.39a.22214836* |
| *HPRT1* | IDT | *Hs.PT.58 v.45621572* |
| *SGK1* | IDT | *Hs.PT.58. 39808948.g* |
| *ESR1* | IDT | *Hs. PT. 58. 14846478* |
| *FOXA1* | IDT | *Hs.PT.58.1788586* |
| *CCND1* | IDT | *Hs00765593_m1* |
| *GREB1* | IDT | *Hs.PT.58.26216464* |

| Reagent/resource | Reference or source | Identifier or catalog number |
|---|---|---|
| *CXCL12* | IDT | *Hs.PT.58.27881121* |
| *TFF1* | IDT | *Hs.PT.58.168461* |
| *TFF3* | IDT | *Hs.PT.58.1814807* |
| *RARA* | IDT | *Hs.PT.58.2437218* |
| *PGR* | IDT | *HS.PT.58.50458902* |
| *AREG* | IDT | *Hs.PT.56a.38817860* |
| *SLC2A1* | IDT | *Hs.PT.58.25872862* |
| *SLC7A5* | IDT | *Hs.PT.58.25573676* |
| *ITPK1* | IDT | *Hs.PT.58.22272228* |
| *CALB2* | IDT | *Hs.PT.56a.20381264* |
| *GAB2* | IDT | *Hs.PT.58.40895708* |
| *ROR1* | IDT | *Hs.PT.58.39481678* |
| *GREB1* | IDT | *Hs.PT.58.26216464* |
| *GATA3* | IDT | *Hs.PT.58.4308511* |
| **Chemicals, enzymes, and other reagents** | | |
| DMEM | Sigma | D6429-500ML |
| Fetal Calf Serum (FCS) | Sigma | F7524-500ML |
| Penicillin streptomycin | Sigma | P4333 |
| Normocin | Invivogen/Labforce | ant-nr-1 |
| Glutamax | Thermo Scientific | 35050-038 |
| Insulin | Sigma | 91077 C |
| RPMI | R8758-500ML | R8758-500ML |
| William's E Medium | Thermo Scientific | 12551032 |
| Phenol red-free DMEM | ThermoFisher Scientific | A3382101 |
| Activated charcoal | Sigma | C7606-125G |
| Dex | Sigma | D2915 |
| Hexadimethrine bromide | Sigma | H9268 |
| Puromycin | InvivoGen | ant-pr-1 |
| Luciferin | Biosynth | L8220 |
| Fulvestrant | Sigma-Aldrich | I4409 |
| 17β-estradiol | Sigma-Aldrich | E2758-5G |
| Camizestrant | Medchem express | HY-136255 |
| Elacestrant | Selleckchem | S9629 |
| Paclitaxel | Medchem express | HY-B0015 |
| Paclitaxel | Selleckchem | S1150 |
| Molecular-grade $H_2O$ | Sigma-Aldrich | W4502 |
| DMSO | Sigma-Aldrich | D2650 |
| Fulvestrant/Faslodex | AstraZeneca | 9112474 |
| Hoechst 33342 | Invitrogen | H3570 |
| Sulforhodamine B | Sigma | 230162 |
| Trichloroacetic acid | Sigma | T6399 |
| CellTiter-Glo 2.0 reagent | Promega | G9242 |
| Click-iT EdU Alexa Fluor 647 assay kit | Invitrogen | C10419 |

| Reagent/resource | Reference or source | Identifier or catalog number |
|---|---|---|
| Cell staining buffer | Biolegend | 420201 |
| Annexin V binding buffer | Biolegend | 422201 |
| Red Blood Cell (RBC) lysis buffer | Sigma | R7757-100ML |
| Collagenase A | Roche | 11088793001 |
| Dispase | StemCellTechnologies | 7913 |
| DNAse I | Sigma | D5025-15KU |
| HEPES | Sigma | 83264-100ML-F |
| Leibovitz's L-15 medium | Gibco, Thermo Scientific | 11415-056 |
| Corning® Matrigel® Basement Membrane Matrix, LDEV-free, 10 mL | Sigma | 356231 |
| Advanced DMEM/F12 | ThermoFisher | ThermoFisher |
| DAPI | ThermoFisher | D1306 |
| RNeasy Plus Mini kit | Qiagen | 74136 |
| iScript cDNA synthesis kit | BioRad | 170-8891 |
| Prime time gene expression master mix | IDT | 1055771 |
| 1X formal fix | Histocom | 9990244 |
| Hematoxylin II and bluing reagent | Ventana, Roche Diagnostics | |
| Blocking buffer - 1X Casein | Surmodics | PBSC-0100-01 |
| Poly L-lysine | Sigma | P2636 |
| 1× protease inhibitor cocktail | Complete Mini, Roche | 11836153001 |
| BCA assay Kit | Thermo Scientific | 23227 |
| PVDF membrane | Immobilon-P, Millipore | IPVH85R |
| WesternBright peroxide and substrates | Advansta | R-03025-C50; R-03027-C50 |
| Western Lightning plus ECL oxidizing and enhanced luminol reagents | Perkin Elmer | 0RT2655; 0RT2755 |
| Sequencing-grade modified trypsin | Promega | PRV5111 |
| Formaldehyde | Sigma | F8775 |
| Glycine | Sigma | G8898-500G |
| Dynabead Protein G beads | ThermoFisher | 10003D |
| QIAquick PCR purification kit | Qiagen | 28104 |
| PowerUP SYBR Green Master Mix | ThermoFisher | A25742 |
| **Software** | | |
| FlowJo (v.5) BD | https://www.flowjo.com/ | |
| CQ1 software | Yokogawa | |
| HALO (v3.1) | | |
| SpectroMine software, version 1.0.20235.13.16424 | Biognosys | |
| SafeQuant R script (v2.3) | PCF, University of Basel | |

| Reagent/resource | Reference or source | Identifier or catalog number |
|---|---|---|
| Ingenuity Pathway Analysis (IPA) | Qiagen | |
| GraphPad PRISM v10 | https://www.graphpad.com/ | |
| ImageJ | https://imagej.net/ij/ | |
| **Other** | | |
| BD ARIA cell sorter | BD Biosciences | |
| IVIS Lumina XR | Caliper LifeSciences | |
| Newton Vilber | https://www.vilber.com/ | |
| Gen5 plate reader | Agilent | |
| Cytoflex analyzer | BD Biosciences | |
| Synergy H1 microplate reader | BioTek | |
| MELODY Sorter | BD Biosciences | |
| Confocal spinning disk microscope | Yokogawa | |
| Microm HM 340E | ThermoFisher Scientific | |
| Ventana DiscoveryXT instrument | Roche | |
| NanoZoomer S60 digital slide scanner | Hamamatsu | |
| Bioruptor Pico device | Diagenode | |
| Freedom Evo 100 liquid-handling platform | Tecan Group Ltd., Männedorf, Switzerland | |

## Cell lines and patient-derived xenograft (PDX) models

The parental MCF-7 and T-47D cell lines were purchased from ATCC, and MCF-7 D538G and Y537S cell lines were kindly provided by Prof. Ben Ho Park (Scott et al, 2017). Briefly, activating *ESR1* mutations D538G and Y537S were produced by a gene targeting approach using recombinant AAV vectors harboring Y537S or D538G mutations, and the *ESR1* knock-in cell lines express mutations constitutively. PCR screening was performed to isolate the neomycin-resistant clones. To remove the neomycin cassette, cells were exposed to Cre-expressing recombinant adenovirus, followed by Sanger Sequencing to confirm the mutant clones (Scott et al, 2017). We sequenced *ESR1* and confirmed its mutations in MCF-7 D538G and Y537S cell lines. All cell lines were routinely tested for mycoplasma contamination. Parental MCF-7, MCF-7 D538G, and Y537S cell lines were propagated in monolayer cultures in DMEM supplemented with 10% fetal calf serum (FCS), penicillin, streptomycin (Sigma, P4333), normocin (Invivogen/Labforce, ant-nr-1), glutamax (Thermo Scientific, 35050-038), and insulin (Sigma, 91077C). Parental T-47D cells were grown in RPMI media supplemented with 10% FCS, penicillin, streptomycin, and normocin. Human hepatocytes immortalized by SV40 large T-antigen were obtained commercially (ATCC PTA-5565) and cultured in William's E Medium (Thermo Scientific, 12551032) with glutamax

supplemented with 5% FCS. The hepatocytes were labeled with an mCherry construct by lentiviral infection. GR activation experiments were performed in monolayer cultures in phenol red-free DMEM (ThermoFisher Scientific; cat. no. A3382101) supplemented with 2.5% charcoal-stripped FCS (activated charcoal Sigma C7606-125G) in the presence of water-soluble Dex (700 nM, Sigma, D2915) or vehicle for 7 consecutive days. The authenticity of the cell lines was validated by STR profiling. The ER + PDX model (PDXO_1, PDXO_2) was kindly donated by Michael T. Lewis, Baylor College of Medicine, transplanted into the mammary fat pads of NSG mice with E2 supplementation, and expanded in our animal facility. The PDX model (PDXO_3) was established in the lab with a patient sample obtained from University Hospital Basel with the project ID: 2018-00729 that was approved by the Swiss authorities (Ethics Committee of Northwestern and Central Switzerland, EKNZ). Patients (only women with breast cancer, regardless of their age, ethnicity, or gender) were consented for donating their tumor sample for research to the lab. The experiments reported in this manuscript conformed to the principles set out in the WMA Declaration of Helsinki and the Department of Health and Human Services Belmont Report.

## Lentiviral infection

Lentiviruses were produced using PEI transfection of HEK293T cells. The titer of each lentiviral batch was determined in MCF-7 D538G and Y537S. For lentiviral infections, cells were spin-infected ($1200 \times g$ for 45 min at 32 °C) in the presence of 8 μg/mL hexadimethrine bromide (Sigma, cat no. H9268) and incubated overnight. Selection with 2 μg/ml puromycin (InvivoGen, cat no. ant-pr-1) was applied 48 h after infection for up to 2 weeks. The cells were labeled with a GFP luciferase construct (pFU-Luc2-eGFP (L2G)) and sorted using a BD ARIA cell sorter (BD Biosciences). The expression of the GFP-luciferase construct was analyzed using a Cytoflex analyzer prior to the in vivo experiments. GR downregulation was performed using shRNA constructs (shGR1 and shGR2) (V3LHS_404051, V3LHS_404052, Dharmacon). Non-targeting shRNA (pGIPZ empty vector) was used as the control.

## Antibodies and compounds

The antibodies used for immunoblot analyses were the following: anti-ERα (Thermo Scientific; MA5-14501), anti-GR (GeneTex; cat. no. GTX101120), anti-ERK2 (Santa Cruz, cat. no. sc-1647), anti-Actin (Sigma-Aldrich, A5441), anti-S6 (Cell Signaling, cat no. 2317), secondary antibodies anti-mouse (Merck, cat. no. GENA931), and anti-rabbit (Merck, cat. no. GENA934). Compounds used for cell treatments were Dex (water-soluble; D2915, Sigma), luciferin (Biosynth; L8220), Fulvestrant (ICI; I4409, Sigma-Aldrich), Camizestrant (Medchem express, HY-136255), Elacestrant (S9629, Selleckchem) 17β-estradiol (E2758-5G, Sigma-Aldrich), and Paclitaxel (Medchem express; HY-B0015 and Selleckchem; S1150). Dex was dissolved in sterile molecular-grade $H_2O$ (Sigma-Aldrich, cat no. W4502), which was therefore used as a vehicle in vitro. Paclitaxel was dissolved in DMSO (Sigma-Aldrich, cat. no. D2650), and Fulvestrant and 17β-estradiol were dissolved in molecular-grade ethanol.

## Animal experiments

All in vivo experiments were performed in compliance with the Swiss Animal Welfare Ordinance and approved by the Cantonal Veterinary Office of Basel-Stadt (Permit ID: National N°35056, Cantonal N°2256). Female NSG (Nod-SCID-Gamma) mice were maintained in the animal facilities of the Department of Biomedicine in accordance with Swiss guidelines on animal experimentation. Mice were maintained in a sterile environment with light, humidity, and temperature control (light–dark cycle with light from 7:00 to 17:00, with a gradual change from light to dark, temperature 21–25 °C, and humidity 45–65%). Mice were grouped and allowed to acclimatize before the experiments. Experimental metastasis assays were performed by injecting MCF-7 D538G and Y537S cells ($0.5 \times 10^6$ cells suspended in 100 µl of media) into the tail vein of 6- to 12-week-old immunocompromised mice. After intravenous injection of cells, the mice were imaged using in vivo bioluminescence imaging to ensure proper injection and to monitor metastatic outgrowth. Bioluminescence imaging was performed using an IVIS Lumina XR (Caliper LifeSciences) and a Newton (Vilber) after intraperitoneal injection of luciferin. In vivo Dex treatment was performed with a clinically relevant dose of 0.1 mg/kg for 2 weeks for 5 consecutive days followed by 2 days of rest from the treatment as previously described (Pang et al, 2006). Paclitaxel was administered intraperitoneally once a week at 25 mg/kg for 2 weeks, and for combination treatments, Dex was administered for 5 consecutive days after each Paclitaxel injection. A combination of Dex and Fulvestrant was administered for 6 weeks: Fulvestrant at 80 mg/kg (Faslodex, 50 mg/ml, AstraZeneca) or its vehicle (Castor oil) was administered subcutaneously twice a week, and 0.1 mg/kg of Dex or its vehicle (PBS) was administered for 5 consecutive days followed by 2 days of rest. All the drugs were administered by a single administrator. For overall survival experiments, independent assessments of fitness and distress signs (e.g., weight loss, immobility, unusual appearance and posture, abnormal respiration) of the animals were performed by a single investigator. No blinding was done. Animals were randomized prior to each experiment based on age and weight. The distress signs were assessed continuously, and the animals were sacrificed and autopsied upon reaching the endpoint. Harvesting of metastatic organs upon sacrifice was performed by single or multiple investigators. Anesthesia for the mice and sacrifice procedures were performed according to the protocols approved by the Cantonal Veterinary Office of Basel-Stadt.

## Cell viability assay

Cells were plated in 96-well black bottom cell-culture plates in complete media. After 24 h, the media was removed and exchanged with phenol red-free DMEM supplemented with 2.5% charcoal-stripped FCS. The cells were treated with 700 nM Dex for 4 days with a refresh on day 3. For experiments with CDK4/6 inhibitors, cells were treated with respective drugs and doses in complete media for 6 days. Dex was refreshed on day 3 of treatment. The CellTiter-Glo 2.0 reagent (Promega, cat no. G9242) was thawed at 22 °C in a water bath to equilibrate the reagent to room temperature. At the end of the treatments, the media was removed and 75 µl of CellTiter-Glo 2.0 reagent was added to each well. The plates were mixed for 2 min on an orbital shaker to induce cell lysis and incubated at room temperature for 10 min to stabilize the luminescent signal. The luminescence was recorded using a Gen5 plate reader. The recorded luminescence from the ATP levels were correlated to the viable cell numbers. Data were plotted as relative cell viability by normalizing the cell numbers to the control. Values measured from the viability assay were analyzed for synergism using the formula $AB/C < A/C \times B/C$, where C is the value of vehicle, A is the value of drug 1, B is the value of drug 2, and AB is the drug combination value (Amante et al, 2023).

## Cell cycle analysis

Cells were treated with 700 nM Dex or vehicle in 2.5% charcoal-stripped FCS containing Phenol red-free DMEM culture medium for 96 h. At the endpoint, the cells were labeled with 10 µM EdU for 2 h. Detection of EdU was conducted using the Click-iT EdU Alexa Fluor 647 assay kit (Invitrogen, cat no. C10419) according to the manufacturer's guidelines. DNA was stained with Hoechst 33342 (Invitrogen, cat no. H3570), and cells were quantified on a Cytoflex analyzer. Results were analyzed with the FlowJo software (v.5).

## Sulforhodamine B (SRB) assay

MCF-7 D538G and Y537S cells were plated in 24-well plates (10,000 cells per well). The medium was changed the next day to 2.5% charcoal-stripped FCS containing phenol red-free DMEM. The cells were then treated with 700 nM Dex or vehicle on day 1, and the treatments were refreshed on day 3 and day 5. After 96 h or 120 h of treatment, the medium was removed and cells were fixed and stained with sulforhodamine B (Sigma, cat no. 230162). Briefly, cells were fixed with 200 µl of cold 3.3% trichloroacetic acid (Sigma, cat no. T6399) per well at 4 °C for 1 h. Plates were washed 4 times with water and air-dried at room temperature overnight. Subsequently, 80 µl of 0.057% sulforhodamine B solution was added to each well, and plates left at room temperature for 30 min. The plates were then washed with 1% acetic acid (v/v) and air-dried at room temperature. Aliquots of 200 µl of 10 mM Tris-Base solution (pH 10.5) were then added to each well and the plates were incubated on a gyratory shaker for 5 min to solubilize the protein-bound dye. The optical density (OD) was measured at 510 nm using a Synergy H1 microplate reader (BioTek). In addition, one plate was fixed on day 0, stained, and measured as described above in order to normalize the cell count.

## Annexin V staining

MCF-7 D538G and Y537S cells were plated in 6-well plates (100,000 cells per well). The cells were starved the next day by changing to 2.5% charcoal-stripped FCS containing phenol red-free DMEM. The cells were treated with 700 nM Dex or vehicle with a treatment refresh on day 3. After 72 h of treatment, and 3 h prior to cell harvest, 500 µM of $H_2O_2$ was added to a well to serve as a positive control. At endpoint, both floating and adherent cells were collected using trypsin–EDTA in FACS tubes. The cells were washed twice with cold Biolegend cell staining buffer and resuspended in Annexin V binding buffer at a concentration of $1 \times 10^6$ cells/ml. 5 µl of Alexa Fluor 647 Annexin V was added to the cell suspension and incubated for 15 min at RT. The cells were then stained with DAPI (1 µg/ml) in 100 µl of Annexin V binding buffer.

The stained samples were quantified by flow cytometry, and the results were analyzed with the FlowJo software (v.5).

## Isolation of epithelial cells from patient-derived xenograft tumors for organoid culture

The primary tumors from patient-derived xenografts (PDX) were resected and chopped in the tissue chopper three times. The chopped tissue was incubated with 5 ml of digestion media containing DMEM F12, 2% P/S, 0.3% collagenase A (Roche), 2.5 U/ml Dispase, 20 mM HEPES at 37 °C for 60 min under shaking. Aliquots of 5 ml of media were added to the digestion mix to inactivate tissue digestion and centrifuged at 250×g for 5 min. The cells were resuspended in 5 ml of Red Blood Cell (RBC) lysis buffer (Sigma R7757-100ML) and incubated at room temperature for 5 min. Aliquots of 10 ml of PBS were added to stop the red blood cell lysis. Cells were centrifuged, resuspended in 5 ml of trypsin, and incubated at 37 °C for 5 min. Aliquots of 10 ml of Leibovitz's L-15 medium (Gibco, Thermo Scientific, cat no. 11415-056) were added to inactivate the trypsinization. This was followed by centrifugation. The cells were then resuspended in 1 ml of Dispase-DNAse (Dispase, StemCellTechnologies, cat no. 7913, DNAse I, Sigma, cat no. D5025-15KU) and incubated at 37 °C for 10 min. Aliquots of 10 ml of medium were added to the Dispase-DNAse mix to stop the enzymatic digestion. Cells were centrifuged, resuspended in FACS buffer, and sterile filtered using 40-µm cell strainers. The single cells were then counted and subsequently stained with antibodies against CD298 (4 µl of CD298 PE/$10^6$ cells) and MHC-H2-PE (Biolegend 125506; 1:500) for mouse cell detection and incubated at 4 °C for 30 min. The antibodies were washed off and stained with DAPI (1:100,000). The cells were then sorted using MELODY.

## Drug testing on patient-derived organoids

Patient-derived tumor organoids (PDO) were established from breast tumors, and patient-derived xenograft tumor organoids (PDXO) were established from primary tumors of PDX as previously described (Guillen et al, 2022). The tissues for the PDOs were obtained from patients of the University Hospital Basel under the project ID: 2018-00729 that was approved by the Swiss authorities (Ethics Committee of Northwestern and Central Switzerland, EKNZ). Patient consent was obtained for all human material established in the lab and experiments were performed according to the principles of the Declaration of Helsinki. PDOs (n = 3 patients) and PDXOs (n = 3 patients) were cultured in 3D using Matrigel (Corning, cat no. 356231) as previously described (Sachs et al, 2018). Briefly, 1600 cells per well were plated in a 384-well plate with 15 µl of matrix (40% of Matrigel, 60% of advanced DMEM), on top of a first layer of 15 µl of the same matrix. Aliquots of 70 µl per well of Clevers medium were added on top. PDOs and PDXOs were cultured for 7–10 days. About 40 µl of the media was exchanged with vehicle- or drug-containing media. Media was refreshed with the drugs every 3 days of treatment and, at the end of the experiment, 50 µl of media was removed, and organoids were washed three times with 50 µl PBS/1% BSA. Organoids were then fixed with 50 µl 8% PFA in PBS for 30 min at RT. PFA was removed and cells were washed three times with 50 µl PBS/1% BSA, and then permeabilized by adding 50 µl of 0.6% Triton-X 100 in PBS.

Organoids were then washed again three times with 50 µl PBS/1% BSA. DAPI (final 5 µg/ml, ThermoFisher, cat no. D1306) in PBS with 1% BSA was added to stain the nuclei. Antibodies against ERα (SP1 Thermo Scientific, MA1-39540, 1:250), GR (Genetex, GTX101120, 1:1000), Ki67 (SP5- Thermo Scientific - MA5-14520, 1:1000), and the secondary goat anti-rabbit Alexa 647 antibodies (#A21245 1:500) diluted in PBS with 1% BSA were added for staining.

## Image analysis of organoids

Images of organoids were acquired as a z-stack (×20, 20 field of views per well, three wells per condition, 70 µm z-range, 5 µm z-distance) in DAPI and Alexa 647 and with brightfield imaging using a CQ1 (Yokogawa) confocal spinning disk microscope and Yokogawa acquisition software. Organoids and nuclei were segmented based on the DAPI channel and intensity features determined with the integrated CQ1 software with the following settings:

1. Organoid level: Gaussian filter size: 30, AdaptiveThreshold, ObjectSize 200 µm, ObjectDetectivity 0.5, ErosionCircle diameter 2 µm, mask was expanded in 3D to threshold 900, holes were filled, mask was dilated by 2 µm, SizeFilter excluded objects < 20 µm$^3$ and > 6,000,000 µm$^3$ (objects with at least one cell are included). RoundnessFilter excludes objects with less than 0.04 circularity to remove debris fibers.
2. Nucleus level: Gaussian filter size: 10, Mean Image mask size 2 µm, Threshold 1500, ErosionCircle diameter 1 µm, FindMaximum in Gaussian after thresholding, remove size4, mask was expanded on Mean Image in 3D to threshold 1100, SizeFilter excluded objects <20 µm$^3$ and >30,000 µm$^3$ (objects with at least one cell are included), holes were filled.
3. Cell level: The nucleus mask was expanded in 3D to threshold 900, mask was dilated by 1.3 µm.

## Co-culture assay with hepatocytes

Immortalized hepatocytes were seeded as a layer on a 24-well plate at a density of $4 \times 10^4$ cells per well and grown close to confluence for 2 days. GFP luciferase-labeled MCF-7 D538G cells were seeded on top of the confluent hepatocytes layer at a seeding density of 1000 cells per well in order to recapitulate disseminated tumor cell seeding in the liver without inducing a burden on the hepatic tissue (Correia et al, 2021). The cells were treated with Dex for 1 h after cancer cell seeding, and the medium was refreshed every other day for 4 days. The co-cultures were collected on day 6, stained with a dead-cell indicator (DAPI, 2 µg/ml), and quantified on a Cytoflex analyzer. The data were then analyzed using the FlowJo software.

## qPCR

Cells were cultured and serum-starved using 2.5% charcoal-stripped FCS containing phenol red-free DMEM. The cells were then treated with Dex or vehicle for the indicated times. RNA isolation from the cells and organoids was performed using the protocol specified in the Qiagen RNeasy Plus Mini kit (Qiagen, cat no.74136). RNAs were quantified using nanodrop, and cDNA was

generated using 1 µg of mRNA (BioRad, iScript cDNA synthesis kit; cat no. 170-8891). For quantitative real-time PCR (qPCR) analysis, the Prime-time gene expression master mix (IDT, cat no. 1055771) and the predesigned probes were obtained from IDT. *GAPDH* (*Hs.PT.39a.22214836*) and *HPRT1* (*Hs.PT.58 v.45621572*) were used as housekeeping genes. The human IDT probes used were: *NR3C1: Hs.PT.58.27480377, SGK1: Hs.PT.58. 39808948.g, ESR1: Hs. PT. 58. 14846478, FOXA1: Hs.PT.58.1788586, CCND1: Hs00765593_m1, GREB1: Hs.PT.58.26216464, CXCL12: Hs.PT.58.27881121, TFF1: Hs.PT.58.168461, TFF3: Hs.PT.58.1814807, RARA: Hs.PT.58. 2437218, PGR: HS.PT.58.50458902, AREG: Hs.PT.56a.38817860, SLC2A1: Hs.PT.58.25872862, SLC7A5: Hs.PT.58.25573676, ITPK1: Hs.PT.58.22272228, CALB2: Hs.PT.56a.20381264, GAB2: Hs.PT.58. 40895708, ROR1: Hs.PT.58.39481678. GREB1: Hs.PT.58.26216464, and GATA3: Hs.PT.58.4308511*. qPCR data are presented as – Delta Ct (Ct gene – Ct housekeeping gene).

## Immunohistochemistry and quantification using HALO analysis

The patient samples used for the analysis were obtained from University Hospital Basel with the project ID: 2023-01270 that was approved by the Swiss authorities (Ethics Committee of Northwestern and Central Switzerland, EKNZ). All the tissues were fixed in 1× formal fix (Histocom, 9990244): PBS solution for 48 h at room temperature. Samples were then dehydrated, embedded in paraffin, and sectioned (3–4 µm) with a Microm HM 340E (ThermoFisher Scientific). All immunohistochemistry experiments were performed using a Ventana DiscoveryXT instrument (Roche Diagnostics) following the Research IHC DAB Map XT procedure. Counterstaining was performed with hematoxylin II and bluing reagent (Ventana, Roche diagnostics). Briefly, for all the staining, including patient samples, slides were pre-treated with cell-conditioning medium 1 (CC1, Roche Diagnostics) for 40–72 min at 95 °C, followed by 8–40 min of incubation with blocking buffer (1X Casein, Surmodics, cat no. PBSC-0100-01). Primary antibodies were diluted in blocking buffer and incubated for 1 h at 37 °C, followed by incubation of secondary antibodies (Peroxidase polymer Goat Anti-Mouse, Nicherei, 414131, and Goat Anti-Rabbit, Nicherei, 414142) for 1 h at 37 °C, and developed with the Discovery ChromoMap DAB detection kit. Primary antibodies used were anti-GR (Cell Signaling Technology – 47411, dilution 1/100), anti-ERα (SP1 – Thermo Scientific – MA5-14501, dilution 1/100), and anti-Ki-67 (SP5 - Thermo Scientific - MA5-14520, dilution 1/50). For the anti-GR antibody, the signal was amplified using the Ventana amplification kit (#760-080). Whole sections were scanned using a NanoZoomer S60 digital slide scanner (Hamamatsu). Nuclear ER and GR abundance in patient samples was analyzed and quantified using HALO (v3.1). The ratio of positive cells was determined using the multiplex ICH v2.3.4 module. Haematoxylin & eosin (H&E) staining of tissue sections was performed using a standard procedure.

## Immunofluorescence

To achieve better adherence and to prevent crowding of the cells at the periphery of the wells, eight-well chamber slides were pre-coated with poly L- lysine (Sigma, cat no. P2636) prior to seeding. Briefly, 5 mg of poly L-lysine (lyophilized powder) was dissolved in 50 ml of sterile tissue culture grade water; 300 µl of poly L-lysine was used per well for

an eight-well chamber slide. The plate was incubated for 5 min, the lysine discarded, and the wells washed with PBS and air-dried for 2 h. In total, 10,000 cells suspended in 400 µl of 10% FCS DMEM were seeded onto the wells. After 24 h, the media was changed to phenol red-free DMEM supplemented with 2.5% charcoal-stripped FCS. Once the cells attained 80% confluence, they were treated with vehicle or Dex (700 nM) for 1 h to activate GR signaling. The medium was discarded and the cells were rinsed once with ice-cold PBS, before 300 µl of 4% PFA was added to the wells and incubated for 12 min. The cells were then washed twice with ice-cold PBS for 5 min each and then 300 µl of 0.3% Triton PBS was added and incubated for 15 min before washing. The cells were incubated with PBS 2% BSA for 1 h to block unspecific sites and were subsequently incubated with 1:500 dilution of anti-GR antibody in 2% BSA PBS at 4 °C overnight. The cells were again washed thrice with PBS for 5 min each and incubated with the secondary antibody conjugated to far-red fluorophore at 1:1000 dilution for 2 h at room temperature on a rocking table in dark conditions. The cells were washed twice with PBS and incubated with DAPI/PBS for 10 min at room temperature before washing again with PBS and imaging under a Nikon Ti2 microscope.

## Immunoblotting

Cells were cultured and serum starved using 2.5% charcoal-stripped FCS containing phenol red-free DMEM for 48 h and then treated with various compounds for the indicated treatment times. The cells were then lysed in RIPA buffer (50 mM Tris-HCl pH 8, 150 mM NaCl, 1% NP-40, 0.5% sodium deoxycholate, 0.1% SDS supplemented with 1x protease inhibitor cocktail (Complete Mini, Roche), 0.2 mM sodium orthovanadate, 20 mM sodium fluoride, and 1 mM phenylmethylsulfonyl fluoride. The cells were sonicated (15 cycles with 30 s ON, 30 s OFF) using a Bioruptor Pico device (Diagenode) and the lysates were collected by centrifugation. Protein concentrations were measured using the BCA assay Kit (Thermo Scientific; cat no. 23227). Laemmli buffer (5x) supplemented with 5% 2-mercaptoethanol was added to an equivalent volume of protein lysate. The lysates (40 µg per lane) were resolved by 10% SDS-PAGE and then transferred to 0.45 µm hydrophobic PVDF membranes (Immobilon-P, Millipore, cat no. IPVH85R). After the proteins were completely transferred, they were blocked with 5% milk diluted in 0.1% Tween20 in TBS (TBST) for 1 h at room temperature and the membranes were incubated overnight with primary antibodies against human GR, ER, ERK2, and S6 at 4 °C (GR 1:500, ER 1:1000, ERK2 1:1000, Actin 1:5000, S6 1:1000). The primary antibodies were washed off three times in TBST and the membranes incubated with a 1:5000 dilution of secondary HRP-coupled anti-mouse or anti-rabbit antibodies for 2 h at room temperature. Membranes were washed three times and developed with a 1:1 ratio of WesternBright peroxide and substrates (Advansta, cat no. R-03025-C50; R-03027-C50) or a 1:1 ratio of Western Lightning plus ECL oxidizing and enhanced luminol reagents (Perkin Elmer, cat no. 0RT2655; 0RT2755). The blots were developed using the Vilber chemiluminescent detection system.

## Sample preparation for LC–MS analysis

MCF-7 D538G and Y537S cells were propagated for 7 days in 2.5% charcoal-stripped FCS containing phenol red-free DMEM in the presence of Dex (700 nM) or vehicle. The cells were then mechanically detached, washed, and snap-frozen until analysis. In

all, 500,000 cells (4 control replicates, 4 Dex-treated replicates, for both MCF-7 Y537S and D538G models, i.e., a total of 16 samples) were collected, washed twice in PBS, and lysed in 50 µl lysis buffer (2% sodium deoxycholate SDC, 0.1 M TRIS, 10 mM TCEP pH 8.5) using strong ultrasonication (10 cycles, Bioruptor, Diagnode). Protein concentrations were determined by tryptophan fluorescence analysis (Wiśniewski and Gaugaz, 2015) using a small sample aliquot. Sample aliquots of 50 µg total protein were reduced for 10 min at 95 °C and alkylated with 15 mM chloroacetamide for 30 min at 37 °C. After dilution of the samples 1:1 (v:v) with 0.1 M TRIS pH 8.5, the proteins were digested by incubation with sequencing-grade modified trypsin (1/50, w/w; Promega, Madison, Wisconsin) overnight at 37 °C. The peptides were then cleaned up using iST cartridges (PreOmics, Munich) according to the manufacturer's instructions. Samples were dried under vacuum and stored at −80 °C until further use. 5 µg aliquots of peptides were labeled with isobaric tandem mass tags (TMTpro 16-plex, ThermoFisher Scientific) as described previously (Ahrné et al, 2016) using a Freedom Evo 100 liquid-handling platform (Tecan Group Ltd., Männedorf, Switzerland). In short, peptides were resuspended in 12.5 µl labeling buffer (2 M urea, 0.2 M HEPES pH 8.3) and 2.5 µl of each TMT reagent was added to the individual peptide samples followed by a 1 h incubation at 25 °C and shaking at 500 rpm. The labeling reaction was quenched by adding 2 µl aqueous 0.75 M hydroxylamine solution and incubating for 5 min at 25 °C. After pooling, the pH of the sample was increased to 11.9 with 1 M phosphate buffer (pH 12) and incubated for 20 min at 25 °C to remove TMT labels linked to peptide hydroxyl groups. Subsequently, the reaction was stopped by adding 2 M hydrochloric acid until pH <2 was reached. Finally, the peptide samples were further acidified using 5% TFA, desalted using Sep-Pak Vac 1cc (50 mg) C18 cartridges (Waters) according to the manufacturer's instructions, and dried under vacuum. TMT-labeled peptides were fractionated by high-pH reversed phase separation using a Xbridge Peptide BEH C18 column (3.5 µm, 130 Å, 1 mm × 150 mm, Waters) on an Ultimate 3000 HPLC system (ThermoFisher Scientific) at a flow rate of 42 µl/min. Peptides were loaded on the column in buffer A and eluted using the following gradient: from 2% B to 15% B over 3 min, to 45% B over 59 min, and to 80% B over 3 min, followed by 9 min at 80% B, to 2% B over 2 min, followed by 2 min at 2% B. Buffer A was 20 mM ammonium formate in water and buffer B was 20 mM ammonium formate in 90% acetonitrile. Both buffers were adjusted to pH 10. Elution of peptides was monitored with a UV detector (205 nm and 214 nm), and a total of 36 fractions were collected, pooled into 12 fractions using a post-concatenation strategy as previously described (Wang et al, 2011), and dried under vacuum.

## LC–MS analysis

Dried peptides were resuspended in 0.1% aqueous formic acid and subjected to LC–MS/MS analysis using an Exploris 480 Mass Spectrometer fitted with an Ultimate 3000 nano-LC (both Thermo-Fisher Scientific) and a custom-made column heater set to 60 °C. Peptides were resolved using a RP-HPLC column (75 µm × 30 cm) packed in-house with C18 resin (ReproSil-Pur C18–AQ, 1.9 µm resin; Dr. Maisch GmbH) at a flow rate of 0.3 µl min⁻¹. The following gradient was used for peptide separation: from 2% B to 10% B over 5 min, to 30% B over 70 min, to 50% B over 15 min, to

95% B over 2 min, followed by 18 min at 95% B. Buffer A was 0.1% formic acid in water and buffer B was 80% acetonitrile and 0.1% formic acid in water. The mass spectrometer was operated in DDA mode with a FAIMS device attached. FAIMS was run in standard resolution mode with two alternating CV voltages of −45 and −60 V. The total cycle time was ~3 s (1.5 s per CV voltage). Each MS1 scan was followed by high-collision-dissociation (HCD) of the most abundant precursor ions with dynamic exclusion set to 30 s. For MS1, the AGC target was set to 300% with a fill time of 25 ms using a resolution of 120,000 FWHM (at 200 $m/z$). MS2 scans were acquired at a target setting of 200%, maximum accumulation time of 100 ms, and a resolution of 30,000 FWHM (at 200 $m/z$). Singly charged ions and ions with an unassigned charge state were excluded from triggering MS2 events. The normalized collision energy was set to 32%, the mass isolation window was set to 0.7 $m/z$, and one microscan was acquired for each spectrum. In addition, the precursor fit threshold was set to 70% at a fit mass window size of 0.7 $m/z$.

The acquired raw files were searched against a protein database containing sequences of the predicted SwissProt entries of *Homo sapiens* (EBI, release date 2020/04/17) and commonly observed contaminants (in total 20,743 sequences) using the SpectroMine software (Biognosys, version 1.0.20235.13.16424). Standard Pulsar search settings for TMTpro ("TMTpro_Quantification") were used, and the resulting identifications and corresponding quantitative values were exported on the PSM level using the "Export Report" function. Acquired reporter ion intensities in the experiments were employed for automated quantification and statistical analysis using our in-house developed SafeQuant R script (v2.3) (Ahrné et al, 2016). This analysis included adjustment of reporter ion intensities, global data normalization by equalizing the total reporter ion intensity across all channels, summation of reporter ion intensities per protein and channel, calculation of protein abundance ratios, and testing for differential abundance using empirical Bayes moderated t-statistics. Finally, the calculated $P$ values were corrected for multiple testing using the Benjamini −Hochberg method.

In total, 8197 proteins were detected, annotated by the respective gene symbols, and ranked according to their logFC in each contrast. Subsequently, the pre-ranked enrichment analysis was performed as implemented in Bioconductor packages fgsea, version 1.22.0 (Korotkevich et al, 2021). The reference human Hallmark gene set was obtained from the Molecular Signatures Database (MSigDB) using the package msigdbr, version 7.5.1 (Liberzon et al, 2011).

## Microarray data analysis

The microarray data were obtained from the GEO database (GSE79761) and annotated using Affymetrix Human Genome U133 Plus 2.0 Array annotation data (R package hgu133plus2.db, version 3.13.0). For genes detected with multiple probes, those with the highest variability (measured by IQR) were kept for the analysis. The function normalizeBetweenArrays from limma package version 3.52.4 (Ritchie et al, 2015) was used to normalize the data by the "Cyclic Loess" method. Differential expression analysis was performed using the limma-trend approach with empirical Bayes moderations of the standard errors. The calculated $P$ values were corrected for multiple testing using the

Benjamini–Hochberg procedure. The gene set enrichment analysis was performed using the fgsea package version 1.22.0 (Korotkevich et al, 2021), with ranking based on logFC in each contrast. The reference human Hallmark gene set was obtained from the Molecular Signatures Database (MsigDB) using the package msigdbr, version 7.5.1 (Liberzon et al, 2011).

## RNA-Seq data analysis

In total, 200 ng of extracted RNA was used to build the library via the Illumina TruSeq Stranded mRNA Library Preparation Kit. Sequencing was done with Illumina NovaSeq 6000 (reagent kit v1.5), PE 2×51. Reads were aligned to the human genome (UCSC version hg38AnalysisSet) with STAR (Dobin et al, 2013), version 2.7.11, using the multi-map settings outFilterMultimapNmax = 10 (maximal number of multiple alignments allowed for a read) and outSAMmultNmax = 1 (maximal number of SAM lines for each mapped read). The output was sorted and indexed with SAMtools (Li et al, 2009), version 1.11. Duplicate reads were marked using MarkDuplicates from Picard, version 3.0.0. Strand-specific coverage tracks per sample were generated by tiling the genome in 20-bp windows and counting overlapping reads in each window using the bamCount function from the Bioconductor package bamsignals (Bioconductor(Gentleman et al, 2004; Liao et al, 2014) version 3.18). The rsubread::featureCounts function was used to count the number of reads (5'ends) overlapping with the exons of each gene, assuming an exon union model.

After removal of the low-expressed genes by filterByExpr from edgeR (Robinson et al, 2009) package, the count matrix contained 22,024 genes and 24 samples. Normalization factors were obtained by calcNormFactors using the trimmed mean of M-values (TMM). Differential gene expression was performed according to the edgeR User's Guide for each genotype (D538G and Y537S) separately. In short, gene-wise dispersion was computed using the estimateDisp function, and quasi-likelihood methods to account for gene-specific variability were used to fit the generalized log-linear model through the function glmQLFit. Gene set enrichment analysis was performed in the same way as in the case of the microarray data.

To obtain a GR activity signature, 341 protein-coding genes with the strongest shared effect at 24 h ($|\log FC| > 2$, FDR < 0.01) for both MCF-7- D538G and Y537S models were selected. Further, these genes were clustered across all 24 samples using k-means clustering with $k = 4$ and mean-centered logCPM values. The cluster "cl_2" containing a subset of 52 genes with the strongest expression onset between the early timepoint (8 h) and the control samples was further shortlisted, defining the GR activity signature.

## ChIP-Seq data analysis

Paired-end reads (PE 2 × 51) were aligned to the UCSC hg38 genome using bowtie2, version 2.4.2, with parameters "--maxins 2000" (the maximum fragment length for valid paired-end alignments), "--no-mixed, --no-discordant, --local" (allowing for only paired, concordant and local alignments). Duplicate reads were marked using MarkDuplicates from Picard, version 3.0.0., The resulting BAM files were sorted and indexed with samtools version 1.20., and used for peak calling with macs2, version 2.2.9.1. The parameters "—nomodel, —shift 0, —keep-dup all, —format BAMPE" were used. The q-value cutoff of significant enrichment of the signal above control sample was

0.05. All "—treatment" samples were analyzed by macs2 together in one run. The list of called peaks was cleaned from those identified in mitochondria, chrX and chrY, and ENCODE blacklist regions. The subsequent analysis was performed in R. Detected peaks were further filtered according to their enrichment above the signal from the input samples, and only peaks with log-enrichment higher than 2 in at least one sample were kept. Altogether, 20,280 peaks were identified.

Binned motif enrichment analysis was performed using monaLisa (Machlab et al, 2022) package, version 1.8.1. Position weight matrices of transcription factors for vertebrates were obtained from JASPAR2020. Function calcBinnedMotifEnrR uses Fisher's test to identify over- or underrepresented motifs in each bin with respect to the remaining ones. The function takes into account GC and k-mer composition differences between fore- and background sets.

The deepTools 3.5.6 suite was used to generate signal heatmap and profile plots, using bed files of defined regions and bigwig files for signal, as input. GREAT (from the Bejerano lab, http://great.stanford.edu/public/html/) was used for peaks to genes association (settings are indicated in the corresponding legends).

## Survival analysis

Survival predictions in Figs. 5A and EV5A,C; Appendix Fig. S4 were performed using R package survival, version 3.4-0 (Therneau, 2020) implementing the Cox proportional hazards model via the survfit function on the METABRIC (Curtis et al, 2012) data (release date August 2019). Clinically assigned ER+ breast cancer patients were selected and stratified based on their *NR3C1* normalized log-expression (Fig. EV5A). Namely, patients within the highest and the lowest quartiles were assigned to the groups of high and low expression, respectively. For Figs. 5A and EV5C; Appendix Fig. S4, the patients were stratified according to their total log-expression of the signature genes. Those in the top quartile were classified as "High", those in the lowest quartile as "Low", all others as "Intermediate".

## Ingenuity pathway analysis (IPA)

For the identification of upstream regulators, the differentially regulated proteins (Proteomics; log FC > 0.5 and FDR < 0.05) and genes (RNA-Seq; Log FC > 1 and FDR < 0.01) were subjected to QIAGEN Ingenuity Pathway Analysis.

## Chromatin immunoprecipitation (ChIP)

Cells were grown as monolayer cultures in 15-cm dishes and serum starved for 48 h using 2.5% charcoal-stripped FCS containing phenol red-free DMEM. They were treated with vehicle or Dex for 60 min and then fixed in 1.5% formaldehyde (Sigma, cat no. F8775) for 10 min at room temperature, followed by quenching by addition of 1 M glycine (Sigma, cat no. G8898-500G). The fixed cells were washed twice and collected by scraping into ice-cold PBS supplemented with 1X protease inhibitor (Roche, cat no. 11836153001). Cells were centrifuged and resuspended in lysis buffer (10 mM EDTA, 50 mM Tris-HCl pH 8.1, 30% Empigen BB, 1% SDS) supplemented with 1× protease inhibitor. The lysed cells were vortexed and incubated on ice for 15 min and subsequently sonicated using a Diagenode Bioruptor sonicator UCD-300 to obtain fragments of 300–1000 bp (4 cycles with 30 s of ON and 30 s

OFF). The supernatant was collected after centrifugation at 4 °C and 10,000×g. 10% of the total lysate was collected as the input fraction. To block SDS, 15 μl of 10% Triton was added to the fragmented lysates after sonication. For each ChIP, 150 μl of lysate (corresponding roughly to 10 million cells) was incubated with IP buffer (2 mM EDTA, 150 mM NaCl, 20 mM Tris-HCl pH 8.1, 0.1× Triton X-100 protease inhibitor), Yeast tRNA, and specific antibodies overnight at 4 °C under rotation. Anti-GR, anti-ER antibodies (2 μg), anti-H3K27ac (1.5 μg), and a normal rabbit IgG antibody were used as a control (Thermo, H-270, 200 μg/ml). Immunoprecipitated complexes were recovered on Dynabead Protein G beads (ThermoFisher, cat. no. 10003D) and, after extensive washing, DNA was recovered by reverse crosslinking (overnight at 65 °C) and purified using the QIAquick PCR purification kit (Qiagen). For ChIP-quantitative PCR (qPCR), real-time PCRs were performed using the fast SYBR green master mix reagent (Applied Biosystems). Primer sequences used for ChIP-PCR experiments were designed using PRIMER3.

| Genes | Forward primer | Reverse primer |
| --- | --- | --- |
| SGK1 | GGGAGGGAGAGGTCAGGAAT | TCGCTTGTTACCTCCTCACG |
| ESR1_1 | ACCATCTAAGTGCATCCCAAA | CTGCAGTTAACGTCCAGGTG |
| ESR1_2 | GTGATCCAAATGCCTGCTCT | TGTCCCCAAGTCCCAATACC |
| ESR1_3 | CACCCCATTCTATCTGCCCT | GGCCGGCTTTCTCTAATGTG |
| AREG | AGCTAGAAGCGTTTGCAGGA | TGGGTGTGTGTTTGTCCACT |
| PGR | GCAGGACGACTTCTCAGACC | GCCTGACCTGTTGCTTCAAT |
| GREB1 | GCACACTTTCCACCTCTGCT | GCTCGCTTGCAAAGTCAAAT |
| TFF1 | GTCGCACTTCTCGAAGGTCT | CAGGTAAGGCGTGCTTCTTC |

## Statistical analysis

Randomization of the animals of the same age and cell-line groups were performed based on standard laboratory practice procedures. The investigators were not blinded to allocation during experiments and outcome assessment. Values represent the mean ± SD unless mentioned otherwise. Depending on the type of experiment, data were tested for normal distribution and analyzed using two-tailed t test, nonparametric Mann–Whitney U test, Ordinary one-way ANOVA, Ordinary two-way ANOVA or the Kruskal–Wallis test as indicated in the figure legends. For paired samples, two-tailed paired t test or Wilcoxon matched-pairs signed-rank test were used for analysis. Kaplan–Meier survival analysis was performed using survival calculation tool from GraphPad Prism and the significance was calculated using the log-rank (Mantel–Cox) test. Parametric statistical tests were used to compare groups with similar variance, otherwise groups were compared using nonparametric statistical test. Sample size was determined to be adequate based on the magnitude and consistency of measurable differences between groups.

## Data availability

The mass spectrometry proteomics data have been uploaded into the MassIVE repository with the dataset identifier MSV000092510. The data

### The paper explained

**Problem**

Elevated expression of Glucocorticoid Receptor (GR) is associated with low tumor grade and improved outcome in early-stage estrogen receptor (ER) positive breast cancer. However, the effects of GR activation and its crosstalk with ER on the metastatic progression of ER+ disease remain largely unknown.

**Results**

Using preclinical metastatic ER+ in vivo and ex vivo models, this study shows that activation of GR by the synthetic glucocorticoid Dexamethasone (Dex), reduces liver metastases and prolongs survival of the animals by inhibiting cell intrinsic mechanisms essential for cancer cell growth in the metastatic niche. We found that activated GR inhibits ER signaling by silencing most estrogen response-related genes and proteins. Moreover, Dex treatment downregulates ESR1 gene expression and triggers protein loss, thereby directly silencing ER. ChIP-Seq revealed that Dex treatment also dramatically decreases chromatin binding of ER on its regulatory elements. Notably, Dex treatment decreases ER abundance in patient-derived tumor organoids (PDO and PDXO). Finally, a GR activity signature predicts a good outcome for patients with ER+ breast cancer and anti-correlates with ESR1 expression.

**Impact**

Therapeutic options for advanced metastatic disease are still an unmet clinical need. This study reveals a mechanism of action of GR which upon activation by glucocorticoids, elicits a direct repression on ER. Hence, this may be clinically beneficial for patients suffering from treatment-refractory ER+ metastatic disease.

can be accessed at http://massive.ucsd.edu/ProteoSAFe/status.jsp?task=2c3ad1c15afe4b1f9014b71fd8bcedb9, with the following credentials. Username: MSV000092510; password: PCF. The RNA-Seq and ChIP-Seq datasets have uploaded into the GEO repository and can be accessed at https://www.ncbi.nlm.nih.gov/geo/query/acc.cgi?acc=GSE286892 (GEO accession number: GSE286892, and token: obixqkwcddgxrap). The code for data analysis can be accessed on the Zenodo repository (link: https://zenodo.org/records/16993024).

The source data of this paper are collected in the following database record: biostudies:S-SCDT-10_1038-S44321-025-00342-z.

## Peer review information

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

## Acknowledgements

We thank members of the Bentires-Alj laboratory for scientific advice and discussions and Michael T. Lewis and Advanced In Vivo Models Core Unit (Baylor College of Medicine) for providing PDX to establish PDXO_1 and PDXO_2 organoid lines. Ben Ho Park (Vanderbilt University Medical Center) kindly provided the MCF-7 Y537S and D538G cells. We thank the following people and facilities at the Department of Biomedicine for their valuable support - Diego Calabrese from the DBM Histology facility for help with Ventana staining for immunohistochemistry, Milica Vulin and Milan MS Obradović for scientific discussions and advice, the DBM Imaging Core facility and the DBM Flow Cytometry facility for assisting with FACS experiments, the DBM animal facility for the maintenance and welfare of the animals, and the Scientific Computing Center at the University of Basel (sciCORE; http://scicore.unibas.ch/). The raw sequencing data were processed using a data pipeline developed and maintained by the DBM Bioinformatics core facility of the University of Basel, and we thank specifically Robert Ivanek and Florian Geier as well as Michael Stadler (FMI), for discussions on computational analysis related to RNA-Sequencing and ChIP-Sequencing. MM is supported by funding from Dr. Arnold U and Susanne Huggenberger-Bischoff Foundation, Switzerland. CJ was supported by a Marie Skłodowska-Curie Actions Intra-European Fellowship (Horizon 2020, Project EPICAN 841872) and the Excellence Junior Research Fund of the University of Basel (3MM1062). MH received research support from Krebsliga beider Basel (Cancer League Basel; KLbB-5572-02-2022) and the Department of Surgery, University Hospital of Basel. Research in the Bentires-Alj laboratory is supported by the European Research Council (ERC advanced grant 694033 STEM-BCPC), the Swiss National Science Foundation, the Krebsliga Beider Basel, the Swiss Cancer League (KFS-4414-02-2018), the Swiss Personalized Health Network (Swiss Personalized Oncology driver project), and the Department of Surgery, University Hospital Basel.

## Author contributions

**Madhuri Manivannan**: Conceptualization; Resources; Data curation; Formal analysis; Funding acquisition; Validation; Investigation; Methodology; Writing —original draft; Project administration. **Charly Jehanno**: Conceptualization; Resources; Data curation; Formal analysis; Supervision; Funding acquisition; Validation; Visualization; Writing—original draft; Project administration; Writing—review and editing. **Michal Kloc**: Data curation; Formal analysis; Validation. **Jorge Gomez Miragaya**: Investigation; Methodology. **Maren Diepenbruck**: Formal analysis; Funding acquisition; Investigation. **Katrin Volkmann**: Data curation; Formal analysis; Funding acquisition. **Marie-May Coissieux**: Resources; Funding acquisition; Investigation. **Marta Palafox**: Resources; Investigation. **Adelin Rouchon**: Data curation; Funding acquisition; Investigation. **Nicolas Kramer**: Funding acquisition; Investigation. **Alexander Schmidt**: Data curation; Formal analysis; Funding acquisition; Methodology. **Yannick Blum**: Formal analysis. **Baptiste Hamelin**: Validation; Investigation. **Helen Schuster**: Resources; Funding acquisition. **Martin Heidinger**: Resources; Validation; Investigation. **Simone Muenst**: Resources; Visualization. **Marcus Vetter**: Resources; Project administration. **Christian Kurzeder**: Resources; Project administration. **Walter P Weber**: Resources; Project administration. **Mohamed Bentires-Alj**: Supervision; Project administration; Formal analysis; Funding acquisition; Writing - review and editing.

Source data underlying figure panels in this paper may have individual authorship assigned. Where available, figure panel/source data authorship is listed in the following database record: biostudies:S-SCDT-10_1038-S44321-025-00342-z.

## Disclosure and competing interests statement

MM, CJ, MK, JGM, MD, NK, KV, MMC, MP, AR, AS, YB, BH, HS, MH, SM, MV, CK, WPW, and MBA declare no competing financial interests. MK is a co-founder of RXcel GmbH. MBA owns equities in and has received laboratory support and compensation from Novartis and has served as a consultant for Basilea. MBA is an editorial advisory board member. WPW received research support from Agendia, paid to the University Hospital Basel for the TAXIS study (OPBC-03, SAKK 23/16, IBCSG 57-18, ABCSG-53, GBG 101). MV is receiving grants from Swiss Cancer Research; Roche, GSK, ExactScience, personal fees and nonfinancial support from GSK Consumer Healthcare, AstraZeneca, Novartis Roche; and personal fees from Genomic Health, Eli Lilly and Company, Merck, MSD, Pfizer, and Daichii Sankyo, ASC Oncology outside the submitted work. SM reported receiving speaker fees from GSK Consumer Healthcare and advisory board fees from Novartis and Diaceutics outside the submitted work. CK received consulting fees from GSK, AstraZeneca, Novartis, Roche, Eli Lilly S.A., Pfizer, Genomic Health, Merck MSD, Novartis, PharmaMar, Tesaro; sits on advisory boards of GSK, AstraZeneca, Novartis, Roche, Eli Lilly S.A., Pfizer, Genomic Health, Merck MSD, Novartis, PharmaMar, Tesaro; received travel support from GSK, AstraZeneca, Roche.

# Expanded View Figures

**Figure EV1.   Nuclear translocation and genetic knockdown of GR in *ESR1* mutant models.**

(**A**) Immunoblots showing GR and ER abundance in *ESR1* wild-type *versus* mutant ER+ cells. ERK2 was used as a loading control. (**B**) Representative immunofluorescence images of MCF-7 D538G cells showing GR nuclear translocation after activation by Dex for 1 h. Scale 200 μM. (**C**) Representative immunohistochemistry images of metastatic livers from MCF-7 D538G xenografts showing the GR nuclear translocation after activation by Dex for 16 h prior to sacrifice; ×40 magnification, scale 50 μm. (**D**) Bar graph representing mRNA levels of *SGK1* in MCF-7 D538G cells after GR activation for 72 h; *n* = 3 biological replicates. Two-tailed *t* test. Data are presented as mean ± SD. (**E**) Bar graph depicting mRNA levels of *NR3C1* in cells transduced with control shRNA or shRNAs targeting GR (shGR1 and shGR2). *n* = 3 to 4 biological replicates. Two-tailed *t* test. (**F**) Bar graph depicting mRNA levels of *SGK1* after Dex treatment in cells transduced with control shRNA or shRNAs targeting GR (shGR1 and shGR2); *n* = 3 biological replicates. Two-tailed *t* test. (**G**) Bioluminescence imaging of mice intravenously injected with MCF-7 D538G cells transduced with control shRNA or shRNAs targeting GR (shGR1 and shGR2); *n* = 5 mice per group. Imaging was performed with Newton Vilber. (**H**) Graph representing estimations of metastatic burden in the whole body of mice over time. (**I**) Representative bioluminescence images of MCF-7 D538G liver metastases harvested at day 47. *n* = 9–10 mice per group. Imaging of livers was performed with Newton Vilber.

▶

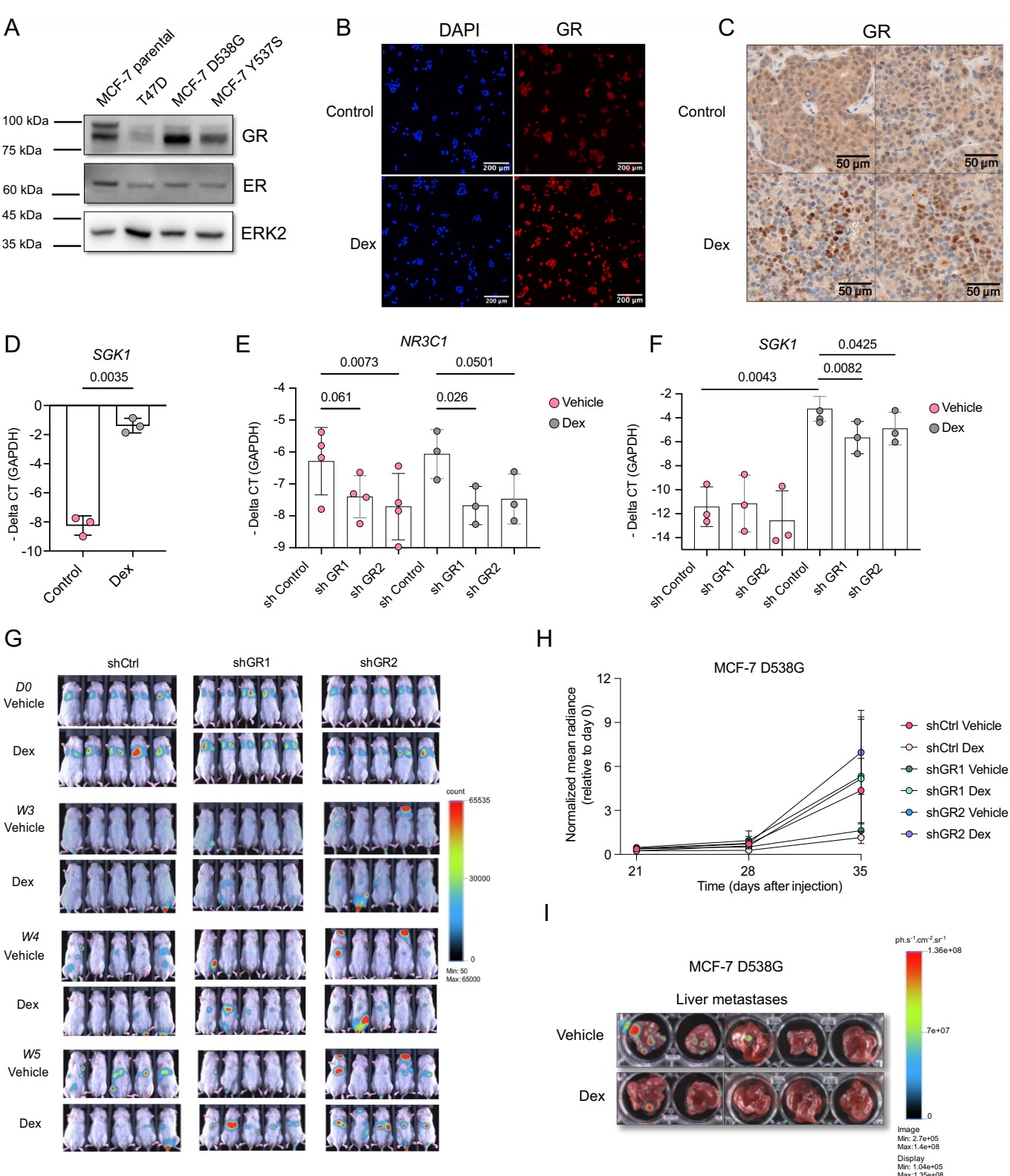

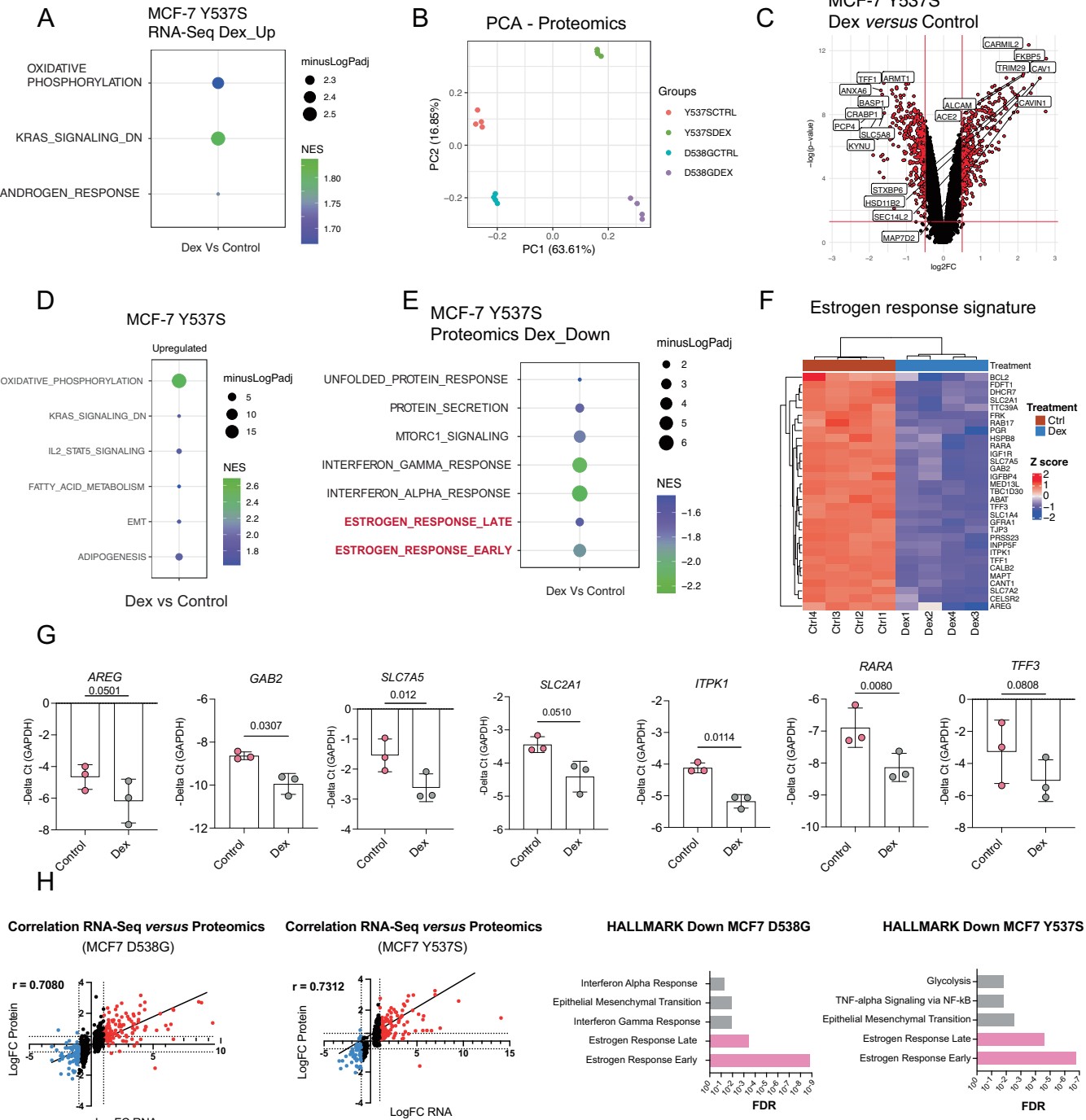

◀  **Figure EV2.** **Transcriptomic and proteomic profiling reveal downregulation of E2 response signatures upon Dex.**

(A) Bubble plot depicting the Hallmark signatures enriched in genes upregulated upon 24 h Dex treatment in MCF-7 Y537S model. Normalized enrichment score (NES) and – Log adjusted-value are indicated. (B) Principal Component Analysis showing treatment group-based separation of MCF-7 D538G and Y537S cells treated with Dex or vehicle for 7 days and analyzed by proteomics; $n = 4$ technical replicates per condition. (C) Volcano plot of differentially abundant proteins (adjusted $P$ value < 0.05 and |logFC| > 0.5) in Dex- versus vehicle-treated MCF-7 Y537S cells. Empirical Bayes moderated t-statistics with Benjamini–Hochberg correction. (D) Bubble plot depicting the NES of upregulated (adjusted $P$ value < 0.05) Hallmark signature after GR activation in MCF-7 Y537S model. Kolmogovo–Smirnov-like running-sum statistics with permutation test and Benjamini–Hochberg correction. (E) Bubble plot depicting the NES of downregulated (adjusted $P$ value < 0.05) Hallmark signature after GR activation in MCF-7 Y537S model. Kolmogorov–Smirnov-like running-sum statistics with permutation test and Benjamini–Hochberg correction. (F) Heatmap depicting the differentially abundant early estrogen response proteins (Hallmark gene set) downregulated after GR activation in MCF-7 D538G model; $n = 4$ technical replicates per condition. (G) Scatter dot plots showing mRNA levels of the early E2-response gene signature in MCF-7 D538G cells treated for 72 h with Dex or vehicle; $n = 3$ biological replicates. Two-tailed $t$ test. Data are presented as mean ± SD. (H) Left panels: correlation between genes and proteins found differentially regulated in the RNA-Seq and proteomics datasets for both MCF-7 D538G and Y537S models. "r" indicates Pearson coefficient, and the line represents simple linear regression. Right panels: Hallmark functional annotation of the genes and proteins whose expression was downregulated upon Dex in both RNA-Seq and proteomics datasets, for both models.

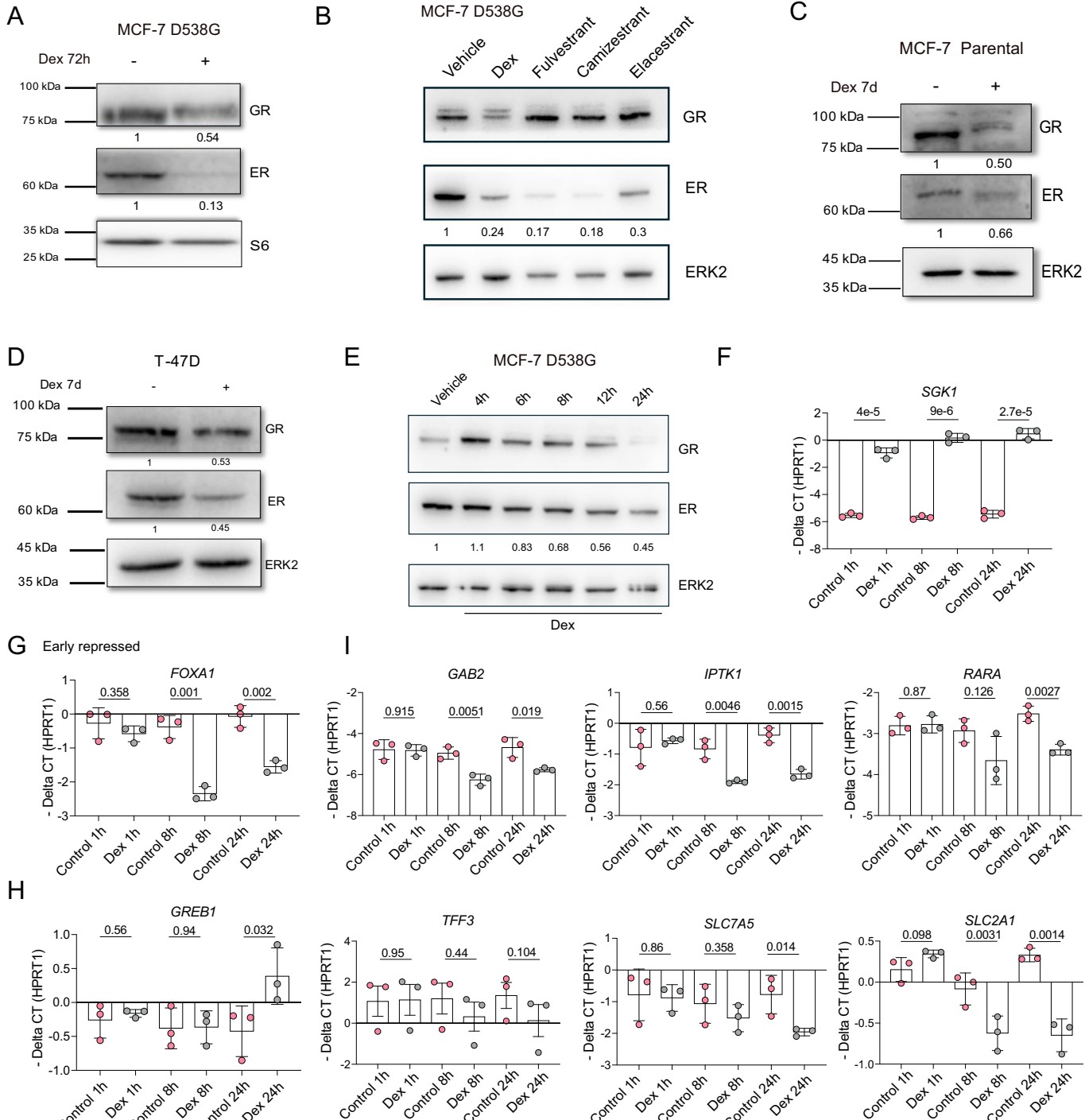

◄ **Figure EV3. GR activation decreases ER abundance and transcriptional activity.**

(A) Immunoblots showing levels of GR, ER and ERK2 (loading control) in MCF-7 D538G cells treated with Dex for 72 h. (B) Immunoblots showing levels of ER, GR and ERK2 (loading control) in MCF-7 D538G cells treated with Dex (700 nM), Fulvestrant (1 μM), Camizestrant (1 μM) or Elacestrant (1 μM) for 72 h. (C, D) Immunoblots showing levels of GR, ER and ERK2 (loading control) in MCF-7 parental and T-47D cells treated with Dex for 7 days. (E) Immunoblots showing the levels of GR, ER and ERK2 (loading control) in MCF-7 D538G cells treated or not with Dex for the indicated times. (F) Scatter dot plots showing the mRNA levels of *SGK1* after GR activation for the indicated times, in MCF-7 D538G cells. $n = 3$ biological replicates. Two-tailed t-test. Data are presented as mean ± SD. (G) Scatter dot plots showing the mRNA levels of *FOXA1* after GR activation for the indicated times, in MCF-7 D538G cells. $n = 3$ biological replicates. Two-tailed *t* test. Data are presented as mean ± SD. (H) Scatter dot plots showing the mRNA level of *GREB1*, at indicated Dex treatment timepoints, in MCF-7 D538G cells. $n = 3$ biological replicates. Two-tailed *t* test. Data are presented as mean ± SD. (I) Scatter dot plots showing mRNA levels of E2-response genes that are late repressed in MCF-7 D538G cells after treatment with Dex or vehicle for the indicated times. $n = 3$ biological replicates. Two-tailed *t* test. Data are presented as mean ± SD.

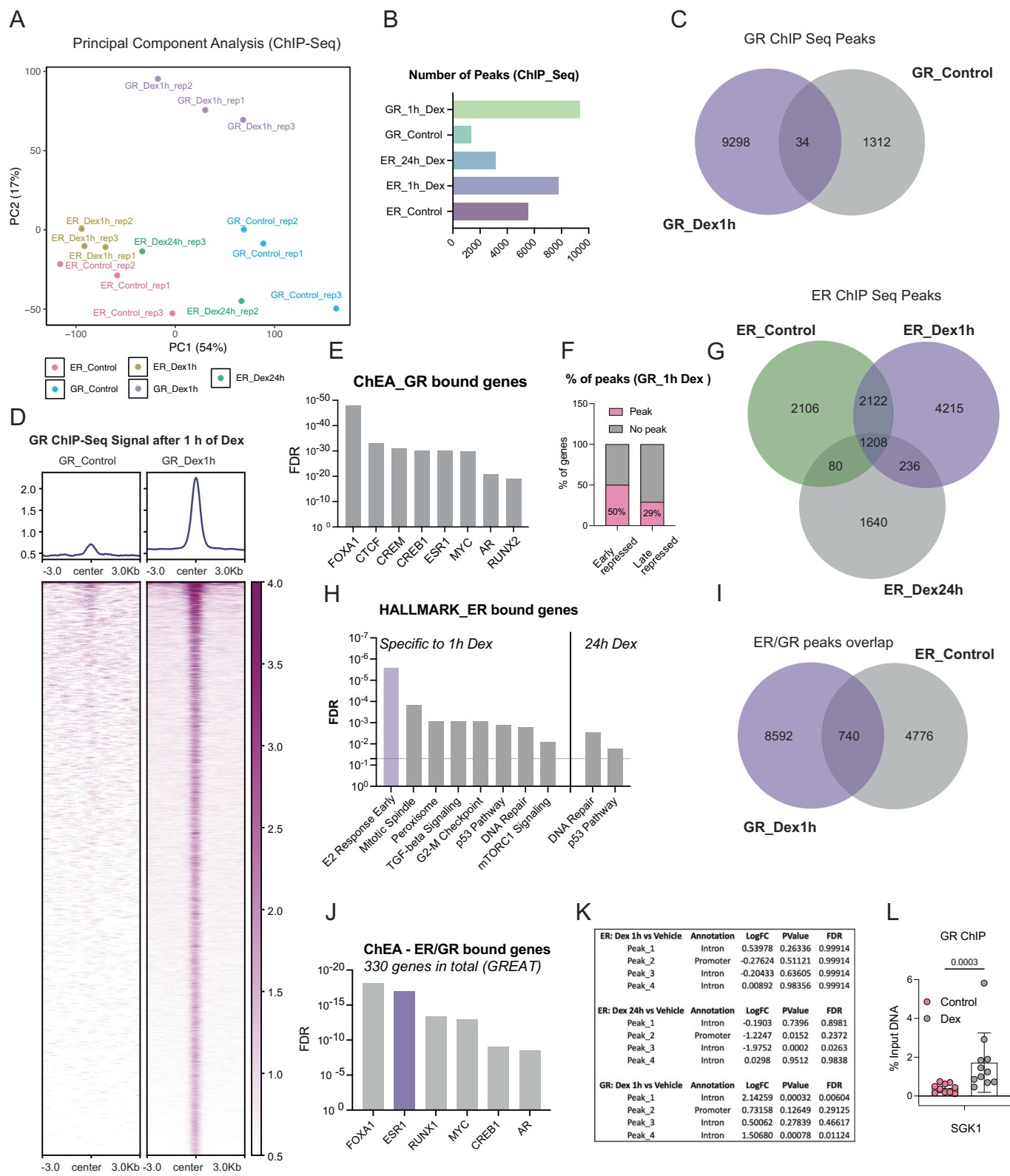

◀  **Figure EV4.  Analysis of ER and GR ChIP-Seq in MCF-7 *ESR1* D538G cells upon 1 h and 24 h Dex.**

(A) Principal component analysis of the ChIP-Seq samples. One technical replicate of ER_Dex24h condition did not pass quality control and was removed due to low percentage (32.4%) of uniquely mapped aligned sequence in comparison to all other samples ( > 80%). (B) Number of peaks detected per biological conditions in the ChIP-Seq experiment. (C) Venn diagram depicting the number of peaks and their overlap in the GR ChIP-Seq in absence or presence of Dex for 1 h. (D) Heatmap of GR ChIP-seq in absence or presence of Dex for 1 h in MCF-7 D538G cells. The heatmap is shown in a horizontal window of $-/+$ 3 kb around the center of the peaks. (E) ChEA transcription factor binding analysis of genes bound by GR in presence of Dex for 1 h. (F) Percentage of GR_1h Dex peaks identified by ChIP-Seq (in promoter, exonic, intronic and intergenic regions) located at the vicinity of early repressed and late repressed ER target genes, identified as downregulated by RNA-Seq upon 8 h and 24 h of Dex, respectively. (G) Venn diagram depicting the number of peaks and their overlap from the ER ChIP-Seq in absence or presence of Dex for 1 h or 24 h. (H) Hallmark functional annotation of genes bound by ER, specifically in the presence of Dex for 1 h or 24 h, using GREAT for peak to gene annotation. (I) Venn diagram depicting the number of peaks and their overlap between the ER_control and GR_Dex1h ChIP-Seq conditions. (J) ChEA transcription factor binding analysis of genes commonly bound by GR and ER, using GREAT for peak to gene annotation. (K) Table highlighting the Log FC, *P* value and FDR values corresponding to the 4 peaks (ER and/or GR) identified at the vicinity of the *ESR1* gene. (L) GR ChIP-qPCR of the GR target *SGK1* in MCF-7 D538G cells treated with Dex or vehicle for 1 h. Data are shown as percentage of input DNA. $n = 3$ experimental replicates. Two-tailed Mann–Whitney test. Data are presented as mean ± SD.

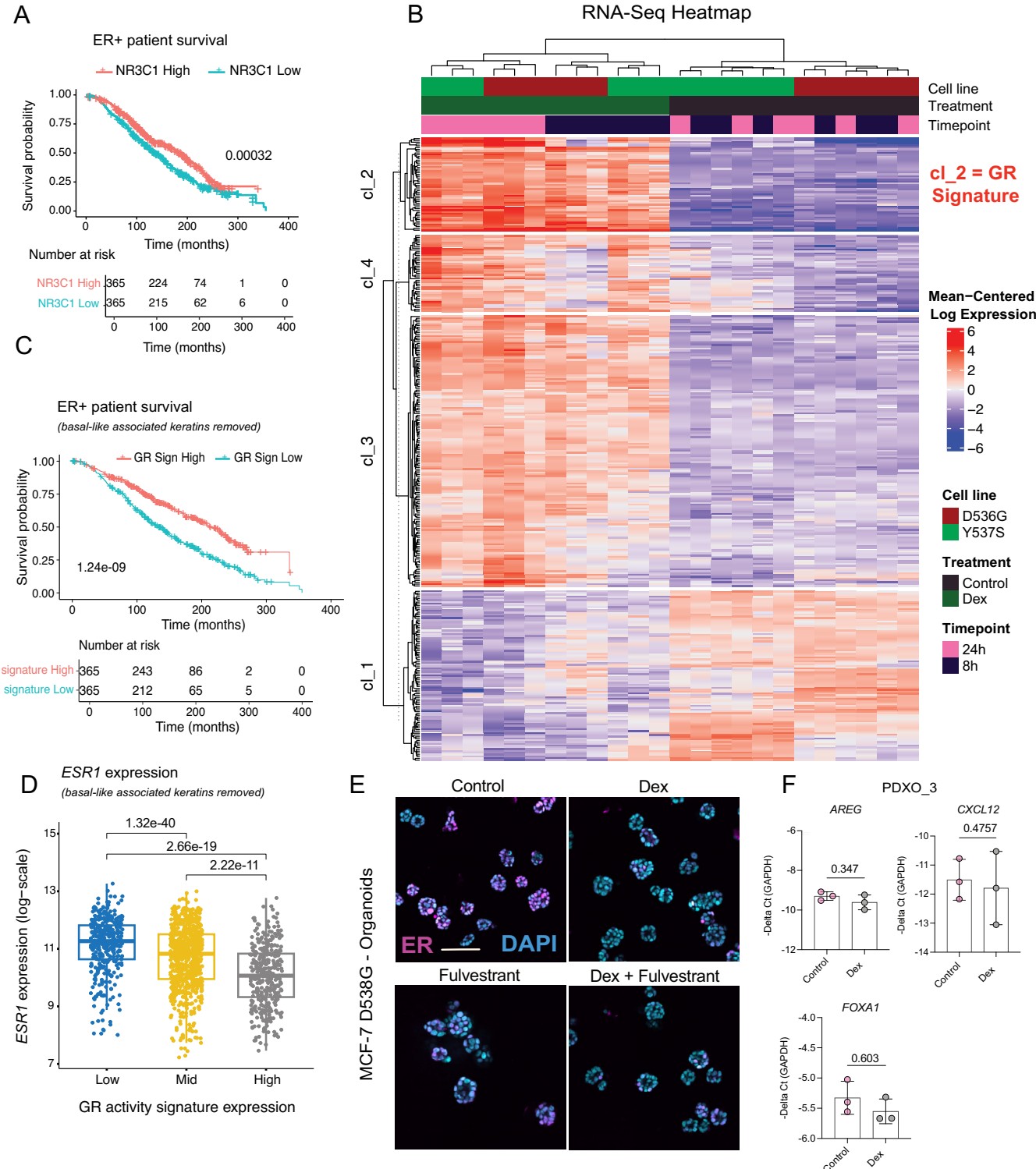

**A** ER+ patient survival

**B** RNA-Seq Heatmap

cl_2 = GR Signature

**C** ER+ patient survival
*(basal-like associated keratins removed)*

**D** *ESR1* expression
*(basal-like associated keratins removed)*

**E** MCF-7 D538G - Organoids

Control / Dex / Fulvestrant / Dex + Fulvestrant

ER — DAPI

**F** PDXO_3

*AREG* / *CXCL12* / *FOXA1*

◀

**Figure EV5. High *NR3C1* expression is associated with increased survival in patient with ER+ disease and GR activity signature generation.**

(A) Kaplan–Meier survival plot of ER+ luminal patients (METABRIC annotation) showing that elevated GR expression is associated with prolonged survival in ER+ breast cancer patients. Patients were stratified based on high and low expression of *NR3C1*; Cox proportional hazard model with log-rank test. (B) Heatmap built from genes differentially regulated following RNA-Seq profiling of both MCF-7 D538G and Y537S models, treated or not with Dex for 8 h and 24 h. k-means (4) method was used for hierarchical clustering. Cluster_2 depicts the GR activity signature composed of 52 protein-coding genes, upregulated upon Dex treatment for 8 h and 24 h (RNA-Seq; Log FC > 2, FDR < 0.01), in both MCF-7 D538G and Y537S models. (C) Kaplan–Meier survival plot showing predictive value of GR activity signature (lacking basal-like keratins-encoding genes *KRT5, KRT6a, KRT6b, KRT6c* and *KRT16*) in patients with ER+ luminal breast cancer (METABRIC). Patients were stratified based on high and low GR activity signature score; Cox proportional hazard model with log-rank test. GR activity signature is composed of 52 protein-coding genes, upregulated upon Dex treatment for 8 h and 24 h in both MCF-7 Y537S and D538G models (see "Methods" for signature generation). (D) Graph showing *ESR1* mRNA expression in breast tumors from patients with ER+ disease (METABRIC), stratified according to the GR activity signature score (without basal-like associated keratins); low, intermediate, high. Wilcoxon matched pairs signed-rank test. (E) Representative confocal images (z-plan, 5 μM z-distance) of 3D-grown MCF-7 D538G cells after 3 days of the indicated treatments. Scale 50 μM. (F) Scatter dot plots showing the mRNA levels of *AREG, CXCL12* and *FOXA1* after 24 h of Dex treatment in PDXO_3 sample: $n = 3$ experimental replicates. Two-tailed *t* test. Data are presented as mean ± SD.

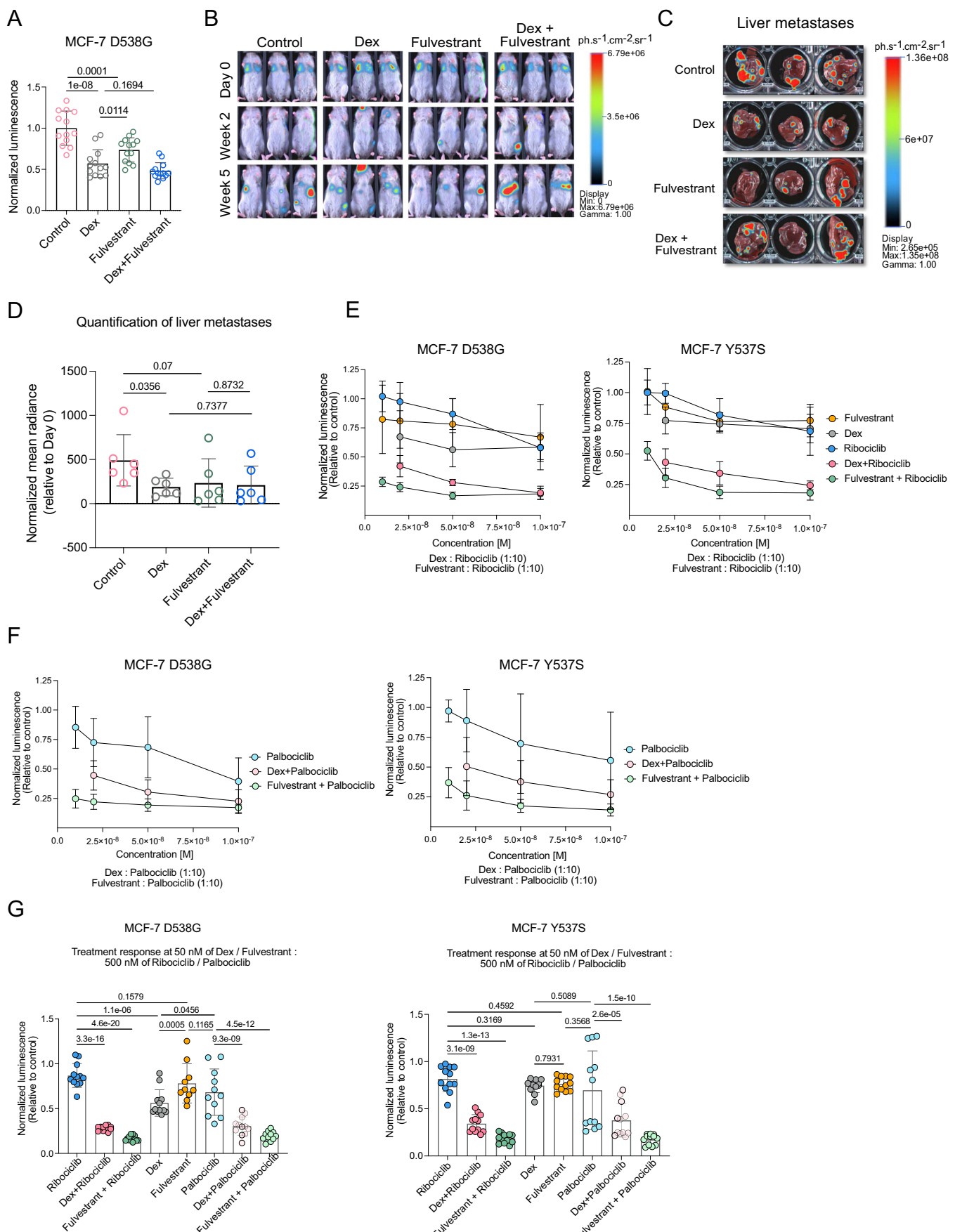

◀ **Figure EV6. GR activation decreases liver metastatic burden of MCF-7 *ESR1* mutant cancer cells compared to Fulvestrant, and synergizes with CDK4/6i in vitro.**

(A) Bar graph showing the viability of MCF-7 D538G cells treated with vehicle, Dex, or Fulvestrant alone or in combination for 4 days; $n = 3$ experimental replicates, with 4–5 technical replicates each. Two-tailed Mann–Whitney test. Data are presented as mean ± SD. (B) Bioluminescence imaging of mice intravenously injected with MCF-7 D538G cells without prior treatments in vitro; $n = 6$ mice per group. Imaging was performed with Newton Vilber. (C) Representative bioluminescence images of liver metastases harvested upon sacrifice; $n = 6$ mice per group. (D) Bar graph showing the quantification of liver metastases harvested and measured at day 47; $n = 6$ mice per group. Two-tailed Mann–Whitney test. Data are presented as mean ± SD. (E) Graphs showing the viability of MCF-7 D538G and Y537S cells upon treatment with Dex, Fulvestrant, Ribociclib or the combination of drugs for 6 days. $n = 2$ experimental replicates, with 6 technical replicates each. Data are presented as mean ± SD. (F) Graphs showing the viability of MCF-7 D538G and Y537S cells upon treatment with Palbociclib or in combination with Dex or Fulvestrant for 6 days. $n = 2$ experimental replicates, with 6 technical replicates each. Data are presented as mean ± SD. (G) Scatter dot plots representing the viability of MCF-7 D538G and Y537S cells treated with 50 nM of Dex or Fulvestrant, 500 nM of Ribociclib or Palbociclib or the combinations as indicated. $n = 2$ experimental replicates, with 6 technical replicates each. One-way ANOVA test. Data are presented as mean ± SD.

