## [Peer Review File · EMBO Molecular Medicine]

Activated Glucocorticoid Receptor is an Estrogen Receptor Silencer in ER+ metastatic breast cancer

Madhuri Manivannan, Charly Jehanno, Michal Kloc, Jorge Gomez-Miragaya, Maren Diepenbruck, Katrin Volkmann, Marie-May Coissieux, Marta Palafox, Adelin Rouchon, Nicolas Kramer, Alexander Schmidt, Yannick Blum, Baptiste Hamelin, Helen Schuster, Martin Heidinger, Simone Muenst, Marcus Vetter, Christian Kurzeder, Walter Weber, and Mohamed Bentires-Alj

Corresponding author: Mohamed Bentires-Alj (m.bentires-alj@unibas.ch)

Review Timeline:

Submission Date:	3rd Mar 25
Editorial Decision:	28th Mar 25
Revision Received:	30th Jul 25
Editorial Decision:	5th Sep 25
Revision Received:	1st Oct 25
Editorial Decision:	15th Oct 25
Revision Received:	17th Oct 25
Accepted:	24th Oct 25

Editor: Lise Roth

Transaction Report:

28th Mar 2025

Dear Momo,

Thank you for the submission of your manuscript to EMBO Molecular Medicine.

We have now received the reports from the 2 referees who reviewed your new submission. Referee #1 already reviewed your previous submission (as referee #3), while referee #2 is a new referee.

As you will see below, both referees acknowledge that the new manuscript is significantly improved but also identify remaining issues that should be addressed (including, but not limited to, unclear clinical relevance and contradictions with previous publications). While most of the points can be addressed by appropriate discussion, we would encourage you to add experimental revisions where possible to strengthen your conclusions.

In particular, the negative correlation between ER and GR and clinical relevance (see reviewer #1 point #1) need to be convincingly demonstrated for further consideration of the manuscript.

Addressing the reviewers' concerns in full will be necessary for further considering the manuscript in our journal, and acceptance of the manuscript will entail a second round of review. EMBO Molecular Medicine encourages a single round of revision only and therefore, acceptance or rejection of the manuscript will depend on the completeness of your responses included in the next, final version of the manuscript. For this reason, and to save you from any frustrations in the end, I would strongly advise against returning an incomplete revision.

We are expecting your revised manuscript within three to four months, if you anticipate any delay, please contact us.

We require:

4) A .docx formatted letter INCLUDING the reviewers' reports and your detailed point-by-point responses to their comments. As part of the EMBO Press transparent editorial process, the point-by-point response is part of the Review Process File (RPF), which will be published alongside your paper.

5) A complete author checklist, which you can download from our author guidelines (<https://www.embopress.org/page/journal/17574684/authorguide#submissionofrevisions>). Please insert information in the checklist that is also reflected in the manuscript. The completed author checklist will also be part of the RPF.

6) All Materials and Methods need to be described in the main text using our 'Structured Methods' format. According to this format, the Methods section includes a Reagents and Tools Table (listing key reagents, experimental models, software and relevant equipment and including their sources and relevant identifiers) followed by a Methods and Protocols section describing the methods, ideally using a step-by-step protocol format. The aim is to facilitate adoption of the methodologies across labs. Please download and fill our Reagents and Tools Table template (.docx), which you can find in our author guidelines: <https://www.embopress.org/page/journal/14693178/authorguide#structuredmethods>.

<https://www.embopress.org/doi/10.15252/msb.20178071>

7) Please note that all corresponding authors are required to supply an ORCID ID for their name upon submission of a revised manuscript.

8) It is mandatory to include a 'Data Availability' section after the Materials and Methods. Before submitting your revision, primary datasets produced in this study need to be deposited in an appropriate public database, and the accession numbers and database listed under 'Data Availability'. Please remember to provide a reviewer password if the datasets are not yet public (see <https://www.embopress.org/page/journal/17574684/authorguide#dataavailability>).

9) For data quantification: please specify the name of the statistical test used to generate error bars and P values, the number (n) of independent experiments (specify technical or biological replicates) underlying each data point and the test used to calculate p-values in each figure legend. The figure legends should contain a basic description of n, P and the test applied. Graphs must include a description of the bars and the error bars (s.d., s.e.m.). Please provide exact p values.

10) Our journal encourages inclusion of *data citations in the reference list* to directly cite datasets that were re-used and obtained from public databases. Data citations in the article text are distinct from normal bibliographical citations and should directly link to the database records from which the data can be accessed. In the main text, data citations are formatted as follows: "Data ref: Smith et al, 2001" or "Data ref: NCBI Sequence Read Archive PRJNA342805, 2017". In the Reference list, data citations must be labeled with "[DATASET]". A data reference must provide the database name, accession number/identifiers and a resolvable link to the landing page from which the data can be accessed at the end of the reference. Further instructions are available at .

11) We replaced Supplementary Information with Expanded View (EV) Figures and Tables that are collapsible/expandable online. EV Figures should be cited as 'Figure EV1, Figure EV2' etc... in the text and their respective legends should be included in the main text after the legends of regular figures.

12) The paper explained: EMBO Molecular Medicine articles are accompanied by a summary of the articles to emphasize the major findings in the paper and their medical implications for the non-specialist reader. Please provide a draft summary of your article highlighting

13) Author contributions: CRediT has replaced the traditional author contributions section because it offers a systematic machine readable author contributions format that allows for more effective research assessment. Please remove the Authors Contributions from the manuscript and use the free text boxes beneath each contributing author's name in our system to add specific details on the author's contribution. More information is available in our guide to authors.

Please also suggest a visual abstract to illustrate your article as a PNG file 550 px wide x 300-600 px high. A cropped portion of this image will serve as thumbnail for the table of content on our webpage.

16) As part of the EMBO Publications transparent editorial process initiative (see our Editorial at <http://embomolmed.embopress.org/content/2/9/329>), EMBO Molecular Medicine will publish online a Review Process File (RPF) to accompany accepted manuscripts.

In the event of acceptance, this file will be published in conjunction with your paper and will include the anonymous referee reports, your point-by-point response and all pertinent correspondence relating to the manuscript. Let us know whether you agree with the publication of the RPF and as here, if you want to remove or not any figures from it prior to publication. Please note that the Authors checklist will be published at the end of the RPF.

I look forward to receiving your revised manuscript.

With kind regards,

Lise

***** Reviewer's comments *****

Referee #1 (Remarks for Author):

The present manuscript by Manivannan et al. addresses an interesting and clinically relevant topic - the potential role of GR activation as a therapeutic strategy in ER+ metastatic breast cancer harboring ESR1 mutations. The authors provide comprehensive transcriptomic, proteomic, and genomic data, suggesting that GR activation by dexamethasone (Dex) significantly suppresses ER expression and transcriptional activity, reducing metastatic burden and enhancing chemotherapy efficacy in preclinical models. Despite these strengths, several important issues and concerns exist. These concerns particularly pertain to clinical applicability, mechanism interpretation, statistics, and certain discrepancies with previously published literature and clinical data.

Major Comments

- GR activation is suggested to induces marked ER degradation and chromatin dissociation in their preclinical models in the current manuscript. However, careful examination of their own clinical metastatic cohort data (Figure 5c) does not convincingly support a negative correlation between ER and GR abundance. In fact, most patient samples show clear nuclear co-expression of ER and GR, contradicting the notion of complete ER loss following GR activation. This inconsistency between their preclinical data (showing ER silencing/degradation) and actual clinical samples (retaining substantial ER positivity even in the presence of active GR) must be explicitly acknowledged and discussed. It raises significant questions about the clinical translatability of their observations and strongly suggests that their experimental ER loss phenomenon may reflect preclinical model-specific or condition-specific artifacts rather than clinically applicable biology.

- The finding and the claim that dexamethasone (Dex) treatment enhances Paclitaxel efficacy, improving survival outcomes and reducing metastatic burden is puzzling. This conclusion is particularly striking and, to some extent, contradictory to the established literature. Prior studies have demonstrated that glucocorticoids can antagonize the effects of chemotherapy agents like Paclitaxel. For instance, research has shown that glucocorticoids selectively inhibit Paclitaxel-induced apoptosis in various human solid tumor cell lines. Moreover, it is well-established that GR activation induces G1 cell cycle arrest. This has been observed in numerous cell types.

Given that Paclitaxel targets actively cycling cells, particularly those in the M-phase, the induction of G1 arrest by GR activation would theoretically reduce the population of cells susceptible to Paclitaxel's mechanism of action. This raises a conceptual contradiction: how could GR-induced G1 arrest enhance, rather than diminish, the efficacy of a drug whose mechanism relies upon active cell-cycle progression?

Therefore, the authors must explicitly address this discrepancy and provide mechanistic explanations supported by additional experiments. Without this, the claim about the clinical relevance of GR activation as a chemosensitizer remains unsupported and contradicts prior work.

- Rapid ER protein loss (6 hours) following Dex treatment was reported and the authors suggest a potential proteasomal degradation mechanism preceding ESR1 mRNA loss at 8 hours. However, the manuscript lacks direct experimental validation (such as proteasome inhibition assays) to confirm the involvement of proteasomal degradation. Given that the timing and mechanism of ER loss is a crucial aspect of their proposed model, it is important that authors explicitly clarify this mechanism. Either experimental validation of the proteasomal pathway or an explicit acknowledgment of this limitation should be included.

- The authors show synergy between Dex and CDK4/6 inhibitors (Ribociclib, Palbociclib). However, since the efficacy of CDK4/6 inhibitors typically depends upon active ER signaling, this observation appears paradoxical in light of their model, where Dex treatment strongly suppresses ER expression and chromatin binding. This conceptual contradiction must be explicitly discussed. Clarification or additional experimental evidence is required to reconcile how suppression of ER signaling by Dex can still permit or enhance synergy with CDK4/6 inhibitors.

- Dex was used at a concentration (700 nM) significantly above typical clinical plasma levels (commonly <100 nM). While higher concentrations ensure GR activation experimentally, it is critical to discuss the clinical translatability of these concentrations. It would strengthen the manuscript greatly if authors could justify their chosen dose clearly, and ideally, demonstrate whether the observed beneficial effects remain evident at more clinically achievable doses which are most often used in the field (e.g., ~50-100 nM).

- Although the authors clearly state their use of statistical tests and replication strategy, the number of biological replicates is limited for certain experiments (n=1 or n=2 in several key assays!!!). While these are explicitly stated, I still feel that every assay should be done several times (=>3) as in my view it is not acceptable having n =1 at this level of science. Here are some examples:

For the analysis of ER canonical target genes (PGR, TFF1) after Dex treatment, the authors explicitly state using only two biological replicates (n=2) with three technical replicates each, yet they applied statistical analysis (Mann-Whitney test) to these data - which is inappropriate.

Similarly, ESR1 mRNA levels following GR activation were also determined using only two experimental replicates (biological replicates) and three technical replicates each, again using statistics with limited sample size

Even more problematic, in confocal imaging quantification of nuclear ER intensity in MCF-7 D538G organoids and ER+ patient-derived tumor organoids (PDO_5), the authors explicitly used only one biological replicate (n=1) with three technical replicates, yet still performed statistical analysis using ANOVA tests (One-way or Two-way ANOVA with Fisher's LSD test).

- Some similar observations (although less detailed and without the translational depth shown here) were previously reported in the context of ESR1 mutations by Tonsing-Carter et al. (2019, Breast Cancer Research). The authors currently do not explicitly acknowledge this prior report in relation to ER mutant models. Therefore, this prior art should be acknowledged.

Additional Considerations:

- The manuscript nicely identifies early (8 h) versus late (24 h) ER-responsive gene sets that respond to GR activation. However, the authors should explicitly indicate whether these gene sets reflect direct GR repression versus secondary effects resulting from loss of ER itself. Clarifying these distinctions will strengthen mechanistic clarity.

- The discrepancy where Dex enhanced Paclitaxel efficacy in ER+ in vivo models but reduced it in vitro using MCF-7 monolayer cultures is highlighted. However, this difference between monolayer and in vivo conditions should be explicitly discussed as it may indicate context-dependent or model-specific effects, affecting the generalizability of their conclusions.

- The authors appropriately acknowledge the contrasting effects of GR activation-anti-metastatic in ER+ liver metastases versus previously reported pro-metastatic in TNBC lung metastases. Although a brief mechanistic hypothesis involving differential ROR1 regulation is provided, a more detailed discussion addressing organ-specific microenvironmental differences and underlying biological context would significantly enhance clarity and scientific rigor. The authors should acknowledge the limited depth of current explanations and discuss potential mechanistic factors influencing these organ-specific outcomes in more detail.

Referee #2 (Comments on Novelty/Model System for Author):

Cross talk ER/GR not novel, however the fact that GR decreases ER expression is new.

Referee #2 (Remarks for Author):

Manivannan et al demonstrated that the activation of the glucocorticoid receptor by its ligand, the dexamethasone, decreases liver metastases, enhances chemotherapy responsiveness and prolongs survival in ESR1 mutant mice model. Finding new therapeutical options for metastatic breast cancers and ESR1 mutant patients is relevant. Indeed, postmenopausal patients are treated with aromatase inhibitors. However, some patients relapse due to the emergence of mutations in the ESR1 gene encoding ERα, which makes it constitutively active. Fulvestrant (and new SERD) are prescribed in metastatic situations. However, 100% of patients relapse in the metastatic setting. Increasing the therapeutic options is important. In addition, Glucocorticoids are widely used in clinic, and their doses and secondary effects are known, justifying the investigation.

With the information that I had, I could see that the manuscript improved considerably thanks to the previous reviewers, especially with the addition of RNA-seq and Chip-seq analyses.

However, I feel the strength of the paper could be improved by a deeper introduction. As it's more described in the discussion, GC has adverse effects on the TNBC. I guess authors should emphasize breast cancer subtypes and treatments and the fact that TNBC does not express ERα and that could explain in part the differences in GC effects. In addition, some big pieces of evidence of the cross-talk between ER and GR are missing in the introduction. Indeed, Yang et al. show that SUMOylated GR represses ERα-dependent gene and enhancer activity via the disassembly of the coregulator complex (N-CoR/SMRT-HDAC3 complex).

In my point of view, the novelty of this study is based on the study of the crosstalk ER/GR in ESR1 mutant cells, specifically the fact that GR binds to the ESR1 promoter and reduces ER expression. That should be clarified in the title. To my knowledge not all metastatic breast cancers harbor ESR1 mutation.

Finally, to claim the clinical relevance of this work, and the novelty of a new therapeutic option, PDX models of ESR1 mutants should be treated by Dex in combination with CDKi.

Minor comments or clarification :

- Line 240 : beneficial for ESR1 mutant patients
- Please provide the effect of serds on GR expression in figEV3b
- Line 71 : clarification of the type of endocrine therapy authors referred to
- Line 358 : ERS??
- All the GR western-blot are not properly cut. Some are too close to the band (example fig3i)

Referee #1 (Remarks for Author):

Reviewer: The present manuscript by Manivannan et al. addresses an interesting and clinically relevant topic - the potential role of GR activation as a therapeutic strategy in ER+ metastatic breast cancer harboring ESR1 mutations. The authors provide comprehensive transcriptomic, proteomic, and genomic data, suggesting that GR activation by dexamethasone (Dex) significantly suppresses ER expression and transcriptional activity, reducing metastatic burden and enhancing chemotherapy efficacy in preclinical models. Despite these strengths, several important issues and concerns exist. These concerns particularly pertain to clinical applicability, mechanism interpretation, statistics, and certain discrepancies with previously published literature and clinical data.

Authors: We thank the Reviewer for his/her appreciation of our work. Below, we will address in detail the different points raised.

Major Comments

Reviewer: GR activation is suggested to induce marked ER degradation and chromatin dissociation in their preclinical models in the current manuscript. However, careful examination of their own clinical metastatic cohort data (Figure 5c) does not convincingly support a negative correlation between ER and GR abundance. In fact, most patient samples show clear nuclear co-expression of ER and GR, contradicting the notion of complete ER loss following GR activation. This inconsistency between their preclinical data (showing ER silencing/degradation) and actual clinical samples (retaining substantial ER positivity even in the presence of active GR) must be explicitly acknowledged and discussed. It raises significant questions about the clinical translatability of their observations and strongly suggests that their experimental ER loss phenomenon may reflect preclinical model-specific or condition-specific artifacts rather than clinically applicable biology.

Authors: We thank the Reviewer for this comment. To perform this analysis, we used a retrospective metastatic cohort of patients with ER+ disease, registered at the University Hospital Basel. In this cohort, we do not know whether some of these patients received or were receiving glucocorticoids (for other indications), at the time of sampling, nor do we have the data regarding the endogenous level of cortisone, which may also account for GR activation. Hence, we did not expect to see a complete loss of ER, as these patients may not have received Dex treatment. In order to properly assess whether Dex administration to patients induces ER loss, one would need to perform a longitudinal prospective study with sampling before and after Dex treatment, and quantification of ER protein abundance by immuno-histochemistry. This is very important and we are planning such a study, however this is beyond the scope of the present manuscript.

In addition, we generated a GR activity signature from our RNA-Seq experiment and demonstrated that it anti-correlates with *ESR1* expression in the METABRIC cohort, stratified for Luminal tumors: The higher the score of this signature, the lower the expression of *ESR1* and *vice versa* (Fig 5b). The fact that this signature reflects GR activity suggests that our observation in preclinical models is not an experimental artefact.

Reviewer: The finding and the claim that dexamethasone (Dex) treatment enhances Paclitaxel efficacy, improving survival outcomes and reducing metastatic burden is puzzling. This conclusion

is particularly striking and, to some extent, contradictory to the established literature. Prior studies have demonstrated that glucocorticoids can antagonize the effects of chemotherapy agents like Paclitaxel. For instance, research has shown that glucocorticoids selectively inhibit Paclitaxel-induced apoptosis in various human solid tumor cell lines. Moreover, it is well-established that GR activation induces G1 cell cycle arrest. This has been observed in numerous cell types. Given that Paclitaxel targets actively cycling cells, particularly those in the M-phase, the induction of G1 arrest by GR activation would theoretically reduce the population of cells susceptible to Paclitaxel's mechanism of action. This raises a conceptual contradiction: how could GR-induced G1 arrest enhance, rather than diminish, the efficacy of a drug whose mechanism relies upon active cell-cycle progression?

Therefore, the authors must explicitly address this discrepancy and provide mechanistic explanations supported by additional experiments. Without this, the claim about the clinical relevance of GR activation as a chemosensitizer remains unsupported and contradicts prior work.

Authors: We thank the Reviewer for this comment and will provide a detailed answer, together with new data.

In our initial manuscript, we observed that the addition of Dex to Paclitaxel prolongs the survival of NSG mice injected with MCF-7 D538G & MCF-7 Y537S cells engineered with GFP-luciferase. This led us to conclude that there is a benefit in combining these two molecules, and that Dex could sensitize *ESR1* mutant cells to chemotherapeutic agents *in vivo*. This is in contrast to what we and others observed in TNBC models and differs from what has been reported with ER models grown *in vitro*. Importantly, we observed no reduction of the metastatic outgrowth upon Paclitaxel administration to the NSG mice injected with MCF-7 *ESR1* mutant cells (despite a rather high dosage; 25 mg/kg/day), in comparison to vehicle treated animals, indicating that these models were intrinsically resistant to Paclitaxel (Figure Appendix S1a and S1b).

We concur with the points raised by the Reviewer that: 1) we are lacking mechanistic molecular evidence to support our claims, 2) the existing literature rather indicates that Dex, by inducing a G0/G1 arrest, offsets the efficacy of paclitaxel as it kills the cells during the G2/M phase. In order to strengthen our claims and answer the Reviewer's comments, we performed several experiments.

First, we compared the response to Paclitaxel *in vitro*, in presence or absence of Dex, using an SRB assay. We observed that the MCF-7 D538G cells were very sensitive to Paclitaxel *in vitro*, as a 10 nM dose kills > 75% of the cells after 3 days of treatment (**Figure for the Reviewers 1a**). This is surprising as we observed no response to Paclitaxel *in vivo*, which suggested that the metastatic MCF-7 D538G cells in the liver can resist Paclitaxel killing, through an unknown mechanism. In comparison, we found that Dex alone reduces cell number *in vitro* and decreases the metastatic burden *in vivo*.

Second, in terms of Paclitaxel sensitivity, the SRB assay revealed that Dex-treated MCF-7 D538G cells were less sensitive to Paclitaxel than vehicle-treated cells (**Figure for the Reviewers 1a-c**). This held true whether Dex and Paclitaxel were added simultaneously or successively (Dex then Paclitaxel). These *in vitro* data therefore indicate that Dex treatment offsets the efficacy of Paclitaxel in MCF-7 D538G cells, similar to what we observed in TNBC cells (**Figure for the Reviewers 1d-e**), which thus cannot explain our *in vivo* data.

Third, to assess the effects of Dex and Paclitaxel treatments on cell cycle progression, we performed an EdU incorporation assay (**Figure for the Reviewers 1f-g**). Interestingly, we observed no G0/G1 arrest upon treatment with Dex alone in the MCF-7 D538G, but only a reduction of the percentage of cells in S-phase. However, upon treatment with Paclitaxel alone we observed that while majority of the cells were arrested in G2/M, a significant fraction of the Dex-treated cells remained arrested in G0/G1 phase and did not enter G2M phase, where paclitaxel exerts its toxicity. It therefore appears, as the Reviewer suggested, that Dex-treated

cells are less sensitive to Paclitaxel because they remain arrested in G0/G1 and do not enter the G2/M phase.

Based on these observations, *in vitro* assays are thus inadequate to explain our *in vivo* observation. Our conclusions are the following: as the MCF-7 D538G cells are resistant to Paclitaxel *in vivo*, it is possible that the reduction of the metastatic burden is only due to the action of Dex rather than a synergistic or a chemosensitizing effect. Based on the newly generated data, we agree with the Reviewer that as such, we cannot claim that Dex acts as a chemosensitizer. We therefore removed that claim and present now the data related to Paclitaxel in the Appendix Figure 1, where we report that Dex can still reduce liver metastases in the presence of chemotherapy.

Figure for the Reviewers 1: Dex induces resistance to Paclitaxel in MCF-7 D538G cells grown *in vitro*.

a. Paclitaxel dose response using SRB assay in MCF-7 D538G cells in the presence or absence of 700 nM Dex for 3 days. Cells were simultaneously treated with Paclitaxel and Dex for 3 days. n = 3 biological replicates. **b.** Paclitaxel dose response using SRB assay in MCF-7 D538G cells in the presence or absence of 700 nM Dex for 3 days. Prior to Paclitaxel treatment, cells were pre-exposed with Dex for 3 days. n = 3 biological replicates. **c.** Quantification of Paclitaxel response in presence or absence of Dex at the indicated concentrations, in the experimental set-up of panel a. n = 3 biological replicates. ns, non-significant, **P < 0.01. Unpaired t-test. Data are presented as mean ± SD. **d.** Paclitaxel dose response using SRB assay in MDA-MB-468 cells in the presence or absence of 700 nM Dex for 3 days. Prior to Paclitaxel treatment, cells were pre-exposed with Dex for 3 days. n = 3 biological replicates. **e.** Quantification of Paclitaxel response in presence or absence of Dex at the indicated concentrations, in the experimental set-up of panel a. n = 3 biological replicates. ns, non-significant, **P < 0.01. Unpaired t-test. Data are presented as mean ± SD. **f.** Click EdU assay in MCF-7 D538G in presence or absence of 700 nM Dex for 4 days, and in the presence or absence of 100 nM Paclitaxel for the last 24h. The proportion of the cells in the different phases of the cells are shown in three separate graphs. n = 3 biological replicates. ns, non-significant, *P < 0.05, **P < 0.01, *** P < 0.001. Ordinary ANOVA test. Data are presented as mean ± SD.

Reviewer: Rapid ER protein loss (6 hours) following Dex treatment was reported and the authors suggest a potential proteasomal degradation mechanism preceding ESR1 mRNA loss at 8 hours. However, the manuscript lacks direct experimental validation (such as proteasome inhibition assays) to confirm the involvement of proteasomal degradation. Given that the timing and mechanism of ER loss is a crucial aspect of their proposed model, it is important that authors explicitly clarify this mechanism. Either experimental validation of the proteasomal pathway or an explicit acknowledgment of this limitation should be included.

Authors: We thank the Reviewer for this comment. We have now performed a proteasome inhibition assay as suggested, and found that ER protein abundance was not rescued when MG-132 was added simultaneously to Dex, *in vitro*. As a positive control, we observed a rescue of GR protein abundance upon Dex and MG-132 treatment. The results of 3 independent immuno-blot experiments are presented below (**Figure for the Reviewers 2**). We also refined the timing of ER protein loss following Dex treatment, and found that it starts around 8 h, concomitantly with *ESR1* silencing (**New Fig EV3e**, Fig 3f). These data therefore rule out the possibility of an ER protein degradation triggered by Dex. We have amended our discussion accordingly.

Figure for the Reviewers 2: MG-132 treatment does not rescue ER protein loss upon Dex. Immunoblots showing levels of ER, GR and ERK2 (loading control) in MCF-7 D538G cells treated with Dex (700 nM), and/or MG-132 (2 μ M) for 16h and the respective ER quantification. n = 3 biological replicates. Ordinary ANOVA test. Data are presented as mean \pm SD.

Reviewer: The authors show synergy between Dex and CDK4/6 inhibitors (Ribociclib, Palbociclib). However, since the efficacy of CDK4/6 inhibitors typically depends upon active ER signaling, this observation appears paradoxical in light of their model, where Dex strongly suppresses ER expression and chromatin binding. This conceptual contradiction must be explicitly discussed. Clarification or additional experimental evidence is required to reconcile how suppression of ER signaling by Dex can still permit or enhance synergy with CDK4/6 inhibitors.

Authors: We thank the Reviewer for his/her comment. For patients with ER+ endocrine resistant breast cancer, CDK4/6i are always given in combination with aromatase inhibitors (AI) or the Selective Estrogen Receptor Degradar (SERD) fulvestrant. These molecules inhibit ER signaling in order to create a synergy with CDK4/6 inhibitors, which are never administered alone to patients. In our study, Dex also inhibits ER signaling, yet with a different mode of action in comparison to classical SERDs or AI. Hence, CDK4/6 inhibitors do not really depend on active ER signaling, their efficacy rather depends on the combination with a treatment that inhibits ER signaling. Please advise on what we should include in the discussion if we misunderstood your comment.

Reviewer: Dex was used at a concentration (700 nM) significantly above typical clinical plasma levels (commonly <100 nM). While higher concentrations ensure GR activation experimentally, it is critical to discuss the clinical translatability of these concentrations. It would strengthen the manuscript greatly if authors could justify their chosen dose clearly, and ideally, demonstrate whether the observed beneficial effects remain evident at more clinically achievable doses which are most often used in the field (e.g., ~50-100 nM).

Authors: We thank the Reviewer for raising this point. Admittedly, we used Dex at 700 nM based on previous preclinical reports. We also agree with the Reviewer that the clinical translatability of the findings is crucial. To address this, we tested whether 50 nM and 100 nM concentrations of Dex *in vitro*, similar to the 700 nM dose, would reduce cell number and expression of *ESR1* and

ER target genes such as *TFF1*, *AREG*, *PGR* using qPCR. We found that both 50 nM and 100 nM doses of Dex reduce cell number and induce ER loss to a similar extent as the 700 nM dose (Figure for the Reviewers 3).

Figure for the Reviewers 3: 50 nM and 100 nM Dex doses reduce MCF-7 D538G cell number and repress *ESR1* and of ER target genes expression.

a. Dex dose response using SRB assay in MCF-7 D538G cells in the presence or absence of 700 nM, 100 nM and 50 nM Dex for 6 days. $n = 3$ biological replicates. $***P < 0.001$. ANOVA. **b.** Scatter dot plots showing the mRNA levels of *ESR1* and ER targets (*TFF1*, *AREG*, *PGR*) and GR target (*SGK1*) after 24 h of Dex treatment in MCF-7 D538G cells: $n = 2$ biological replicates. ns, non-significant, $*P < 0.05$, $**P < 0.01$; $***P < 0.001$. Unpaired t-test. Data are presented as mean \pm SD.

Reviewer: Although the authors clearly state their use of statistical tests and replication strategy, the number of biological replicates is limited for certain experiments ($n=1$ or $n=2$ in several key assays!!!). While these are explicitly stated, I still feel that every assay should be done several times ($\Rightarrow 3$) as in my view it is not acceptable having $n = 1$ at this level of science.

Here are some examples:

For the analysis of ER canonical target genes (*PGR*, *TFF1*) after Dex treatment, the authors explicitly state using only two biological replicates ($n=2$) with three technical replicates each, yet they applied statistical analysis (Mann-Whitney test) to these data - which is inappropriate.

Similarly, *ESR1* mRNA levels following GR activation were also determined using only two experimental replicates (biological replicates) and three technical replicates each, again using statistics with limited sample size.

Authors: We thank the Reviewer for spotting these errors. We now have three biological replicates for the qPCR data in Figure 3c, Figure 3f, Figure EV3 and Figure 4. We now present the data as -Delta Ct and perform t-test to assess statistical significance for each treatment timepoint.

Reviewer: Even more problematic, in confocal imaging quantification of nuclear ER intensity in MCF-7 D538G organoids and ER+ patient-derived tumor organoids (PDO_5), the authors explicitly used only one biological replicate (n=1) with three technical replicates, yet still performed statistical analysis using ANOVA tests (One-way or Two-way ANOVA with Fisher's LSD test).

Authors: We thank the Reviewer for spotting these errors. We have now provided three biological replicates for the quantification of ER in MCF-7 D538G organoids, upon different treatments (**New Figure 5e and EV5c**), and used ANOVA to infer statistical significance. For the quantification of ER in ER+ PDO and PDXO, we have quantified ER in 2 newly generated PDO and 1 new PDXO. Therefore, we now provide 2 independent graphs, one for ER quantification in PDO, the other one for ER quantification in PDXO. These graphs are based on 3 different biological replicates, each of these replicates representing individual patient-derived samples. Hence, we now have 6 patient-derived samples, (3 PDOs, 3 PDXOs), and have used ANOVA to infer statistical significance (**New Figure 5f and 5g**).

Reviewer: Some similar observations (although less detailed and without the translational depth shown here) were previously reported in the context of ESR1 mutations by Tonsing-Carter et al. (2019, Breast Cancer Research). The authors currently do not explicitly acknowledge this prior report in relation to ER mutant models. Therefore, this prior art should be acknowledged.

Authors: We thank the Reviewer for this comment. We have cited this publication from the Conzen lab multiple times and discussed it more in depth now, especially their findings related to primary tumor settings in the presence of selective GR modulators, and related to the transcription factor binding competition between ER and GR at the vicinity of the ER target gene, *CCND1*.

Additional Considerations:

Reviewer: The manuscript nicely identifies early (8 h) versus late (24 h) ER-responsive gene sets that respond to GR activation. However, the authors should explicitly indicate whether these gene sets reflect direct GR repression versus secondary effects resulting from loss of ER itself. Clarifying these distinctions will strengthen mechanistic clarity.

Authors: We thank the Reviewer for this comment. To answer this question, we used our RNA-Seq genesets and overlapped the genes down-regulated upon Dex at 8 h and 24 h with the HALLMARK gene signature for E2 response, to identify which ER targets are down-regulated. We therefore have 2 genesets : 1) The ER target genes down-regulated post 8 h of Dex treatment (including *ESR1* itself, *PGR*, *CXCL12*, etc.), 2) the ER target genes down-regulated post 24 h of Dex (including *AREG*, *TFF1*, etc.). Of note, all the genes (except 1) that were down-regulated at 8 h of Dex remained down-regulated at 24 h Dex, with a decreased - Log FC at 24 h of Dex, indicating that repression of ER target genes is stronger upon prolonged Dex exposure (**Figure for the Reviewers 4**). To identify the genes likely to be repressed by GR among these down-regulated ER target genes, we used our GR ChIP-Seq dataset following Dex stimulation and

scored whether any genes belonging to each of these two signatures located at the vicinity of the GR peaks (in promoter, exonic, intronic and intergenic regions). We found that 50 and 29% of these early repressed and late repressed genes are located at the direct vicinity of a GR peak, respectively. These results indicate that the early repression is more likely due to an active GR binding. Also, it is possible that GR peaks located far from the gene of interest may contribute to repression. We have included this new analysis into our manuscript (**New Fig EV4f**). For the genes whose downregulation may result from ER loss, our ER ChIP-Seq experiment revealed a dramatic eradication of ER genome-wide (Fig 4c). This clearly indicates that ER loss may contribute to downregulation of all late repressed ER targets, while at 8 h of Dex, where ER protein loss only starts, the early repressed genes are less likely to be affected. We have amended our results and discussion sections accordingly.

FC comparison 8h versus 24h Dex

Figure for the Reviewers 4: Comparison of ER target genes averaged Log FC upon 8h and 24h of Dex.

Reviewer: The discrepancy where Dex enhanced Paclitaxel efficacy in ER+ in vivo models but reduced it in vitro using MCF-7 monolayer cultures is highlighted. However, this difference between monolayer and in vivo conditions should be explicitly discussed as it may indicate context-dependent or model-specific effects, affecting the generalizability of their conclusions.

Authors: We thank the Reviewer for this comment, and invite him/her to refer to our reply in the previous comment about Dex and Paclitaxel.

Reviewer: The authors appropriately acknowledge the contrasting effects of GR activation-anti-metastatic in ER+ liver metastases versus previously reported pro-metastatic in TNBC lung metastases. Although a brief mechanistic hypothesis involving differential ROR1 regulation is provided, a more detailed discussion addressing organ-specific microenvironmental differences and underlying biological context would significantly enhance clarity and scientific rigor. The authors should acknowledge the limited depth of current explanations and discuss potential mechanistic factors influencing these organ-specific outcomes in more detail.

Authors: We thank the Reviewer for this comment. We have deepened our discussion in this regard and have underlined the limitations of our study.

Referee #2 (Comments on Novelty/Model System for Author):

Cross talk ER/GR not novel, however the fact that GR decreases ER expression is new.

Referee #2 (Remarks for Author):

Reviewer: Manivannan et al demonstrated that the activation of the glucocorticoid receptor by its ligand, the dexamethasone, decreases liver metastases, enhances chemotherapy responsiveness and prolongs survival in ESR1 mutant mice model.

Finding new therapeutical options for metastatic breast cancers and ESR1 mutant patients is relevant. Indeed, postmenopausal patients are treated with aromatase inhibitors. However, some patients relapse due to the emergence of mutations in the ESR1 gene encoding ER α , which makes it constitutively active. Fulvestrant (and new SERD) are prescribed in metastatic situations. However, 100% of patients relapse in the metastatic setting. Increasing the therapeutic options is important. In addition, Glucocorticoids are widely used in clinic, and their doses and secondary effects are known, justifying the investigation.

Authors: We thank the Reviewer for his/her appreciation of our work, and for underlining its clinical relevance. We will address in detail the different points raised below.

Reviewer: With the information that I had, I could see that the manuscript improved considerably thanks to the previous reviewers, especially with the addition of RNA-seq and Chip-seq analyses. However, I feel the strength of the paper could be improved by a deeper introduction. As it's more described in the discussion, GC has adverse effects on the TNBC. I guess authors should emphasize breast cancer subtypes and treatments and the fact that TNBC does not express ER α and that could explain in part the differences in GC effects. In addition, some big pieces of evidence of the cross-talk between ER and GR are missing in the introduction. Indeed, Yang et al. show that SUMOylated GR represses ER α -dependent gene and enhancer activity via the disassembly of the coregulator complex (N-CoR/SMRT-HDAC3 complex).

Authors: We thank the Reviewer for his/her suggestions. We have modified our introduction accordingly, reinforcing the importance of the breast cancer subtypes when studying the effect of glucocorticoids. Also, we have deepened our discussion, especially in regards to the aforementioned study. Indeed, our data uncovering a direct GR-mediated repression of *ESR1* highlights a novel mechanism through which GR can repress ER-dependent gene transcription, independently of SUMO-GR mediated recruitment of co-repressors.

Reviewer: In my point of view, the novelty of this study is based on the study of the crosstalk ER/GR in ESR1 mutant cells, specifically the fact that GR binds to the ESR1 promoter and reduces ER expression. That should be clarified in the title. To my knowledge not all metastatic breast cancers harbor ESR1 mutation.

Authors: We thank the Reviewer for his/her suggestions. Indeed the novelty of the study lies in the fact that we used metastatic *ESR1* mutant cell lines, however, we detected that GR activation by Dex induces ER loss in *ESR1* wild-type cell lines as well. Therefore, we would rather keep the title as such, as we think it nicely conveys the core message of our study.

Reviewer: Finally, to claim the clinical relevance of this work, and the novelty of a new therapeutic option, PDX models of ESR1 mutants should be treated by Dex in combination with CDKi.

Authors: We thank the Reviewer for his/her comments. To answer this question, we have performed a metastasis assay, by injecting MCF-7 D538G GFP luciferase cells intravenously into NSG mice (**Figure for the Reviewers 5**). The timeframe of the revision was incompatible with an *in vivo* experiment using PDXs, hence the use of a mutant cell line as a surrogate. Two weeks following cell injection, we treated the animals for 4 weeks with either 1) vehicle, 2) Dex only (0.1 mg/kg/day), 3) Dex + Ribociclib (25 mg/kg/day) or 4) Fulvestrant (200 mg/kg/day) + Ribociclib. We have quantified the metastatic burden in the liver by bioluminescence imaging. We observed that the addition of Ribociclib increases the efficacy of Dex or Fulvestrant in comparison to vehicle treatment. Additional pharmacokinetics experiments are required to optimize the dosage of Dex in combination with Ribociclib *in vivo*. Indeed, other publications have used Dex *in vivo* at a much higher dose (4 mg/kg/day; Prekovic *et al.*, Nature Comm, 2021).

Figure for the Reviewers 5: Combination of Dex with CDK4/6i Ribociclib in NSG mice intravenously injected with MCF-7 D538G GFP luciferase cells.

a. Representative bioluminescence images of animals at week 6 in the four treatment arms. b. Bar graph showing the quantification of liver metastases at week 6. n = 7 to 8 mice per group. Ordinary One-way ANOVA test. Data are presented as mean ± SD.

Minor comments or clarification :

Reviewer: Line 240 : beneficial for ESR1 mutant patients

Authors: We thank the Reviewer for his/her comment and have modified it accordingly.

Reviewer: Please provide the effect of serds on GR expression in figEV3b

Authors: We thank the Reviewer for his/her comments. We have added a new immunoblot showing the effects of SERDs on GR protein abundance (Fig EV3b) and found no effect.

Reviewer: Line 71 : clarification of the type of endocrine therapy authors referred to

Authors: We thank the Reviewer for his/her comments. We clarified the endocrine therapy that we were referring to.

Reviewer: Line 358 : ERS??

Authors: We defined “ERS” as an acronym for “Estrogen Receptor Silencer” line 342 to describe GR effect of ER expression.

Reviewer: All the GR western-blot are not properly cut. Some are too close to the band (example fig3i)

Authors: We replaced the aforementioned blot as suggested by the Reviewer, and made sure our GR blots are properly cut.

5th Sep 2025

Dear Prof. Bentires-Alj, Dear Momo,

Thank you for submitting your revised manuscript to EMBO Molecular Medicine, and please accept my apologies for the delay in getting back to you as I was on annual leave when the referee report was submitted. Referee #1 reviewed your revised manuscript in light of the issues mentioned by both referees, as Referee #2 was unavailable at this time. As you will see below, Referee 1 acknowledges the work that has been done but also raises some concerns that should be addressed in an additional round of revisions. In particular, the referee indicates that the GR activity signature should be removed from the manuscript and that the lack of clinical evidence of GR down-regulating ER in breast cancer should be acknowledged.

If you would like to discuss further these points, I am available to do so via email or video. Let me know if you are interested in this option.

As EMBO Press usually only allows one round of revisions, please be aware that this will be your last opportunity to address these issues. The revised manuscript will be reviewed again, and we cannot guarantee a positive outcome at this stage.

Moreover, please address the following editorial requests:

- 1/ Please include in the manuscript file the keywords and Paper Explained.
- 2/ Please replace the heading "Main" in the manuscript text with "Introduction".
- 3/ An email bounced for Katrin Volkmann - katrin.volkmann@unibas, please check and correct if needed. There is a discrepancy in the author's name for Simone Muenst (manuscript) vs. Simone Münst (submission system), please correct as needed.
- 4/ Data Availability section: please add the Zenodo link.
- 5/ Acknowledgements: please note that the funding information provided in the system and in the manuscript should match - Should the Dr. Arnold U. and Susanne Huggenberger-Bischoff Foundation, the Excellence Junior Research Fund of the University of Basel, the Swiss National Science Foundation, the Swiss Personalized Health Network (Swiss Personalized Oncology driver project), and the Department of Surgery, University Hospital Basel also be added as funders to our system?
- 6/ Please remove the Authors Contributions from the manuscript and use the free text boxes beneath each contributing author's name in our system to add specific details on the author's contribution.
- 7/ Please rename the conflict of interest to "Disclosure statement and competing interest".
- 8/ References: please correct the heading to "References" and the format to alphabetical order, 10 author names listed before et al.
- 9/ The Datasets need legends added to the file in a separate tab/worksheet.
- 10/ Appendix: the legends need to be corrected to "Appendix Figure S1" etc.
- 11/ Source Data: please submit with a completed checklist and upload the files as one Zip file per figure.
- 12/ Please address the queries from our data editors in the figure legends:
 - Please note that the exact p values are not provided in the legends of figures 1F, 2A, B, D; 3G, 5A, B, C, E; EV3 F, EV5 A, EV6 A, G; S1 A, F, S4 B
 - Please indicate the statistical test used for data analysis in the legends of figures 3D, E; EV2 C

As part of the EMBO Publications transparent editorial process initiative (see our Editorial at <http://embomolmed.embopress.org/content/2/9/329>), EMBO Molecular Medicine will publish online a Review Process File (RPF) to accompany accepted manuscripts.

This file will be published in conjunction with your paper and will include the anonymous referee reports, your point-by-point response and all pertinent correspondence relating to the manuscript. Let us know whether you agree with the publication of the RPF and as here, if you want to remove or not any figures from it prior to publication.

I look forward to receiving your revised manuscript.

With kind regards,

Lise

**** Reviewer's comments ****

Referee #1 (Remarks for Author):

The authors have done a good job addressing most of the remarks; however, I feel that certain points still require clarification and that some major issues remain unresolved. On behalf of Reviewer 2, I have also reviewed their comments and your responses (as requested by the editor), and I agree that these have been addressed thoroughly and satisfactorily.

Major remark:

- The authors need to clarify that their "GR activity signature" may not reflect direct action of GR, but rather secondary downstream gene activation as it was based on proteomic measurements. While this in itself would be acceptable, the inclusion of KRT5, KRT16, and KRT6A/B/C in the signature is highly problematic. These keratins are canonical basal markers and likely drive much of the observed signal, which would explain the apparent reduction in ER (a well-established feature of basal versus luminal breast cancers, independent of GR). Importantly, in the context of breast cancer there is no evidence that GR directly induces expression of these keratins; if anything, multiple studies have demonstrated that GR negatively regulates basal programs and reinforces luminal profiles. This represents a major flaw in the current interpretation and should be directly addressed, with the keratin-driven signature removed from the manuscript. Furthermore, the authors should clearly acknowledge that no clinical proof exists for GR down-regulating ER in breast cancer.

Therefore, the previous point that I raised has not been properly addressed. However, as I do believe that the manuscript provides some advancements for the field, I would be fine with publication provided that the signature is removed and the limitations, including the lack of clinical proof, are explicitly discussed.

Minor remark:

- Thank you for the reply in relation to the CDK4/6 inhibitor comment. would like to clarify my concern. The clinical efficacy of CDK4/6 inhibitors has been consistently demonstrated in ER+ breast cancers. Mechanistically, ER signaling up-regulates Cyclin D1, which activates the CDK4/6-Rb pathway; CDK4/6 inhibitors act by blocking this axis. Clinical trials such as PALOMA-2 and PALOMA-3 established efficacy specifically in ER+ disease, where ER signaling remains present, albeit attenuated by AI or SERD treatment. There is no evidence of comparable benefit in ER- cancers. In contrast, your model proposes that Dex strongly suppresses ER expression and chromatin binding. This appears closer to complete suppression, which conceptually conflicts with the requirement for residual ER signaling in CDK4/6 inhibitor efficacy. For this reason, the explanation that CDK4/6 inhibitors "do not really depend on active ER signaling" is not consistent with the literature. I encourage the authors to explicitly discuss this paradox in the manuscript and to clarify whether Dex truly abolishes ER signaling or only partially attenuates it. If the observed synergy occurs through an ER-independent mechanism, this should be clearly stated and, where possible, supported experimentally.

- Finn RS, Martin M, Rugo HS, et al. Palbociclib and Letrozole in Advanced Breast Cancer. *N Engl J Med.* 2016;375(20):1925-1936. doi:10.1056/NEJMoa1607303

- Turner NC, Ro J, André F, et al. Palbociclib in Hormone-Receptor-Positive Advanced Breast Cancer. *N Engl J Med.* 2015;373(3):209-219. doi:10.1056/NEJMoa1505270

- O'Leary B, Finn RS, Turner NC. Treating cancer with selective CDK4/6 inhibitors. *Nat Rev Clin Oncol.* 2016;13(7):417-430. doi:10.1038/nrclinonc.2016.26

EMM-2025-21509-V2
9th September 2025
Reply to the Reviewer

Reviewer: The authors have done a good job addressing most of the remarks; however, I feel that certain points still require clarification and that some major issues remain unresolved. On behalf of Reviewer 2, I have also reviewed their comments and your responses (as requested by the editor), and I agree that these have been addressed thoroughly and satisfactorily.

Reviewer: The authors need to clarify that their "GR activity signature" may not reflect direct action of GR, but rather secondary downstream gene activation as it was based on proteomic measurements. While this in itself would be acceptable, the inclusion of KRT5, KRT16, and KRT6A/B/C in the signature is highly problematic. These keratins are canonical basal markers and likely drive much of the observed signal, which would explain the apparent reduction in ER (a well-established feature of basal versus luminal breast cancers, independent of GR). Importantly, in the context of breast cancer there is no evidence that GR directly induces expression of these keratins; if anything, multiple studies have demonstrated that GR negatively regulates basal programs and reinforces luminal profiles. This represents a major flaw in the current interpretation and should be directly addressed, with the keratin-driven signature removed from the manuscript. Furthermore, the authors should clearly acknowledge that no clinical proof exists for GR down-regulating ER in breast cancer.

Author: We thank the Reviewer for his/her comment, and would like to kindly clarify some of questions he/she may have regarding the validity of our GR activity signature. This signature was generated using RNA-Seq data and not proteomics as mentioned by the Reviewer in the Report. In the experiment, MCF-7 ESR1 D538G and Y537S cell lines were both treated with Dex for 8 h (to capture direct GR targets) and 24 h. The genes in the GR signature showed strong early expression at 8 h after GR activation that was sustained through the late timepoint (24h). The technical details of how the GR signature was established are provided in the Methods section (page 33):

"To obtain a GR activity signature, 341 protein-coding genes with the strongest shared effect at 24 h ($|\logFC| > 2$, $FDR < 0.01$) for both MCF7- D538G and Y537S models were selected. Further, these genes were clustered across all 24 samples using k-means clustering with $k = 4$ and mean-centered \logCPM values. The cluster "cl_2" containing a subset of 52 genes with the strongest expression (upregulation) onset between the early timepoint (8 h) and the control samples was further shortlisted defining the GR activity signature."

We apologize if this was not clear in the current version of the manuscript, and will modify the text, legends and methods accordingly. Indeed, we found some basal-associated keratins (KRT5, KRT16, and KRT6A/B/C) upregulated by Dex. These genes may appear surprising for the Reviewer but our signature also encompasses very classical GR targets including *SGK1*, *FKBP5*, *ZBTB16* (reported in Prekovic *et al.*, 2023 EMBO Mol med), which are among the top up-regulated genes by Dex. This provides high confidence that our signature is representative of GR activity in ER+ cell lines, and that the genes comprising it are not secondary downstream targets (by experimental design: RNA-Seq post 8 h Dex). Importantly, the predictive power of our GR activity signature and its anti-correlation with *ESR1* expression (using the METABRIC cohort) has been assessed on patients with Luminal (A/B) breast cancer only. TNBC patients (hence including basal-like) were analyzed separately (Appendix Figure S4), and therefore do not interfere with the predictive power of the signature in Luminal patients, nor bias the observed effect. Nevertheless, to address the comment of the Reviewer and to definitively rule out that the effects of our signature may be driven by these basal-associated keratin-encoding genes (KRT5, KRT16, and KRT6A/B/C), we removed them from the signature and repeated the analysis. The results show that the predictive power of the signature in Luminal patients remains similar (**Figure 1**), and that the

presence of these keratin-encoding genes is not driving the effect. However, since our signature has been unbiasedly and computationally derived, we think that removing these genes would be cherry picking and a manipulation of our data, and therefore would like to keep our data as initially presented.

Figure 1: Kaplan Meier survival plot showing GR activity signature (without keratin-encoding genes) predictive value in patients with Luminal A, Luminal B or ER+ luminal breast cancer (METABRIC).

We would also like to bring the attention of the Reviewer to the Appendix Figure 4 panel A, where we present the results of a multivariate Cox regression on ER+ patient METABRIC cohort. This approach considers additional covariates that might contribute to the differences in survival (here it's age of the patient and luminal status). Low GR activity was associated with a 39% increase in the hazard of death (HR = 1.39).

We hope these explanations address the comments of the Reviewer.

Therefore, the previous point that I raised has not been properly addressed. However, as I do believe that the manuscript provides some advancements for the field, I would be fine with publication provided that the signature is removed and the limitations, including the lack of clinical proof, are explicitly discussed.

Reviewer: Thank you for the reply in relation to the CDK4/6 inhibitor comment. would like to clarify my concern. The clinical efficacy of CDK4/6 inhibitors has been consistently

demonstrated in ER+ breast cancers. Mechanistically, ER signaling up-regulates Cyclin D1, which activates the CDK4/6-Rb pathway; CDK4/6 inhibitors act by blocking this axis. Clinical trials such as PALOMA-2 and PALOMA-3 established efficacy specifically in ER+ disease, where ER signaling remains present, albeit attenuated by AI or SERD treatment. There is no evidence of comparable benefit in ER- cancers. In contrast, your model proposes that Dex strongly suppresses ER expression and chromatin binding. This appears closer to complete suppression, which conceptually conflicts with the requirement for residual ER signaling in CDK4/6 inhibitor efficacy. For this reason, the explanation that CDK4/6 inhibitors "do not really depend on active ER signaling" is not consistent with the literature. I encourage the authors to explicitly discuss this paradox in the manuscript and to clarify whether Dex truly abolishes ER signaling or only partially attenuates it. If the observed synergy occurs through an ER-independent mechanism, this should be clearly stated and, where possible, supported experimentally.

Author: We thank the Reviewer for his/her comment and better understand the nuance that he/she wanted to underline. Despite the strong inhibitory effects of Dex that we observe on ER expression and transcriptional activity, the suppression of ER signaling is arguably not complete. Indeed upon Dex, we still observe residual ER chromatin binding (ChIP-Seq), residual protein expression to a similar extent as fulvestrant (Western-blot), and not a full loss of expression of ER targets (RNA-Seq). We apologize if this was not clear, and will clarify our claims and discuss it accordingly in the manuscript.

EMM-2025-21509-V2
12th September 2025
Reply to the Reviewer:

Reviewer: I understand that the survival analysis didn't change and that is good I do appreciate that data which is completely consistent with previously published reports. My comment was more on the correlation of GR activity and ER levels (so figure/panel 5b). That they don't address that – so the question still remains what happens to ER level correlations if they remove the KRT genes? This is what I find concerning, not the relationship to survival. So I do feel that basal KRTs shouldn't be in their signature anyway and they should rerun the analysis for the ER levels without.

Author: To answer the Reviewer's comment, we used a GR activity signature from which we removed the keratin-encoding genes (as previously) and repeated the correlation with *ESR1* mRNA expression in the METABRIC cohort (within ER+ patients only) (equivalent of Fig 5b). The data show similar results, with high GR activity correlating with lower *ESR1* expression and vice-versa (**Figure 1**), indicating that the keratins are not driving the observed effect. We are happy to provide these new analyses in supplementary material of our manuscript or to let them accessible through the peer-reviewing report made publicly available by the journal.

Figure 1: Graph showing *ESR1* mRNA expression in breast tumors from patients with ER+ disease (METABRIC), stratified according to the GR activity signature (without keratins) score; low, intermediate, high. Pairwise Welch's t-tests between the indicated groups.

15th Oct 2025

Dear Prof. Bentires-Alj, Dear Momo,

Thank you for submitting your revised study. Referee #1 has reviewed your manuscript and is satisfied with the revisions. I will therefore be able to accept your manuscript once the following minor editorial concerns are addressed:

1/ Manuscript text:

- Please indicate in track changes mode any new modification in the text.
- In the methods, please provide the following information:
 - o Patient material: If collected and within the bounds of privacy constraints report on age, sex and gender or ethnicity for all study participants. Include the full statement confirming that the experiments conformed to the principles set out in the WMA Declaration of Helsinki and the Department of Health and Human Services Belmont Report.
- Data availability section: Please note that the datasets must be publicly available before acceptance of the manuscript.
- Acknowledgements: Please ensure that the information in the submission system and the manuscript are consistent.
- Disclosure statement and competing interest: please include the following sentence: "M.B.A. is an editorial advisory board member."
- Figure legends: Please address the query from our data editors in the figure legends: indicate the statistical test used for data analysis in the legends of figures 3D, E; EV2 C

2/ Checklist, section 'Human research participants': this section is not related to patient consent and Helsinki declaration ("If collected and within the bounds of privacy constraints report on age, sex and gender or ethnicity for all study participants"); please correct.

We note that you agree with the publication of the Review Process File (RPF).

I look forward to receiving your revised manuscript at your earliest convenience.

With kind regards,

Lise

***** Reviewer's comments *****

Referee #1 (Remarks for Author):

I would like to thank the authors for their professional and thorough revision. The manuscript has been substantially improved, and I am pleased to endorse it for publication.

The authors addressed the remaining editorial issues.

24th Oct 2025

Dear Prof. Bentires-Alj, Dear Momo,

Thank you for submitting your revised files. I am pleased to inform you that your manuscript is accepted for publication and is now being sent to our publisher to be included in the next available issue of EMBO Molecular Medicine.

With kind regards,

Lise
